## OPEN

# Genomic and functional analyses of fungal and bacterial consortia that enable lignocellulose breakdown in goat gut microbiomes

Xuefeng Peng[1,2], St. Elmo Wilken[1], Thomas S. Lankiewicz[1,3], Sean P. Gilmore[1], Jennifer L. Brown[1], John K. Henske[1], Candice L. Swift [1], Asaf Salamov[4], Kerrie Barry[4], Igor V. Grigoriev [4], Michael K. Theodorou[5], David L. Valentine [6] and Michelle A. O'Malley [1,3] ✉

The herbivore digestive tract is home to a complex community of anaerobic microbes that work together to break down lignocellulose. These microbiota are an untapped resource of strains, pathways and enzymes that could be applied to convert plant waste into sugar substrates for green biotechnology. We carried out more than 400 parallel enrichment experiments from goat faeces to determine how substrate and antibiotic selection influence membership, activity, stability and chemical productivity of herbivore gut communities. We assembled 719 high-quality metagenome-assembled genomes (MAGs) that are unique at the species level. More than 90% of these MAGs are from previously unidentified herbivore gut microorganisms. Microbial consortia dominated by anaerobic fungi outperformed bacterially dominated consortia in terms of both methane production and extent of cellulose degradation, which indicates that fungi have an important role in methane release. Metabolic pathway reconstructions from MAGs of 737 bacteria, archaea and fungi suggest that cross-domain partnerships between fungi and methanogens enabled production of acetate, formate and methane, whereas bacterially dominated consortia mainly produced short-chain fatty acids, including propionate and butyrate. Analyses of carbohydrate-active enzyme domains present in each anaerobic consortium suggest that anaerobic bacteria and fungi employ mostly complementary hydrolytic strategies. The division of labour among herbivore anaerobes to degrade plant biomass could be harnessed for industrial bioprocessing.

Microbial consortia in the herbivore digestive tract have co-evolved with their hosts to utilize lignocellulosic hydrolysates for conversion into short-chain fatty acids (SCFCs)[1]. Although herbivore gut microbiomes contain archaea, bacteria and eukaryotic microorganisms, most biomass-degrading enzymes and microbial genomes from herbivore gut microbiomes are bacterial. This is because bacteria are the most abundant members of the community[2]. Advances in sequencing technology over the past decade have spurred investigations of the herbivore microbiome[2–5]. However, these studies are limited by a lack of mechanistic insight into the chemical productivity displayed by diverse anaerobic consortia. Rare microbes are difficult to cultivate, sequence and characterize[2,5], and non-bacterial microbial relationships in gut consortia have not been well studied. For example, anaerobic gut fungi were recently recognized to possess a wide range of biomass-degrading enzymes[6] that are central to the lignocellulolytic ability of herbivorous animals[7], yet the role of these seemingly functionally redundant fungi in the herbivore gut microbiome has yet to be investigated.

We used parallel enrichment experiments to study biomass-degrading consortia from goat faeces. Faecal communities were enriched with four types of biomass (alfalfa, bagasse, reed canary grass and xylan) and two antibiotic treatments (chloramphenicol, penicillin–streptomycin) to identify cross-domain partnerships. Enrichment cultures converged to a minimal set of microorganisms that were stable after more than ten culture generations,

and unchanged after cryopreservation. Reconstruction of MAGs enabled microbial community analysis at the species level with metabolic annotation that was verified through metabolomic measurements of each microbial community.

## Results

**Goat faecal metagenome contains uncultured bacterial and archaeal taxa.** Over 1.5 Tbp ($10^{12}$ bp) of metagenome sequencing (Supplementary Data 1) enabled the recovery of 2,452 high-quality bacterial and archaeal MAGs from goat faeces, all of which are >80% complete with <10% contamination evaluated by CheckM[8]. Of these, 719 are unique at the species level based on the recently proposed criteria of species definition[9,10] (Fig. 1 and Supplementary Table 1). This collection of MAGs is similar in scale and quality (91.8% mean completeness and 1.4% mean contamination, Supplementary Table 2) to other anaerobic microbiomes previously published[2–4,11–14]. We performed a comparative analysis of the genome-wide average nucleotide identity for open reading frames and quantified the increase in phylogenetic diversity contributed by MAGs in the present study. Compared with 8,178 genomes from 3 of the largest ruminant gut metagenomic datasets[2–4], the Genomic Encyclopedia of Bacteria and Archaea (GEBA) collection[15], a recent human gut bacteria collection[11] and 221 additional reference genomes from the National Center for Biotechnology Information's (NCBI's) RefSeq[16] (Supplementary Table 2), 677 of the 719 MAGs (94%) in this dataset were previously unidentified at the species

[1]Department of Chemical Engineering, University of California, Santa Barbara, CA, USA. [2]Marine Science Institute, University of California, Santa Barbara, CA, USA. [3]Joint BioEnergy Institute, Lawrence Berkeley National Laboratory, Berkeley, CA, USA. [4]Department of Energy Joint Genome Institute, Lawrence Berkeley National Laboratory, Berkeley, CA, USA. [5]Department of Animal Production, Welfare and Veterinary Sciences, Harper Adams University, Newport, UK. [6]Department of Earth Science, University of California, Santa Barbara, CA, USA. ✉e-mail: momalley@ucsb.edu

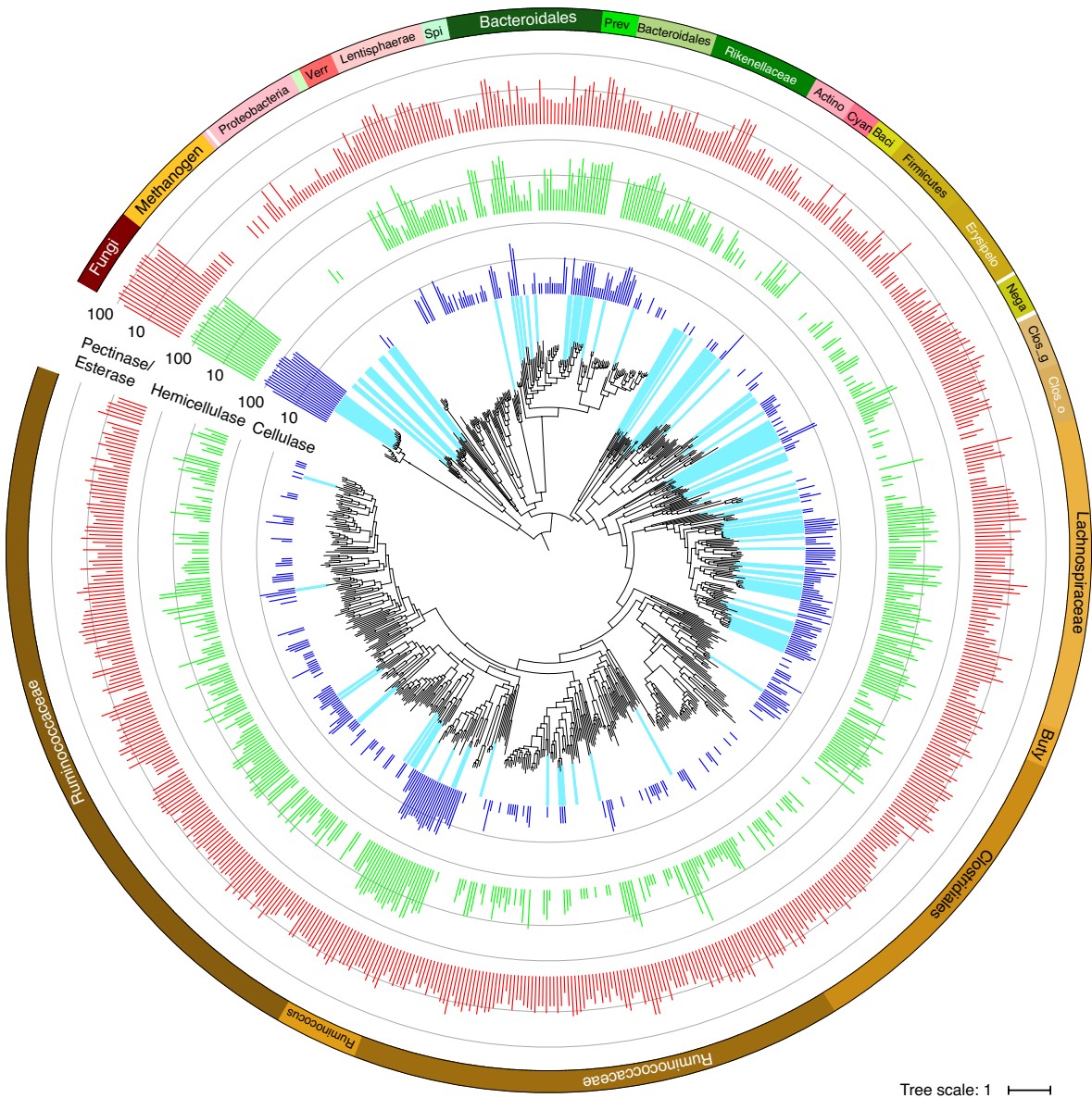

**Fig. 1 | Phylogenomic tree of bacteria, archaea and fungi in the goat faecal metagenome.** Prokaryotic MAGs are divided into 51 groups (coloured in the outer ring) by phylogeny. Phylogeny was determined from a tree, including 8,178 reference genomes from the Hungate Collection, cow rumen, the GEBA and human gut collections, and 221 additional reference genomes from the NCBI collection (Extended Data Fig. 1 and Supplementary Data 2). Coloured bars plotted on a logarithmic scale represent the number of CAZymes (including cellulase, hemicellulase and pectinase/esterase) found in each MAG/eukMAG. The cyan bars radiating from the tip of the tree leaves indicate MAGs that were present in at least one enrichment culture at the end of the experiment (G10). Cyan, Cyanobacteria; Actino, Actinobacteria; Verr, Verrucomicrobia; Spi, Spirochaetes; Prev, *Prevotella* spp.; Baci, Bacilli; Erysipelo, Erysipelotrichaceae; Nega, Negativicutes; Clos_g, *Clostridium* spp.; Clos_o, Clostridiales; Buty, *Butyrivibrio* spp.

level (Extended Data Fig. 1 and Supplementary Data 2). The MAG collection contributed by the present study underscores the vast untapped metabolic potential of gut microbes that perform foregut and hindgut fermentations in herbivores.

Although a number of recent studies have used metagenomics to assess the metabolic potential and interactions within the herbivore rumen[3,12,13], less attention has been paid to the hindgut of herbivores. Microbes located in the hindgut often originate from and were active in the rumen, yet they encounter recalcitrant plant material that is not completely processed in the foregut. In addition, gut microbes cultured directly from the rumen (via fistulated animals) have proved to be extremely difficult to stabilize in culture[2], possibly due to strict nutritional or physiological requirements that are

difficult to mimic outside the rumen. Therefore, it was hypothesized that gut microbial enrichment cultures from faeces would be more robust and resilient in culture, because these microbes experience a broad range of biological, chemical and physical conditions. Three-quarters (531) of the assembled MAGs were Firmicutes and more than half of them belong to the family Ruminococcaceae, most of which were previously unidentified and not enriched in culture except for the genus *Ruminococcus* (Fig. 1, Extended Data Fig. 2 and Supplementary Data 3). The second most abundant phylum (12%) among the MAGs was Bacteroidetes (85), of which the most abundant group belongs to the family Rikenellaceae (Supplementary Table 1 and Supplementary Data 4). In addition, 25 archaeal MAGs were recovered, of which *Methanobrevibacter* sp.

was the most abundant, and 3 *Methanosphaera stadtmanae* MAGs and 7 *Thermoplasmata* MAGs with the potential to generate CH$_4$ using methanol were also recovered. The rest of the prokaryotic MAGs were from the phyla Proteobacteria (23), Lentisphaerae (21), Actinobacteria (10), Verrucomicrobia (8), Cyanobacteria (7), Spirochaetes (6), Planctomycetes (2) and Elusimicrobia (1).

**Fungal MAGs reassembled from goat faecal metagenome.** Eighteen MAGs >40 Mbp in size were recovered with the same assembly and binning pipeline used to reconstruct prokaryotic MAGs. They were termed 'eukMAGs' because >80% of the genes in the eukMAGs were classified as 'Eukaryota' by BLAST+[17] (Supplementary Data 5). It is particularly challenging to recover eukaryotic MAGs, especially fungal MAGs, because their genomes are typically >10 Mbp and they are often characterized by long and frequent repeat regions of low GC content. All eukMAGs in the present study were classified to the fungal subphylum Neocallimastigomycota, commonly known as the anaerobic gut fungi. Phylogenetic analysis based on single-copy orthologues revealed that 12 of the eukMAGs belong to the genus *Neocallimastix* (Extended Data Fig. 3) and are probably the same species as the sequenced strain *Neocallimastix californiae* G1 (Extended Data Fig. 4), which was previously isolated from the faeces of a goat[6,18]. One eukMAG from the genus *Caecomyces* and five eukMAGs from the genus *Piromyces* were reconstructed from only the initial generation of enrichment cultures (Supplementary Data 5). The completeness of the eukMAGs was estimated by benchmarking the number of universal single-copy orthologues to those common to the fungal kingdom using the tool Benchmarking Universal Single-Copy Orthologs (BUSCO)[19,20]. On average, our eukMAGs are 73% complete when compared with the corresponding reference genome with 14% duplicated BUSCOs (Supplementary Data 5). Most of the metagenomic reads from the enrichment cultures can be accounted for by mapping to the MAGs and eukMAGs (Extended Data Fig. 5). To date, no previous MAG datasets have identified anaerobic fungi, because they are typically low-abundance members of the digestive tract microbiome[4], yet they have been recently found to contain a wealth of biomass-degrading enzymes[6] and multi-enzyme cellulosomes[18]. This collection of bacterial and archaeal MAGs and eukMAGs from the goat faecal microbiome serves as a rich resource for metagenomic studies of gut microbiomes, covering microbial taxa not yet included in published collections from the rumen microbiome or otherwise[2,3].

**Abundance of biomass-degrading enzymes in the goat gut microbiome.** The annotated prokaryotic MAGs and eukMAGs were analysed for content and diversity of carbohydrate-active enzymes (CAZymes) that fall into the functional categories of cellulase, hemicellulase and pectinase/esterase according to the CAZy database[21] (Fig. 1 and Supplementary Table 3). It is important to note that glycoside hydrolase (GH) families GH5, GH8, GH44 and GH51 are versatile in function and can hydrolyse both cellulose and hemicellulose depending on the specific subfamily[21]. Anaerobic fungi from the genus *Neocallimastix* represented by eukMAGs contained up to >100 of each type of CAZyme per genome, and were enriched only in antibiotic-treated (penicillin–streptomycin (PS) or chloramphenicol (CM)) and cultured consortia (Supplementary Data 5). Among prokaryotic MAGs, the taxa containing the largest number of CAZymes included anaerobic bacteria from the genera *Ruminococcus*, *Butyrivibrio* and *Prevotella*, and from the families Paenibacillaceae, Bacteroidaceae and Ruminococcaceae (Supplementary Fig. 1, Supplementary Table 4 and Supplementary Data 4). These taxa were enriched in antibiotic-free consortia grown on lignocellulose, and generally include more than five cellulases and more than ten hemicellulases and pectinases/esterases per MAG (Supplementary Data 4).

The number of lignocellulose-active genes in an average *Neocallimastix* eukMAG is about an order of magnitude higher than the average MAG from the genera *Prevotella*, *Butyrivibrio* and *Ruminococcus*, indicating the vast hydrolytic potential of Neocallimastigomycota (Extended Data Fig. 6). Major bacterial cellulases include GH5 and GH9, as well as one GH44 unique to bacterial MAGs (Supplementary Fig. 2a and Supplementary Data 6). Major cellulases sourced from fungi include GH5, GH6, GH9, GH45 and GH48, and, of these, GH6 and GH45 were found only in eukMAGs. CAZymes GH48 and GH6 are well-known abundant proteins in fungal cellulosomes[18]. Hemicellulases, common to both MAGs and eukMAGs, include GH5, GH10, GH11, GH26 and GH43 (Supplementary Fig. 2b). Less commonly found CAZymes GH62 and GH98 were restricted to bacterial MAGs[22,23]. This serves as evidence of functional complementarity of CAZymes contributed by anaerobic fungi and bacteria in the herbivore digestive tract. Notably, *Neocallimastix* eukMAGs and MAGs from the anaerobic bacteria *Ruminococcus* and *Clostridium* spp. also contained dockerin-associated CAZymes, indicating the potential to produce cellulosomes (Supplementary Data 6). Cellulosomes in anaerobic environments are multi-enzyme complexes deployed by all known anaerobic fungi and select anaerobic bacteria, which assist in synergistic breakdown of lignocellulose through enzyme tethering and rearrangement on a flexible protein scaffold[24,25].

**Metabarcoding reveals rapid community stabilization during enrichment.** Cultivation of diverse, anaerobic gut consortia from large herbivores has proved to be exceptionally difficult, and has precluded a complete picture of metabolic potential and exchange within ruminants. In the present study, a source digestive tract microbiome (goat faeces) was challenged with four different carbon substrates and two chemical treatments aimed at selective enrichment of population members that were capable of retaining viability in laboratory culture. Thirty-six consortia were enriched that were each subcultured ten times with three replicate cultivations (Fig. 2 and Extended Data Figs. 7 and 8, 'Consecutive Batch Culture'). Amplicon sequence variant (ASV)-based marker-gene analysis tracked the community composition of both prokaryotes and eukaryotes (Supplementary Data 7–9), which almost all stabilized by the third generation (G3; 15 d from the initial inoculation; Supplementary Figs. 3 and 4). A small percentage of the source faecal microbiota (0–2.2%) was enriched in these cultures (Supplementary Table 5).

Stabilization times for these enriched microbial communities were remarkably short compared with those observed in anaerobic digestors, where longer-duration selection methods are traditionally employed to sculpt community membership[26,27]. Repressed bacterial growth in consortia treated with PS allowed anaerobic fungi and methanogenic archaea to become dominant members of the community. Archaea specifically from the genus *Methanobrevibacter* were enriched by up to 177-fold from their abundance in the source inoculum (Supplementary Table 6). Community composition based on the 16S ribosomal RNA V4 region indicates that Firmicutes and Bacteroidetes were similar in relative abundance and together accounted for ~90% of the source faecal microbiome (Fig. 2b). Fungal members of the source microbiome identified by the internal transcribed spacer region 2 (ITS2) consist of mainly anaerobic fungi from the genera *Piromyces* (41%), *Caecomyces* (26%) and *Neocallimastix* (12%) (Fig. 2b and Supplementary Table 7). The prokaryotic community in the goat faecal microbiome from the present study exhibits similarity to that in the cow faecal microbiome[28,29], both primarily consisting of Bacteroidetes and Firmicutes that include *Prevotella* and *Bacteroides* spp. and many unclassified members of Bacteroidales, Ruminococcaceae and Lachnospiraceae. However, in enrichment cultures, substrate type played an important role in shaping the community compositions (Extended Data Fig. 9). For example, by G3, *Streptococcus*, *Butyrivibrio* and *Pseudobutyrivibrio* spp., and other taxa from

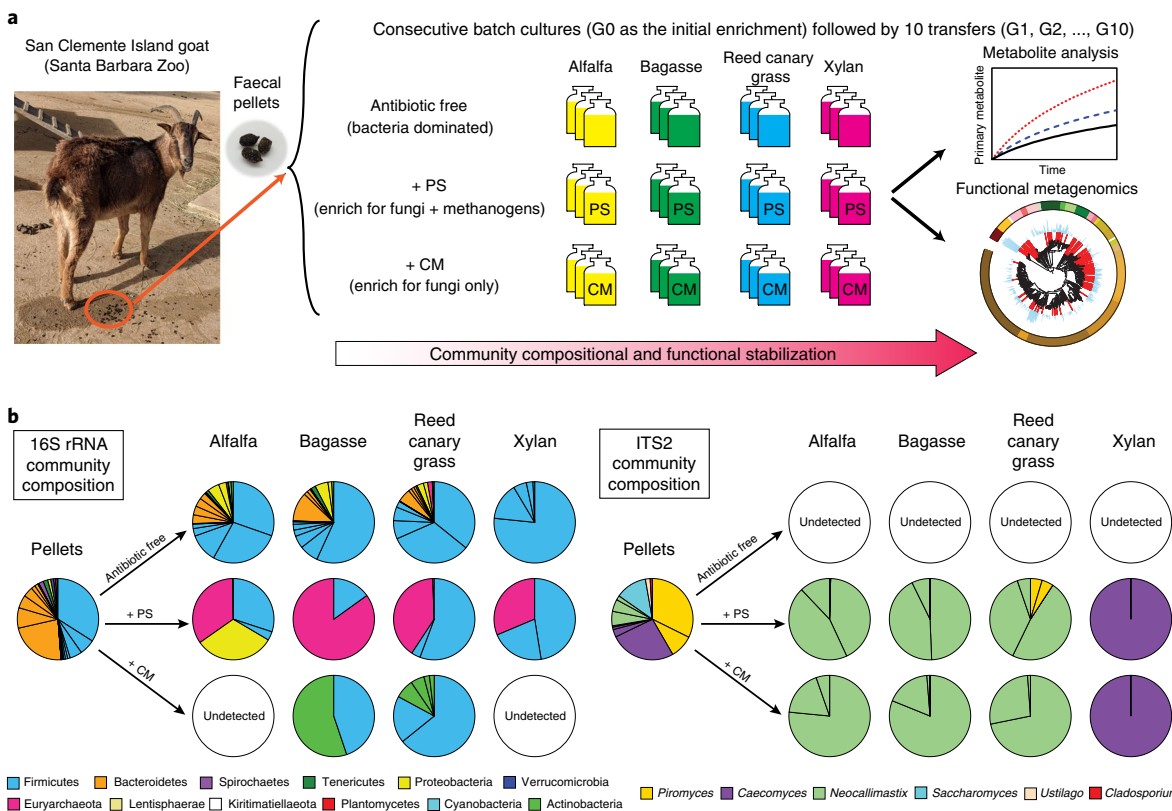

**Fig. 2 | Enrichment strategies. a,b,** Enrichment strategies to characterize cross-domain anaerobic lignocellulolytic consortia (**a**) and microbial composition of enriched communities (**b**). Freshly produced faecal pellets from a San Clemente Island goat served as the source microbiome for 396 parallel microbial enrichment experiments. Enrichment cultures were initiated by challenging the faecal consortia with four types of substrates and two types of antibiotics to bias survival of different microbial communities. Triplicate cultures were inoculated for each condition. Penicillin-streptomycin (PS) was used to inhibit bacterial growth and chloramphenicol (CM) was used to inhibit both bacterial and archaeal growth. Membership within the parallel enrichments was tracked via metabarcoding and whole metagenome assemblies (for G0, G5 and G10), and metabolomic analyses of headspace and liquid cultures were completed at each generation. Prokaryotic and fungal community composition in the source microbiota (faecal pellets) and the consortia at the end of the enrichment experiment (G10) was assessed by metabarcoding of the 16S rRNA V4 region and the ITS2. In the 16S rRNA community composition pie charts, each colour represents a phylum and each slice within the same phylum represents a different family. In the ITS2 community composition pie charts, each colour represents a genus and each slice within the same genus represents an ASV cluster grouped at 97% identity threshold.

the families Ruminococcaceae and Lachnospiraceae dominated antibiotic-free communities grown on the three lignocellulosic substrates (Supplementary Table 6). In contrast, *Selenomonas* and *Enteroccocus* spp. accounted for >80% of the consortia grown on xylan without antibiotics.

Antibiotic treatments selected for a very specific set of bacteria, but their relative abundance in the source microbiome was at a level that was not detectable (Supplementary Table 6e,f). The relatively high abundance of Erysipelotrichaceae and Ruminococcaceae in PS consortia indicate that these bacterial lineages have developed resistance to PS. However, neither of these families was present in CM-treated consortia in which a number of CM-resistant bacteria from the classes Actinobacteria and Bacilli were present (Supplementary Table 6f). Analysis based on ITS2 revealed that by G3 *Neocallimastix* sp. dominated antibiotic-treated (PS or CM) consortia grown on lignocellulosic substrates whereas *Caecomyces* sp. dominated antibiotic-treated (PS or CM) consortia grown on xylan (Supplementary Fig. 3f,g). The differential enrichment of fungi known to possess contrasting rhizoid morphologies could be a function of substrate and might indicate functional specialization among the anaerobic fungi. For example, *Neocallimastix* sp. grows an extensive rhizoid network that enables these fungi to penetrate complex lignocellulosic substrates, whereas the absence of rhizoids in *Caecomyces* sp. may drive its preference for soluble

substrates[30]. These distinct microbial communities, selected by different substrate types and antibiotic treatments, demonstrate the high efficiency of our approach to enriching low-abundance members of the source microbiome that specialize in biomass degradation.

**Mapping the metabolic potential for biomass breakdown.** The reconstruction and annotation of 719 MAGs and 18 eukMAGs enabled estimation of metabolic potential of the microbial community members at the species level. This helped decipher the functional compartmentalization and redundancy among consortia members during lignocellulose breakdown and fermentation. Each MAG's capacity to utilize or produce a chemical was determined by including the corresponding metabolic pathways that are at least 75% complete (Fig. 3). Quantification of major metabolic products by metabolomics provided validation of reconstructed metabolism and benchmarked the performance of each enriched consortium. MAG-based analysis indicated that consortia membership is heavily shaped by the substrate used during enrichment. For example, in the antibiotic-free consortium grown on the most lignin-rich substrate, bagasse (Supplementary Table 8), two bacteria from the family Lachnospiraceae (Lachnospiraceae G11 and *Pseudobutyrivibrio* sp. AR14) were highly enriched, whereas they were absent in an antibiotic-free consortium grown on alfalfa (Supplementary Data 4).

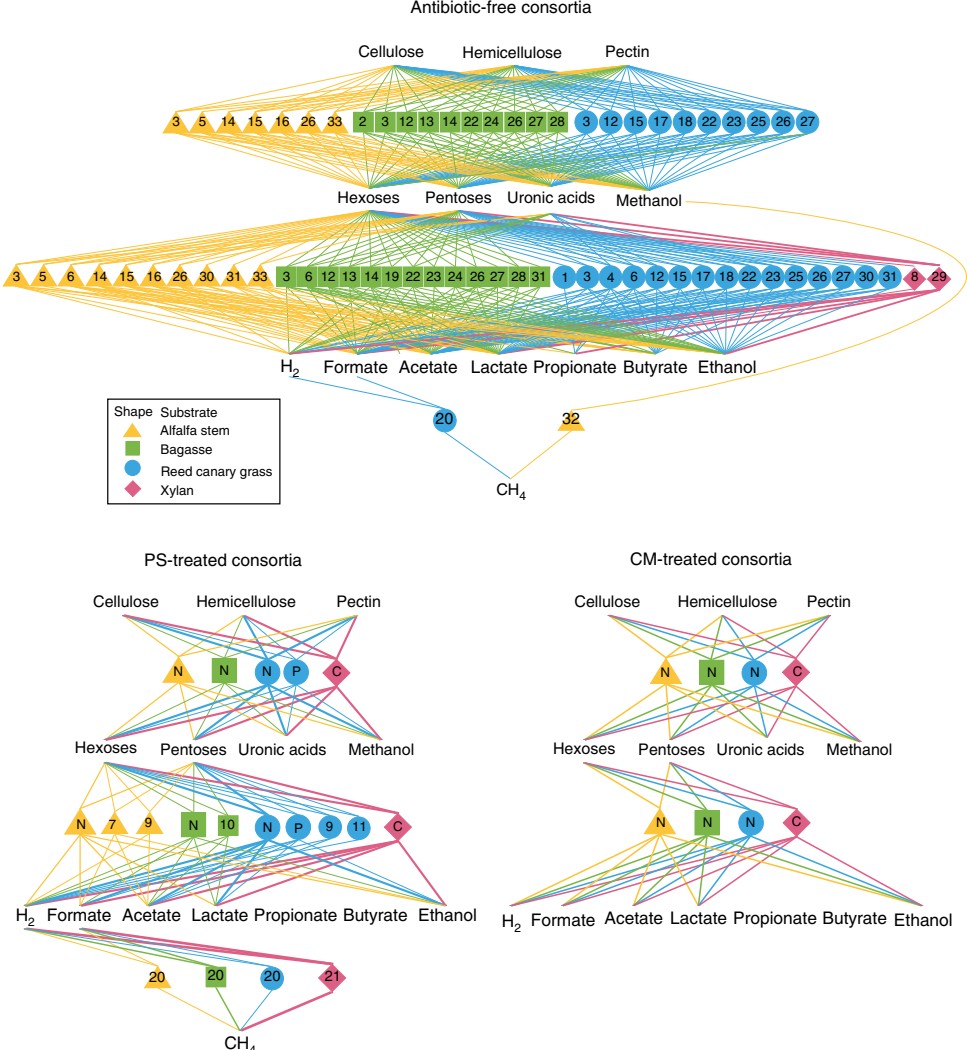

**Fig. 3 | Cross-feeding between microorganisms in enriched anaerobic consortia.** Each shape contains a number or letter representing a MAG contributing to >1% of the total community in each indicated consortium at G10. Each shape corresponds to a type of carbon substrate. Triangles represent alfalfa stem; squares, bagasse; circles, reed canary grass; diamonds, xylan. The thickness of the connecting lines is scaled with the relative microbial abundance of the connected MAG in the corresponding consortium. A line is connected between a MAG and a metabolite if the pathway responsible for the utilization or production of the metabolite is at least 75% complete in the reconstructed MAG. At least two cellulases, hemicellulases or pectinase/esterases are required for a connecting line between a MAG and cellulose, hemicellulose or pectin, respectively. The MAGs are numbered or lettered as: 1, *Anaerovibrio* sp.; 2, Bacteroidales; 3, *Butyrivibrio* sp.; 4, *Clostridium cochlearium*; 5, Coriobacteriaceae; 6 and 7, *Escherichia coli*; 8, *Enterococcus faecium*; 9–11, Erysipelotrichaceae; 12, Lachnospiraceae G11; 13, Lachnospiraceae JC7; 14, *Lachnoclostridium clostridioforme*; 15–19, Lachnospiraceae; 20, *Methanobrevibacter thaueri*; 21, *Methanobrevibacter millerae*; 22 and 23, *Prevotella ruminicola*; 24, *Pseudobutyrivibrio* sp. AR14; 25, *Pseudobutyrivibrio ruminis*; 26, *Ruminococcus albus*; 27, *Ruminococcus flaveflaciens*; 28, Ruminococcaceae; 29, *Selenomonas ruminantium*; 30, *Streptococcus equinus*; 31, *Streptococcus gallolyticus*; 32, Thermoplasmata; 33, *Treponema* sp.; N, *Neocallimastix* sp.; P, *Piromyces* sp.; C, *Caecomyces* sp. The completeness of each metabolic pathway in these MAGs is listed in Supplementary Data 4.

Conversely, the relative abundance of different MAGs from the same family (Lac1) in antibiotic-free consortia grown on alfalfa and reed canary grass was 40–65 times higher than that in the antibiotic-free consortium grown on bagasse (Supplementary Data 4). Methanol-utilizing archaea Thermoplasmata and *Methanosphaera stadtmanae* were enriched only in antibiotic-free consortia grown on alfalfa, which has the highest pectin content of all the four substrates. A primary degradation product of pectin is methanol and this probably explains the enrichment of these taxa on alfalfa. Overall, these data show that substrate type selects for a suite of microbes equipped with the metabolism to depolymer-ize the corresponding carbon substrate, and utilize the solubilized products for fermentation and methanogenesis.

**Consortia dominated by fungi produce more methane.** Bacterially dominated consortia were characterized by potential biomass degradation strategies that differed from those in fungally domi-nated consortia in terms of the fate of fermented monosaccharides (Fig. 3). Some of the rare MAGs (<1% in relative abundance) har-bour metabolic potentials that are unique compared with the more abundant members in the consortium (Supplementary Fig. 5). For example, 'Rum17' (Ruminococcaceae) accounted for a mere 0.5% of the bacterial community enriched on xylan (XG10R1), but it was the only MAG possessing an almost-complete butyrate production pathway in this consortium (Supplementary Fig. 5d and Supplementary Data 4), explaining the source of measured butyrate accumulation in this sample (Fig. 4). In antibiotic-free

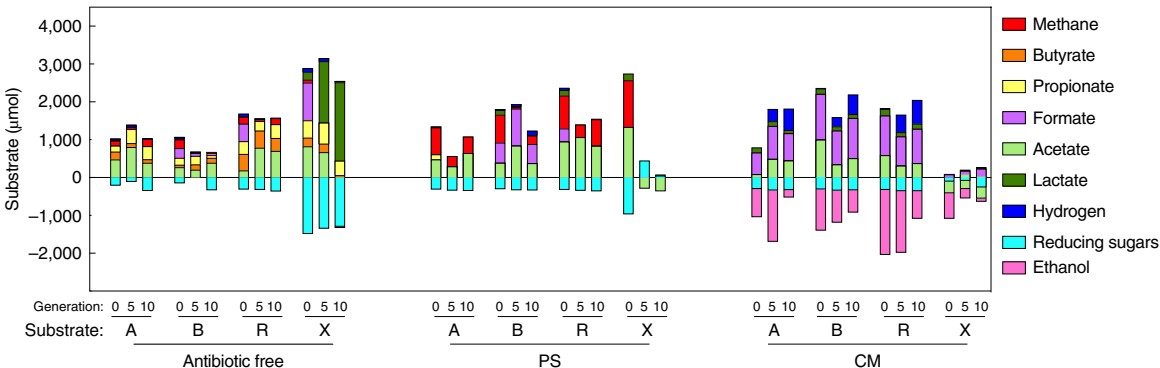

**Fig. 4 | Net change in primary metabolic products during parallel enrichments.** Measurements are grouped by antibiotic treatment and the type of carbon substrate used to drive enrichment. The three bars under each subgroup represent measurements made at G0 (0), G5 (5) and G10 (10). Substrates include alfalfa stems (A), bagasse (B), reed canary grass (R) and xylan (X). Each bar represents the average of three biological replicates.

consortia, there was a high degree of functional redundancy among cellulolytic and fermentative bacteria from different phyla, whereas anaerobic fungi dominated antibiotic-treated consortia membership. Methanogenic archaea became one of the most abundant prokaryotic members in PS consortia (dominated by fungi, archaea and some resistant bacteria), wherein carbon was not diverted by bacteria to produce propionate and butyrate, and as a result PS consortia produced the highest amount of $CH_4$ (Fig. 4, Supplementary Fig. 6 and Extended Data Fig. 10a).

When grown on alfalfa, PS consortia produced nearly twice as much $CH_4$ as the antibiotic-free consortia (Fig. 5). When grown on alfalfa or reed canary grass, very little $H_2$ accumulation was observed in the PS-selected consortia, whereas a small amount of $H_2$ build-up occurred in antibiotic-free consortia, (Extended Data Fig. 10b), suggesting a more efficient metabolic product exchange in PS consortia compared with antibiotic-free consortia. As expected, CM-treated consortia (dominated by fungi) did not produce $CH_4$ but did produce $H_2$ due to the presence of anaerobic fungi and the absence of methanogens[31] (Fig. 5). Methanogens use $H_2$ and produce $CH_4$, and this accounts for the different metabolic product profiles between PS consortia and CM consortia. Seven Firmicutes MAGs were recovered in PS consortia, and the family Erysipelotrichaceae was the most abundant among them (Supplementary Data 4 and Supplementary Fig. 5e–h). These PS-resistant Firmicutes can utilize hexose sugars while producing formate, acetate and lactate. Hence, they might contribute to preventing catabolite repression of anaerobic fungi by maintaining consistent, but low, levels of simple sugars[6].

**Antibiotic-free consortia are functionally redundant.** In antibiotic-free consortia, most enriched bacteria probably occupy a mixed trophic level with the dual capability of degrading plant cell walls and fermenting simple sugars. The result is that these consortia demonstrate a high degree of functional redundancy, with 44 bacteria capable of hydrolysis and 78 capable of fermentation (Supplementary Table 9). Abundant cellulolytic and hemicellulolytic bacteria from the genera *Ruminococcus*, *Prevotella*, *Butyrivibrio* and *Pseudobutyrivibrio*, and the family Lachnospiraceae, were enriched. Most of these bacteria can produce formate, acetate and lactate (Fig. 3), which are typical of gut microbial communities[32]. Although less than half of the microbial community can produce butyrate and <20% of the community can produce propionate, the potential for butyrate and propionate production is redundantly spread across four bacterial phyla including Proteobacteria, Actinobacteria, Firmicutes and Bacteroidetes (Supplementary Data 4). The antibiotic-free consortium grown on xylan was dominated by *Selenomonas ruminantium* (72%), and another bacterium, *Enterococcus faecium*, was also

enriched. There were large amounts of reducing sugars (simple sugars measured by 3,5-dinitrosalicylic acid (DNS) assay) available in xylan culture media relative to complex fibres, which probably contributed to the very low microbial diversity observed. The moderate level of $CH_4$ production and the high production of propionate and butyrate in antibiotic-free consortia clearly indicate that carbon flow was diverted towards SCFAs and away from $CH_4$.

**Fungus–methanogen partnerships enable cellulose degradation.** Factors that modulate lignocellulose degradation and $CH_4$ production by ruminant microbiomes are poorly understood, further motivating characterization of herbivore microbiota. Given the high levels of $CH_4$ production observed in fungally dominated, PS-treated consortia compared with bacterially dominated, antibiotic-free consortia (Figs. 4 and 5, and Extended Data Fig. 10), the performance of these enriched lignocellulolytic consortia was compared after long-term passaging, cryopreservation and revival. When grown on cellulose paper (Whatman filter paper), the PS consortium enriched on alfalfa degraded almost twice as much substrate as the antibiotic-free consortium enriched on reed canary grass after 7 d of growth (Fig. 6a). Excess reducing sugars were released from the PS consortium, but not from the antibiotic-free consortium, when grown on cellulose paper (Fig. 6b). By contrast, the PS consortium and the antibiotic-free consortium degraded comparable amounts of reed canary grass after 7 d of growth (Fig. 6a). This suggests that, despite the advantage of fungally dominated consortia compared with bacterially dominated consortia in degrading cellulose, these advantages did not lead to increased depolymerization of crude, lignin-rich substrates.

The enzymatic strategies for hydrolysing lignocellulose deployed by fungally dominated PS consortia and bacterially dominated, antibiotic-free consortia were compared by enumerating the number of cellulases, hemicellulases, pectinases and esterases with and without dockerin domain associations in each consortium. We found that the number of cellulosomal CAZymes in PS consortia grown on lignocellulosic substrates was more than two orders of magnitude higher than that found in antibiotic-free consortia, suggesting that cellulosomes are a key biomass degradation strategy employed by fungally dominated consortia (Fig. 6c). By comparison, the total numbers of cellulases, hemicellulases, pectinases and esterases in antibiotic-free consortia grown on various lignocellulosic substrates were higher than those in PS consortia (Fig. 6c). The larger number of CAZymes observed in antibiotic-free consortia compared with PS consortia is attributed to the large numbers of GH5, GH8, GH9, GH16, GH26, GH30, GH43, GH28, carbohydrate esterase CE4 and CE8 (Supplementary Fig. 7 and Supplementary Data 10). Nevertheless, GH6 and GH45 (cellulases) were enriched

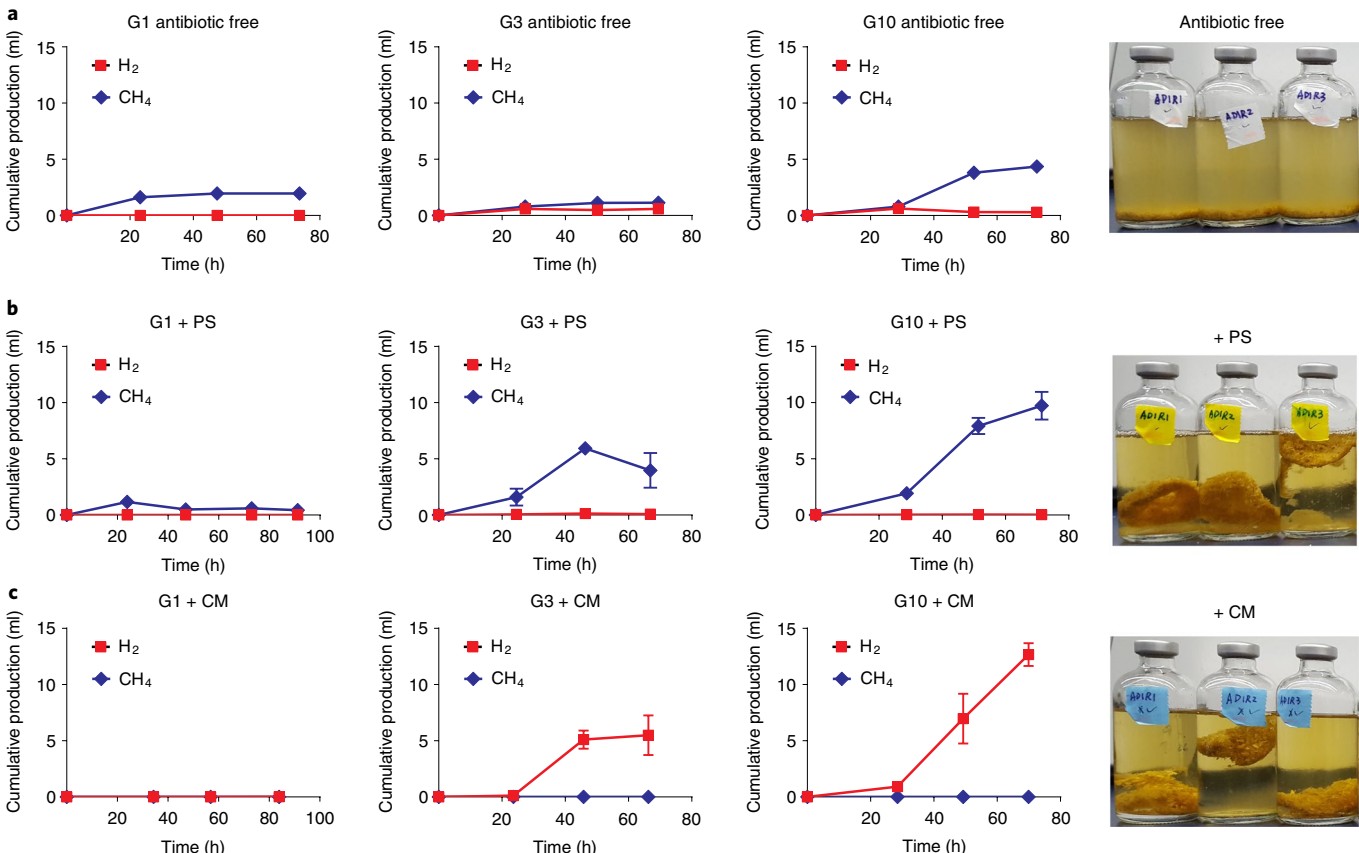

**Fig. 5 | Cumulative production of H₂ and CH₄ by enrichment cultures grown on alfalfa stems at G1, G3 and G10. a–c**, Photographs of the cultures from G1 are shown on the right. **a**, Results for antibiotic-free cultures. The turbid liquid media are characteristic of bacterial growth. **b**, Results for PS-treated cultures. **c**, Results for CM-treated cultures. The symbols represent the mean (n = 3) and the error bars the s.d. The clear liquid media in antibiotic-treated cultures indicate low prokaryotic abundance. The clumped alfalfa stems floating in the liquid media are characteristic of anaerobic fungal growth, because fungi associate directly with the substrate and typically float due to entrapped fermentation gases.

only in PS consortia, and there were larger numbers of GH48, GH11 and polysaccharide lyase 3 (PL3) in PS consortia than in antibiotic-free consortia (Supplementary Table 10). In addition, among the MAGs enriched in antibiotic-free consortia, there were on average around eight CAZyme gene clusters (CGCs) defined by the presence of at least one CAZyme gene, one transporter gene and one transcription factor gene[33] (Supplementary Table 11 and Supplementary Data 11). Among the Bacteroidetes MAGs enriched in antibiotic-free consortia with plant substrates, there were on average 17–22.5 polysaccharide utilization loci (PULs) defined by the tandem presence of *SusC* and *SusD* genes[34,35] (Supplementary Table 11 and Supplementary Data 12).

Additional experiments were performed to determine the long-term stability of CH₄ production by the best-performing lignocellulolytic consortium (PS consortium grown on alfalfa). This consortium was maintained in the laboratory for >3 years while consistently producing CH₄ (Supplementary Fig. 8). The eukaryotic member of the consortium was an anaerobic fungus from the genus *Neocallimastix*, and the dominant bacterial members were from the family Erysipelotrichaceae (Supplementary Fig. 9). *Ruminococcus* spp. were present at low abundance and *Methanobrevibacter* spp. were responsible for CH₄ generation. Furthermore, both the prokaryotic and the eukaryotic parts of this consortium were stable after cryopreservation at −80 °C for more than 1 year (Supplementary Fig. 9). Importantly, this consortium produced CH₄ after reviving from cryopreservation, indicating its potential long-term utility in lignocellulosic bioprocessing applications.

## Discussion

We report ~400 parallel microbial enrichment experiments that provide valuable insights into metabolic connectivity between gut microbes, and could inform design of microbial consortia for lignocellulose deconstruction and bioproduct generation. Differential enrichment with distinct antibiotics and substrates enabled us to disentangle the goat gut microbiome into specific taxa and permitted analysis of complex microbial consortia that were dominated by either bacteria or fungi. Although metagenomic and metabolomic analyses showed that both bacterially dominated and fungally dominated consortia were capable of producing CH₄, we found that consortia dominated by fungi produced more than twice as much CH₄ and degraded twice as much cellulose relative to bacterially dominated consortia.

Comparative analysis of metagenomes revealed that the metabolic potential of the microbial community plays a fundamental role in determining the outcome of lignocellulosic biomass deconstruction and fermentation. Bacteria dominated antibiotic-free consortia, and less common members of the rumen microbiome were cultivated only with the assistance of PS or CM. A few dominant members in antibiotic-free consortia, from the family Lachnospiracea and the order Bacteroidales, fermented a considerable portion of the hydrolysed carbon into propionate and butyrate, in addition to formate and hydrogen. Shunting carbon towards SCFAs reduces substrate availability for hydrogenotrophic methanogens, and this shift is consistent with the metabolic dynamics previously reported in rumen microbiomes[36–39]. In contrast, there were much higher

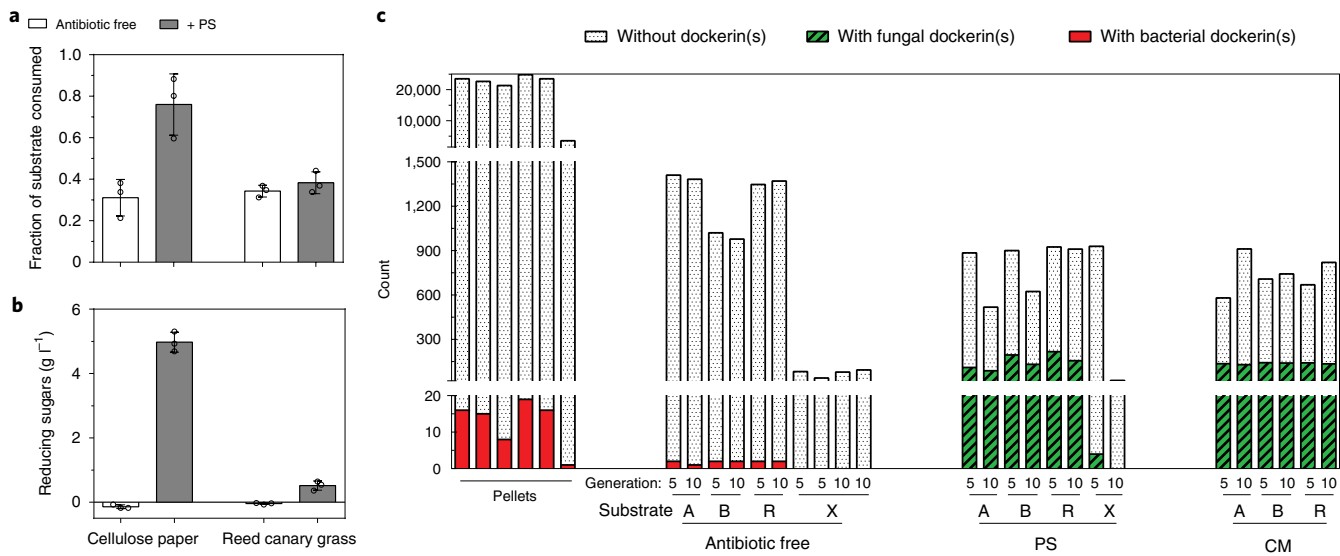

**Fig. 6 | Fungal CAZymes and cellulosomes drive lignocellulosic efficiency of consortia dominated by anaerobic fungi and methanogenic archaea.**
**a,b**, The fraction of substrate consumed (**a**) and the release of reducing sugars (**b**) by enrichment cultures grown on cellulose paper and reed canary grass. Each bar in **a** and **b** represents the average of three replicates and the error bars represent the s.d. ($n = 3$); the 'antibiotic-free' consortia were initially selected on reed canary grass whereas the 'PS' consortia were initially selected on alfalfa—both were passaged long term, cryopreserved and revived before this experiment. **c**, Number of CAZymes classified as cellulase, hemicellulase, pectinase or esterase (Supplementary Table 3) in all enrichment cultures from G5 and G10, distinguished by the presence or absence of bacterial or fungal dockerin domains fused to CAZymes. Note that the *y* axis has been broken into three different scales. Red boxes represent bacterial CAZymes associated with dockerin(s), green boxes with black slanted stripes represent fungal CAZymes associated with dockerin(s) and black dotted patterns represent CAZymes unassociated with dockerin(s). Capitalized letters represent the substrate type: alfalfa stems (A), bagasse (B), reed canary grass (R) and xylan (X). The fraction of cellulose paper consumed by 'PS' consortia was significantly higher than that by 'antibiotic-free' consortia (two-tailed Student's *t*-test, $P = 0.017$). Paired Student's *t*-tests showed that the number of CAZymes with fungal dockerin(s) in PS or CM consortia grown on plant substrates ($n = 6$) was significantly higher than the number of CAZymes with bacterial dockerin(s) in antibiotic-free consortia (two-tailed Student's *t*-test, $P = 0.0007$ comparing PS and antibiotic-free consortia and two-tailed Student's *t*-test, $P = 1.5 \times 10^{-8}$ comparing CM and antibiotic-free consortia). Paired Student's *t*-tests also showed that the total number of CAZymes was significantly lower in PS or CM consortia than in antibiotic-free consortia (two-tailed Student's *t*-test, $P = 0.006$ comparing PS and antibiotic-free consortia and two-tailed Student's *t*-test, $P = 0.002$ comparing CM and antibiotic-free consortia).

levels of $CH_4$ production in PS consortia because anaerobic fungi are incapable of butyrate and propionate production. Altogether, these observations suggest that simple cultivation tools can be applied to shift the output of a microbial community to a target product. For example, if the goal of a designed consortium is $CH_4$ production, then the consortium should streamline fermentation products for methanogenesis by excluding bacteria that produce butyrate and propionate.

High-level functional metagenomics also enabled identification of thousands of CAZymes from the goat gut microbiome. Antibiotic-treated consortia degraded a larger amount of cellulose compared with antibiotic-free consortia, despite the higher number of total CAZymes detected in the latter. Although we did not compare relative transcription levels or expressed proteins in the present study, it is worthwhile noting that, relative to antibiotic-free consortia, PS consortia encoded a larger number of dockerin-associated CAZyme genes that are generally considered part of enzyme-tethered systems. Synergy between freely diffusive and enzyme-tethered (cellulosomal) systems of CAZymes can result in increased cellulose deconstruction compared with microbial consortia that rely solely on the action of free CAZymes, and this has been reported previously[25]. Moreover, antibiotic-treated consortia permit the survival of fungi, which feature rhizoid networks capable of crude lignocellulose penetration that expose surfaces of plant polymers for free and cellulosomal CAZymes to act on. These findings highlight the disproportionately large role that rare microbes such as anaerobic fungi play in lignocellulosic biomass degradation,

despite abundances of fungi that are orders of magnitude lower than those of gut bacteria in the rumen[40].

Comparison of the resultant microbial communities, obtained by challenging the source microbiome with four different types of carbon sources, indicates that substrate composition plays a critical role in shaping the community composition. Specifically, in antibiotic-free enrichment cultures, simple reducing sugars were enriched for a small number of specialist fermentative bacteria that dominated the community, whereas complex lignocellulose substrates were enriched for many functionally redundant lignocellulolytic bacteria (Supplementary Table 9). Similarly, antibiotic-treated cultures enriched on crude plant material were probably selected for anaerobic fungi capable of rhizoid development. On the other hand, the dominance of the bulbous fungus *Caecomyces* sp. in antibiotic-treated cultures enriched on xylan suggests that they are better at using reducing sugars alone as a carbon substrate than filamentous anaerobic fungi (*Neocallimastix* and *Piromyces* spp.). With regard to community complexity, the PS consortia enriched in the present study consist of fewer members, whereas the antibiotic-free consortia include an order of magnitude higher number of bacterial taxa that present an engineering challenge to their use in biomass conversion. Limiting the number of consortia members simplifies metabolic model construction, and less complex communities are more amenable to scaling up and testing under industrial conditions. The ease of cryopreservation, as well as the community stability, of this simple consortium further improves the industrial utility of this community.

The in-depth analysis of the goat faecal microbiome, which partially represents the hindgut microbiome, along with our enrichment cultures, serves as a rich resource for the biotechnology community interested in designing processes for lignocellulolytic biomass conversion into $CH_4$-rich products or other targets (that is, SCFA production). Furthermore, our results indicate that it is currently challenging to co-cultivate anaerobic fungi with fast-growing cellulolytic and fermentative bacteria, despite the observed coexistence of these organisms in the rumen. To design consortia with both anaerobic fungi and bacteria, it will be crucial to better understand the physicochemical parameters that enabled anaerobic fungi to persist and even thrive as stable members in the gut of herbivores[41].

Our evaluation of how selective interventions tune the membership and chemical output of gut consortia provides a knowledge base to design minimal systems or synthetic consortia for the bioconversion of lignocellulose into value-added chemicals.

## Methods

**Enrichment of consortia.** Freshly voided faecal pellets were collected from a San Clemente Island goat at Santa Barbara Zoo, and these pellets served as source material for 396 parallel anaerobic enrichment experiments (Fig. 2). In the laboratory, an anaerobic environment was created by flushing and filling an Aldrich AtmosBag (catalogue no. SKU Z555525) three times with $CO_2$, and the initial inoculum containing the source microbiome from the faecal pellets was prepared in this anaerobic glove bag. Eight faecal pellets were transferred from 50-ml centrifuge tubes into a 73-ml sealed serum bottle containing 50 ml of anaerobic culture medium 'MC−'[42] (see Culture media below). The faecal pellets in the culture medium were homogenized into small particles by vortexing for 3 min, and were then used to inoculate 36 serum bottles prepared with 3 antibiotic treatments and 4 carbon substrate conditions.

**Culture media.** All enrichment cultures were based on a previously described medium recipe for MC−[42]. Medium MC− Includes $6 \, g \, l^{-1}$ of sodium bicarbonate (VWR Chemicals, LLC), $0.45 \, g \, l^{-1}$ of potassium phosphate dibasic (Acros Organics ACS reagent), $0.45 \, g \, l^{-1}$ of potassium phosphate monobasic (Sigma-Aldrich, catalogue no. P5655), $0.9 \, g \, l^{-1}$ of ammonium sulfate (Sigma-Aldrich, catalogue no. A4418), $0.9 \, g \, l^{-1}$ of sodium chloride (Macron fine chemicals, ACS reagent), $0.09 \, g \, l^{-1}$ of magnesium sulfate heptahydrate (Sigma-Aldrich, catalogue no. M5921), $0.09 \, g \, l^{-1}$ of calcium chloride dihydrate (Fisher Chemicals, catalogue no. BP510), $1 \, g \, l^{-1}$ of L-cysteine hydrochloride (Sigma-Aldrich, catalogue no. C7477) as reducing reagent, 7% clarified ovine rumen fluid (Bar Diamond, Inc.), $0.25 \, g \, l^{-1}$ of Bacto Yeast Extract (BD REF 212750) and $0.5 \, g \, l^{-1}$ of Bacto Casitone (BD REF 225930), 0.1% (v:v) vitamin supplement (ATCC, catalogue no. MD-VS) and 1% (w:v) of carbon substrate (see below for details about carbon substrates). The vitamin solution includes $2 \, mg \, l^{-1}$ of folic acid, $10 \, mg \, l^{-1}$ of pyridoxine hydrochloride, $5 \, mg \, l^{-1}$ of riboflavin, $2 \, mg \, l^{-1}$ of biotin, $5 \, mg \, l^{-1}$ of thiamine, $5 \, mg \, l^{-1}$ of nicotinic acid, $5 \, mg \, l^{-1}$ of calcium pantothenate, $0.1 \, mg \, l^{-1}$ of vitamin $B_{12}$, $5 \, mg \, l^{-1}$ of $p$-aminobenzoic acid, $5 \, mg \, l^{-1}$ of thioctic acid and $900 \, mg \, l^{-1}$ of monopotassium phosphate. In addition, $40 \, g \, l^{-1}$ of sodium 2-mercaptoethane sulfonate (Sigma-Aldrich, catalogue no. M1511) was added to the vitamin solution. All culture media were prepared anaerobically under a $CO_2$ (Praxair CD 50, 99.5% purity) atmosphere and included $1 \, mg \, l^{-1}$ of resazurin sodium salt (Sigma-Aldrich, catalogue no. R7017) as a colour indicator for anoxia. All enrichment cultures were incubated at 39 °C.

The three antibiotic treatments include PS (Sigma-Aldrich, catalogue nos. P3032 and S9137, $2 \, mg \, ml^{-1}$ final concentration), CM (C1919, $2 \, mg \, ml^{-1}$ final concentration) and a control without antibiotics added. The stock CM solution (100×) was prepared in 40% ethyl alcohol (molecular biology grade, Sigma-Aldrich, catalogue no. E7023), resulting in 0.4% ethyl alcohol in the CM-treated cultures. CM was applied to one group of cultures to bias selection for anaerobic fungi. PS was applied to a second group of cultures to bias selection of anaerobic fungi and methanogenic archaea.

The four carbon substrates are alfalfa stems (A), sugarcane bagasse (B), reed canary grass (R) and xylan (X; TCI America, catalogue no. X0064). Alfalfa stems and reed canary grass were provided by the US Department of Agriculture, Agricultural Research Service, US Dairy Forage Research Center, and they were milled in a Model 4 Wiley Mill (Thomas Scientific) using a 4-mm screen size (courtesy of P. J. Weimer). Sugarcane bagasse was provided by Alma Plantation, LLC and was ground to approximately 5 mm in size using a Mr. Coffee Electric Coffee Bean Grinder (model no. IDS57-RB). Xylan was extracted from powdered corn cores.

**Consortia maintenance and primary metabolite analysis.** In this article, the initial generation of enrichment cultures is referred to as G0, and the subsequent culture generations were referred to by their consecutive batch culture numbers (G1, G2, and so on, to G10). The initial enrichment cultures were each separately subcultured (10% v:v) into fresh medium with an appropriate carbon substrate

every 3 d. Exceptions to the transfer schedule occurred during G0 and G1, which were allowed to grow for 5 and 4 d, respectively, before subculturing. The delayed transfer in these cases enabled maximum development of the community[43].

Three replicates for each culturing condition were included (3 antibiotic treatments and 4 carbon substrates), resulting in a total of 36 enrichment cultures maintained at the same time. For each sample from each batch, a unique identifier with the format SGxRy existed, where S represents the carbon substrate (A for alfalfa stems, B for bagasse, R for reed canary grass and X for xylan), x represents the batch number (0–10) and y represents the replicate number (1, 2 or 3). For samples treated with PS, an additional -PS is appended to the end of the sample identifier, and, for samples treated with CM, an additional -CM is appended.

Culture activity was monitored daily by sampling the headspace of anaerobically sealed bottles to measure pressure accumulated[44], as well as $H_2$ and $CH_4$ concentrations (Extended Data Fig. 8). $H_2$ and $CH_4$ concentrations in the serum bottle headspace were measured on a gas chromatograph (GC)-pulsed, discharge helium ionization detector[45] (Thermo Fisher Scientific TRACE 1300) via direct injection (100 µl) using a Valco Precision sampling syringe (Series A-2). Ultra-high-purity helium (Praxair, part no. HE 5.0UH-55), which was further purified with a heated helium purifier (VICI product no. HP2), was used as the carrier gas with a flow rate set at $5 \, ml \, min^{-1}$. $H_2$ and $CH_4$ are separated on a TracePLOT TG-BOND Molecular Sieve 5A GC column (30 m) at a constant temperature of 30 °C, and eluted at 0.98 and 3.80 min, respectively. Peak areas were integrated using the software Chromeleon Chromatography Data System 7 (Thermo Fisher Scientific). The temperature in the injector module and the detector was set at 150 °C. $H_2$ and $CH_4$ concentration standards (Douglas Fluid & Integration Technology LLC) of 0.1%, 0.5%, 1%, 2%, 5%, 10% and 20% (balanced with helium) were used to determine $H_2$ and $CH_4$ concentrations in sample gases. Production of total gas, $H_2$ and $CH_4$ was calculated using measurements of total headspace pressure, and $H_2$ and $CH_4$ concentrations in Matlab (v.2017b). The sum of total gas/$H_2$/$CH_4$ production from each sampling timepoint was calculated as the cumulative total gas/$H_2$/$CH_4$ production.

Concurrently (Extended Data Fig. 8), 1 ml of the supernatant from the culture (the serum bottle was gently inverted to homogenize the culture medium) was transferred using a syringe to a 1.5-ml microcentrifuge tube, which was immediately stored frozen at −80 °C until analysis on an Agilent 1260 Infinity high-performance liquid chromatograph (HPLC, Agilent) equipped with an auto-sampler unit (1260 ALS). Separation of formate, acetate, lactate, propionate, butyrate and ethanol was achieved using a Bio-Rad Aminex HPX-87H column for organic acids (part no. 1250140) set to 35 °C and a flow rate of $0.5 \, ml \, min^{-1}$, with a mobile phase consisting of 5 mM sulfuric acid. Guard columns included before the analytical column were a 0.22-µm mesh filter (part no. 50671551, Agilent), followed by a polyether ether ketone guard cartridge (part no. ANX993515, Transgenomic). Before loading into the HPLC, all samples were pre-treated by acidification with a 1:10 volume of 50 mM sulfuric acid. After vortexing and incubating at room temperature for 5 min, acidified samples were centrifuged at $21,000g$ for 5 min and the supernatant was syringe filtered (0.22 µm) into an HPLC vial (Eppendorf, catalogue no. FA-45-24-11). In-house concentration standards were prepared at concentrations of 0.01% and 0.1% (w:v) with blank culture medium MC− as the base using sodium formate (ACS Grade, Fisher Chemical, catalogue no. S648500), sodium acetate (ACS Grade, Fisher Chemical, catalogue no. S210500), L-lactic acid sodium (99%, extra pure, Acros Organics, catalogue no. 439220100), propionic acid (TIC America, catalogue no. P0500), $n$-butyric acid (99%, Acros Organics, catalogue no. 108111000) and ethyl alcohol (molecular biology grade, Sigma-Aldrich, catalogue no. E7023). Then, 20 µl of each acidified sample or standard was injected into the HPLC. Peak areas were integrated using the software OpenLab CDS ChemStation (edition C.01.02, Agilent). Reducing sugar concentrations were assayed separately using a low-concentration protocol for the DNS method[46,47]. Standards made with D-(+)-glucose (Sigma-Aldrich, catalogue no. G8270) were prepared at concentrations of 0.1% and 1% (w:v).

Finally, the headspace pressure of the sample was reduced to atmospheric pressure with a syringe and needle connected to the pressure transducer. After each batch of consortia samples was transferred into the next batch, the remaining 45 ml of cultures was harvested by centrifuging at 4,000 r.p.m. for 20 min at 1 °C. The supernatant was discarded and RNA*later* (QIAGEN, catalogue no. 76106) was added to biomass pellets before storage at −80 °C.

**Nucleic acid extraction from microbial enrichment samples.** DNA and RNA from the same sample were extracted following the QIAGEN AllPrep DNA/RNA/miRNA Universal handbook using the protocol for 'cells' with the following modifications: frozen biomass pellet samples were thawed and centrifuged at 4 °C for 10 min at 12,000 r.p.m. in a fixed-angle rotor (Eppendorf, catalogue no. F-34-6-38). Then the supernatant was decanted and discarded using a pipette. The pellets were transferred into 2-ml bead-beating tubes containing 1 ml of 0.5-mm zirconia/silica beads (Biospec, product no. 11079105z) and 700 µl of buffer RLT plus from the QIAGEN AllPrep DNA/RNA/miRNA Universal kit (catalogue no. 80224). All samples were lysed on a Biospec Mini-BeadBeater-16 for 1.5 min, followed by cooling on ice for 2 min before a second round of bead beating for 1.5 min. Then, all the bead tubes containing samples were centrifuged at 13,000 r.p.m. for 3 min (Eppendorf, catalogue no. FA-45-24-11). The supernatant

in each sample was transferred into a 1.5-ml microcentrifuge tube and centrifuged again at 13,000 r.p.m. for 3 min before nucleic acid extraction. The resultant quantity and quality of DNA and RNA were measured using a Qubit (Thermo Fisher Scientific) and TapeStation 2200 (Agilent). DNA samples were sent to the Joint Genome Institute (JGI) for marker-gene sequencing and analysis[48], as well as shotgun metagenomic sequencing as described below. RNA samples from the same generations from which DNA was extracted were not sequenced due to low quality.

**Marker-gene and metagenome sequencing.** High-resolution marker-gene (16S, 18S and ITS) analysis was performed on an Illumina MiSeq sequencer (300 bp × 2) for 6 of the 11 generations (G0, G1, G3, G5, G8 and G10) to track enrichment of the community under different selective pressures. Detailed procedures of library preparation and sequencing methods for each individual sample are presented in Supplementary Data 13. In summary, three sets of marker genes were used in the marker-gene sequencing: V4 region of the 16S rRNA (515F: 5′-GTGCCAGCMGCCGCGGTAA-3′; 805R: 5′-GGACTACHVGGGTWTCTAAT-3′), V4 region of the 18S rRNA (515F-Y: 5′-GTGYCAGCMGCCGCGGTAA-3′; 926R: 5′-CCGYCAATTYMTTTRAGTTT-3′) and the second region of the internal transcribed spacer (ITS9: 5′-GAACGCAGCRAAIIGYGA-3′; ITS4: 5′-TCCTCCGCTTATTGATATGC-3′). DNA, 30 ng was amplified using the 5PRIME HotMasterMix amplification kit (Quantabio, QIAGEN). The prepared libraries were quantified using KAPA Biosystem's next-generation sequencing library quantitative PCR (qPCR) kit and run on a Roche LightCycler 480 real-time PCR instrument. The quantified libraries were then multiplexed with other libraries, and the pool of libraries was sequenced on the Illumina MiSeq sequencer using MiSeq reagent kits v.3, following a 2 × 300 indexed run recipe. Raw sequence files are available at the JGI Genome Portal under JGI project IDs 1140136, 1132607, 1149268 and 1149266.

One of each set of three replicates from G0, G5 and G10 was chosen for shotgun metagenomic sequencing. Sample DNA, 200 ng, was sheared to 300 bp using a Covaris LE220 focused ultrasonicator. The sheared DNA fragments were size selected by double-solid phase reversible immobilization and then selected fragments were end-repaired, A-tailed and ligated with Illumina-compatible adaptors from Integrated DNA Technologies containing a unique molecular index barcode for each sample library. The prepared libraries were quantified using KAPA Biosystem's next-generation sequencing library qPCR kit and run on a Roche LightCycler 480 real-time PCR instrument. The quantified libraries were then multiplexed with other libraries, and the pool of libraries was prepared for sequencing on the Illumina HiSeq sequencing platform utilizing a TruSeq paired-end cluster kit, v.4, and Illumina's cBot instrument to generate a clustered flow cell for sequencing. Sequencing of the flow cell was performed on the Illumina HiSeq2500 sequencer using HiSeq TruSeq SBS sequencing kits, v.4, following a 2 × 150 indexed run recipe. The total number of reads sequenced was >1.5 Tbp ($10^{12}$ bp).

**Metabarcoding analysis.** For 16S-V4 sequencing results, raw forward and reverse reads were merged with the 'fastq_mergepairs' function in USEARCH v.11 (ref. [49]). Merged reads were trimmed to 291 bp and filtered in R package DADA2 (v.1.8.0) with maxEE set to 2. Errors learned from 1,036,304,835 total bases in 3,561,185 reads from 22 samples were used for sample inference with the dada2 algorithm[50]. After removal of chimeric sequences, there were 8,171 16S-V4 ASVs from 233 samples. Taxonomy was assigned to ASVs using SILVA taxonomic training data formatted for DADA2 v.132 release[51].

For 18S-V4 sequencing results, raw forward and reverse reads were merged with the 'fastq_mergepairs' function in USEARCH v.11 (ref. [49]). Merged reads were trimmed to 421 bp and filtered in R package DADA2 (v.1.8.0) with maxEE set to 1. Errors learned from 235,701,060 total bases in 559,860 reads from 2 samples were used for sample inference with the dada2 algorithm[50]. After removal of chimeric sequences, there were 4,310 18S-V4 ASVs from 165 samples. Taxonomy was assigned to ASVs using SILVA taxonomic training data formatted for DADA2 v.132 release[51].

ITS2-sequencing raw reads were trimmed to the ITS2 region and merged using ITSxpress v.1.8.0 (ref. [52]), which implements ITSx[53] and BBMerge[54]. Merged reads with lengths smaller than 40 bp were discarded and then filtered in R package DADA2 (v.1.8.0) with maxEE set to 2. Errors learned from 214,653,240 total bases in 596,259 reads from 2 samples were used for sample inference with the dada2 algorithm[50]. After removal of chimeric sequences, there were 2,093 ITS2 ASVs from 151 samples. Taxonomy was assigned to ASVs using UNITE general FASTA release for Fungi v.02.02.2019 (https://doi.org/10.15156/BIO/786343), and refined with a local blast database constructed with the six publicly available genomes or transcriptomes of anaerobic fungi including *Neocallimastix californiae*, *Caecomyces churrovis*, *Anaeromyces robustus*, *Orpinomyces* sp. C1A, *Piromyces finnis* and *Piromyces* sp. E2.

As an ASV-based analysis pipeline is highly sensitive to any nucleotide variations in amplicons, it provides a view of the microbial communities with the finest resolution (strain level or lower). To provide a perspective of the community composition of the goat faecal microbiome and associated enrichment cultures at a higher taxonomic level, we clustered all the ASVs by a threshold of 97% similarity, using the UPARSE-OTU algorithm implemented in USEARCH[55]. Clustering 8,171

16S-V4 ASVs at 97% similarity level resulted in 877 operational taxonomic units named 'ASV_Clusters', which divides prokaryotic communities in our samples to a level between genus and species. Clustering 4,310 18S-V4 ASVs at 97% similarity level resulted in 24 operational taxonomic units named 'ASV_Clusters', which divides eukaryotic communities in our samples to a level between genus and species. Clustering 2,093 ITS2 ASVs at 97% similarity level resulted in 647 operational taxonomic units named 'ASV_Clusters', which divides fungal communities in our samples to a level between genus and species. Taxonomic classification of ASV_Clusters was determined by consensus of all ASVs that belong to the same ASV_Cluster.

Alpha diversity (16S, 18S and ITS2) in each sample was represented by the number of ASV_Clusters. To assess beta diversity, non-metric multidimensional scaling using UniFrac distance metric[56] was performed with the R package phyloseq v.1.8.0 (ref. [57]). Permutational multivariate analysis of variance[58] was performed to test the hypothesis that the community composition of antibiotic-free consortia at G10 was the same regardless of the substrate type, and it was implemented by the R package vegan[59] with 9,999 permutations.

**Metagenome assembly and analysis.** Metagenomic reads were quality filtered with the BBDuk tool in the BBMap software package (v.38.00)[60] at the JGI as part of their standard analysis pipeline, and we performed additional quality filtering of the reads using Trimmomatic[61]. Metagenomic assembly was performed with SPAdes 3.11.1 (ref. [62]) using *k*mers of the lengths 21, 33, 55, 77, 99 and 127. Read mapping was performed using bowtie2 (ref. [63]) with default settings and unsupervised genome binning was performed using MetaBat2 (ref. [64]). The quality of prokaryotic MAGs was evaluated using CheckM[8]. All MAGs were pooled and then de-replicated with dRep[10] using the parameters '-comp 80 -con 10–S_ algorithm gANI -sa 0.965 -nc 0.6', which de-replicate genome bins at the species level[9], resulting in 719 MAGs. A previous study reconstructed 913 MAGs from the cow rumen using a 99% average nucleotide identity threshold[3], but the number of unique cow rumen MAGs is only 627 at the species-level threshold used to define MAGs in the present study (96.5% similarity and 60% alignment)[9,10].

Some 18 MAGs of size >40 Mbp were reconstructed from the enrichment consortia, and were termed 'eukMAGs' because >80% of the genes in the eukMAGs were classified as 'Eukaryota' by BLAST+[17]. The completeness of the eukMAGs were estimated by benchmarking the number of universal single-copy orthologues to those common to the fungal kingdom using the tool BUSCO[19,20]. The eukMAGs' taxonomy was determined by performing whole-genome average nucleotide comparison with the isolated genomes of anaerobic fungi using dRep, and constructing a phylogenomic tree based on 60 single-copy orthologues that were present in >60% of the eukMAGs. A phylogenetic tree including all MAGs and eukMAGs was constructed using PhyloPhlAn[65] and visualization of the tree with CAZyme counts were performed using the interactive Tree of Life[66]. Taxonomic classification of the MAGs from the present study was inferred from a phylogenetic tree including all MAGs and reference genomes from the Hungate Collection, the GEBA and additional genomes from NCBI RefSeq (Supplementary Table 2 and Supplementary Data 14).

To calculate the coverage of each of the de-replicated MAGs and anaerobic fungi in each metagenome sample, all raw reads from that sample were mapped using bowtie2 (ref. [63]) to a concatenated fasta file including the 719 prokaryotic MAGs, the genomes of *Neocallimastix californiae* and *Caecomyces churrovis*, and the *Piromyces* eukMAG ag0r2_cm_bin.1_171. Note that for read mapping we selected the genomes of *N. californiae* and *C. churrovis* because they represent a more complete version of the corresponding eukMAGs, which share ~99% average nucleotide identity with them (Supplementary Data 5). The average coverage of each MAG and eukMAG/anaerobic fungal genome was calculated as:

$$\text{Average coverage} = \frac{\sum(\text{Coverage of each contig} \times \text{Length of the contig})}{\sum \text{Length of each contig}}.$$

A quality filter was applied that requires the ratio of s.d. to mean contig coverage in each MAG to be <0.35, which removes a few low-abundance MAGs that exhibited highly uneven contig coverage. MAGs removed by this automated quality filter were manually examined to ensure that they were truly absent from the corresponding sample. The sum of average coverage of all 719 MAGs in each sample is summarized in Supplementary Table 12. The relative abundance (%) of each MAG in each metagenome sample is calculated as the product of the average coverage and the size of the MAG (bp) divided by the total size of the metagenome (bp), normalized by the CheckM-estimated completeness of the MAG.

Assembled scaffolds were uploaded to IMG/M[67] for functional annotation; eukMAGs were annotated using Augustus[68,69] trained on the genome of *Neocallimastix californiae*. CAZyme genes in MAGs were identified by their protein family (Pfam) domains[70] (Supplementary Data 15). Genes annotated as GH and PL were divided into three categories based on the Carbohydrate-Active EnZymes database[21,71] (Supplementary Table 3): cellulase (GH5, GH6, GH7, GH8, GH9, GH12, GH44, GH45, GH48, GH51, GH74 and GH124), hemicellulase (GH5, GH8, GH10, GH11, GH16, GH26, GH30, GH43, GH44, GH51, GH62, GH98 and GH141) and pectinase/esterase (GH28, PL1, PL2, PL3, PL9, PL10, CE1, CE2, CE3, CE4, CE5, CE6, CE7, CE8, CE12, CE15 and CE13). The presence of CAZymes

defines each MAG's hydrolytic capability. Box plots comparing the number of different families of CAZymes between MAG and eukMAGs and between different antibiotic treatments were generated using the R package ggplot2 (ref. [72]). Analysis of variance (ANOVA) was used to test the hypothesis that the number of CAZymes within each family is the same between MAGs and eukMAGs and between different antibiotic treatments. Dunn's test[73] was performed to perform pair-wise comparisons and a table summarizing results was generated using the R package rcompanion[74].

Furthermore, dockerin-containing CAZymes were found by looking for enzymes with any CAZyme domain listed above, in addition to containing a domain assigned to the Pfams 02013 and 00404 for fungal- and bacterial-type dockerin domains, respectively. Cohesin domains were found by searching for Pfam 00963. The completeness of catabolic pathways to ferment different types of monosaccharides and uronic acids, and the completeness of pathways to produce SCFAs and $CH_4$, were calculated for each MAG based on Kyoto Encyclopedia of Genes and Genomes (KEGG)[75] and MetaCyc database of metabolic pathways and enzymes[76]. The enzyme commission (EC) numbers associated with each metabolic pathway are shown in Supplementary Data 16. When multiple modules could complete a pathway, the most complete (highest fraction of identified EC numbers relative to the total number of EC numbers required for that module) was used. Pathways >75% complete were considered present and included in the corresponding MAG or eukMAG.

We searched for CGCs using the dbCAN2 (ref. [77]) standalone tool (v.2.0.11) with the CGC-Finder[33] turned on. The CGC-Finder defines CGCs as genomic regions containing at least one CAZyme gene, one transporter gene (classified against the Transporter Classification Database[78]) and one transcription factor gene (classified against the databases CollecTF[79], RegulonDB[80] and DBTBS[81]). Of particular interest among all the CGCs are the PULs found in Bacteroidetes and those CGCs containing at least one ABC transporter commonly found in Gram-positive bacteria. PULs were identified by searching for the presence of SusC/D pairs as detailed previously[34], implemented by the tool PULpy[35]. ABC transporter-containing CGCs were identified by searching for the main Pfam PF00005 as well as the following ABC transporter-associated families: PF00950, PF03109, PF01061, PF12679, PF12698, PF12730, PF13346, PF06182, PF00664, PF06472, PF13748, PF04392, PF09822, PF06541, PF14510 and PF16949.

**Measurement of hydrolytic performance.** A follow-up experiment was performed to compare the hydrolytic potential of an antibiotic-free consortium enriched on reed canary grass and a consortium treated with PS enriched on alfalfa stems. On each assayed consortium, 1 ml of consortium was inoculated in triplicate into 9 ml of MC− containing 100 mg of cellulose filter paper (Whatman, catalogue no.1001-090) or reed canary grass (US Department of Agriculture, Agricultural Research Service US Dairy Forage Research Center). The weight loss of the substrates was determined by lyophilizing the remaining substrate after 7 d of growth using a FreeZone 4.50-l Benchtop Freeze Dry System (Labconco Corp., part no. 7750020). The percentage of substrate remaining was calculated as the final mass remaining after lyophilization divided by 100 mg. A DNS assay using the low concentration conditions as described above was performed to measure the amount of reducing sugar in the spent medium at the end of 7 d of microbial growth.

**Cryopreservation analysis of enriched microbial consortia.** The consortium treated with PS enriched on alfalfa stems was cryogenically preserved following a previously described protocol[42]. In short, after G10 this consortium treated with PS enriched on alfalfa stems was continually subcultured every 4 d. At G145, after transferring the culture into fresh culture medium, all the remaining supernatant was removed using a pipette under a $CO_2$ atmosphere, leaving behind the undigested plant material and microbial biomass in the culturing vessel. Then fresh culture medium MC− supplemented with 15% glycerol was added back into the culture vessel and mixed well. The new mixture was transferred to 2-ml cryogenic vials (Corning, catalogue no. 430488) and flash frozen anaerobically in liquid nitrogen before storing at −80 °C. After storage at −80 °C for 10 d, the cryopreserved enrichment culture was thawed anaerobically at room temperature (20 °C) and the content in each cryogenic vial was transferred into 9 ml of fresh culture medium MC− with alfalfa stems under a $CO_2$ atmosphere and incubated at 39 °C. The production of total gas, $H_2$ and $CH_4$ was measured as described above.

Three replicates of the cryorevived cultures treated with PS enriched on alfalfa stems were revived for community composition analysis using marker genes. DNA was extracted from these samples following the same procedure described above. The V4 region of the 16S rRNA was amplified as well as the second region of the ITS2 for amplicon sequencing in the Biological Nanostructures Lab at the University of California, Santa Barbara. The Illumina 16S Metagenomic Sequencing Library Preparation[82] was followed to prepare the sequencing library with the following modifications. Amplicon PCR for 16S-V4 region was performed using primers 16S-515F_Adapter (5′-TCGTCGGCAGCGTCAGATGTGTAT AAGAGACAGGTGCCAGCMGCCGCGGTAA-3′) and 16S-805R_Adapter (5′-TCTCGTGGGCTCGGAGATGTGTATAAGAGACAGGGACTACH VGGGTWTCTAAT-3′) and Phusion high-fidelity DNA polymerase (New England BioLabs, catalogue no. M0530). The thermal cycle started with 30 s at 98 °C, followed by 30 cycles of 10 s at 98 °C, 30 s at 50 °C and 45 s at 72 °C. The final elongation at 72 °C was 10 min long. Amplicon PCR for the second region

of the internal transcribed spacer was performed using primers ITS9_Adapter (5′-TCGTCGGCAGCGTCAGATGTGTATAAGAGACAGGAACGCAGCRA AIIGYGA-3′) and ITS4 (5′- GTCTCGTGGGCTCGGAGATGTGTATAAGA GACAGTCCTCCGCTTATTGATATGC-3′) and Phusion high-fidelity DNA polymerase. The thermal cycle started with 30 s at 98 °C, followed by 30 cycles of 10 s at 98 °C, 30 s at 48 °C and 45 s at 72 °C. The final elongation at 72 °C was 10 min long. The index PCR was performed with Phusion high-fidelity DNA polymerase. The thermal cycle started with 30 s at 98 °C, followed by 8 cycles of 10 s at 98 °C, 30 s at 55 °C and 30 s at 72 °C. The final elongation at 72 °C was 10 min long. The quantity and quality of final PCR products were determined using a Qubit (Thermo Fisher Scientific) and TapeStation 2200 (Agilent). Identical quantities of each sample were pooled, and the products were sequenced on an Illumina MiSeq 2×300 PE platform. We sequenced 21 samples together with 19 samples obtained in another study. Sequencing results were analysed in DADA2 following the same procedure described above.

**Reporting Summary.** Further information on research design is available in the Nature Research Reporting Summary linked to this article.

## Data availability

The metagenome-sequencing reads can be accessed at the JGI under JGI Project IDs and at the Sequence Read Archive by their SRA IDs listed in Supplementary Data 13. Contigs for each MAG are available at NCBI's Whole Genome Shotgun database under accession nos. SAMN11294286–SAMN11295004 (Supplementary Data 4) and project no. PRJNA530070 (https://www.ncbi.nlm.nih.gov/bioproject/? term=prjna530070).

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

## Acknowledgements

We thank the following for funding support: the National Science Foundation (NSF, grant no. MCB-1553721), the Office of Science (BER) of the US Department of Energy (DOE) (grant no. DE-SC0010352), the Institute for Collaborative Biotechnologies (grant nos. W911NF-09-D-0001 and W911NF-19-2-0026) from the US Army Research Office, and the Camille Dreyfus Teacher-Scholar Awards Program. We also thank the California NanoSystems Institute (CNSI) Challenge Grant Program, supported by the University of California, Santa Barbara (UCSB) and the University of California, Office of the President, for their support. This work was part of the DOE Joint BioEnergy Institute (http://www.jbei.org) supported by the Office of Biological and Environmental Research of the DOE Office of Science through contract no. DE-AC02–05CH11231 between Lawrence Berkeley National Laboratory and the DOE. S.E.W. received funding support from the Dow Discovery Fellowship and C.L.S. from the NSF Graduate Research Fellowship Program. The sequencing conducted by the US DOE Joint Genome Institute, a DOE Office of Science User Facility, is supported by the Office of Science of the US DOE under contract no. DE-AC02-05CH11231. This work used the computing capabilities of the Extreme Science and Engineering Discovery Environment (XSEDE), which is supported by NSF grant no. ACI-1548562. Specifically, it used the Bridges system, which is supported by NSF award no. ACI-1445606, at the Pittsburgh Supercomputing Center. Additional supercomputing resources were provided by the Center for Scientific Computing (CSC) at UCSB, which is supported by NSF grant no. CNS-0960316. The CSC is supported by the California NanoSystems Institute and the Materials Research Science and Engineering Center (MRSEC; NSF DMR 1720256) at UCSB. We thank B. Henrissat for feedback and discussion on this manuscript and the CAZyme analysis.

## Author contributions

X.P., S.P.G., J.K.H., C.L.S., M.K.T., D.L.V. and M.A.O. conceived of the study. X.P., S.P.G., J.K.H., C.L.S. and M.K.T. participated in the study design. X.P., S.P.G., J.K.H. and C.L.S. performed the enrichment experiments. S.W. and T.S.L. performed chemical analysis and modelling experiments after the enrichment experiment. X.P., S.W., T.S.L., S.P.G., J.L.B., C.L.S., D.L.V. and M.A.O. processed the data and performed computational analysis. I.V.G., A.S. and K.B. performed, coordinated and aided in the analysis of DNA-sequencing, and the IMG/M provided functional annotation of metagenomes. X.P. managed the logistics of data collection and integration. X.P. and M.A.O. were the primary writers of the paper, with contributions from S.W. and T.S.L. All authors reviewed and approved the final manuscript.

## Competing interests

The authors declare no competing interests.

## Additional information

**Extended data** is available for this paper at https://doi.org/10.1038/s41564-020-00861-0.

**Correspondence and requests for materials** should be addressed to M.A.O.

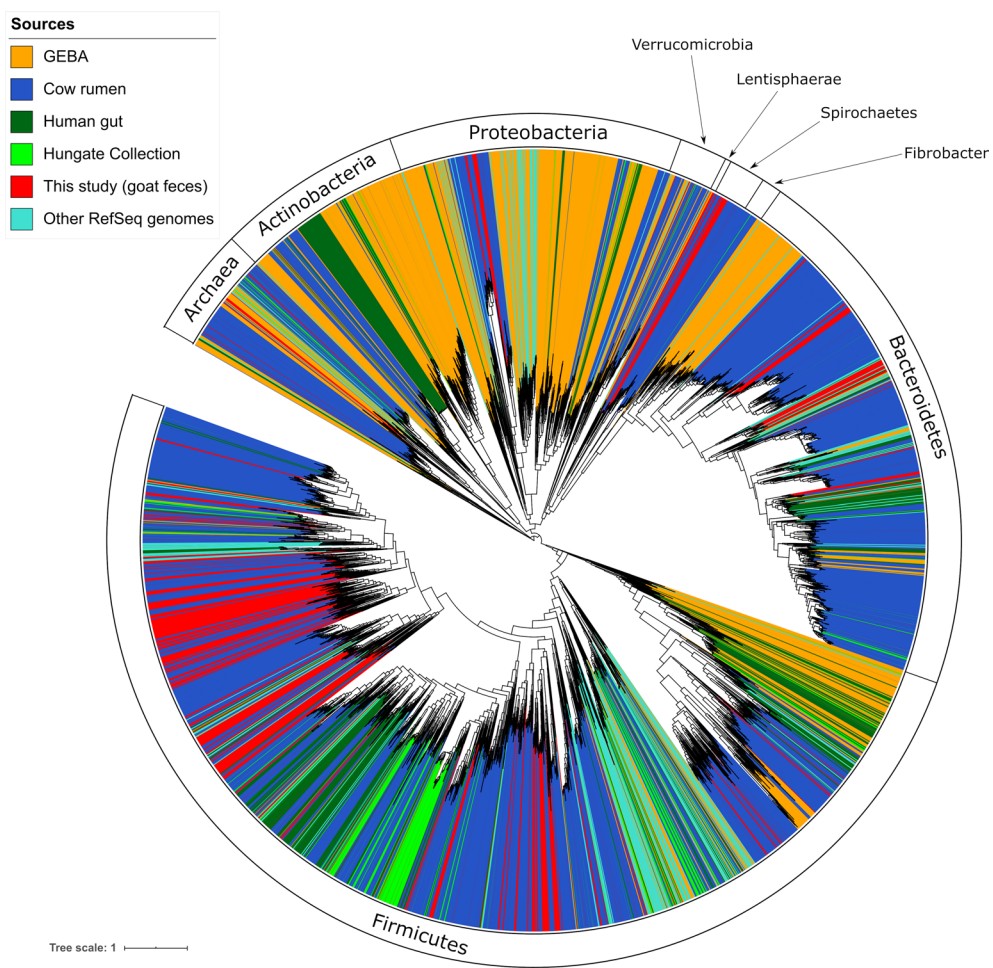

**Extended Data Fig. 1 | Phylogenomic tree including the 719 metagenome-assembled genomes (MAGs) reconstructed from the goat fecal microbiome and reference genomes and MAGs.** The phylogeny was constructed from 400 broadly conserved proteins using PhyloPhlAn[65] and visualized using interactive Tree of Life[66]. The leaves of the tree were colored by the genome's source: red for the MAGs from this study, bright green for the Hungate Collection[2], blue for the MAGs from cow rumen[3,4], dark green for bacterial genomes from human gut[11], orange for the Genomic Encyclopedia of Bacteria and Archaea isolates[15], and cyan for additional reference genomes retrieved from the RefSeq database at NCBI[16] (Supplementary Data 14). Phylum-level classification is labeled at the outer ring of the tree. Note that only the genomes and MAGs dereplicated (Supplementary Data 2) at the species level (96.5% average nucleotide identity and 60% alignment coverage) are represented in this tree.

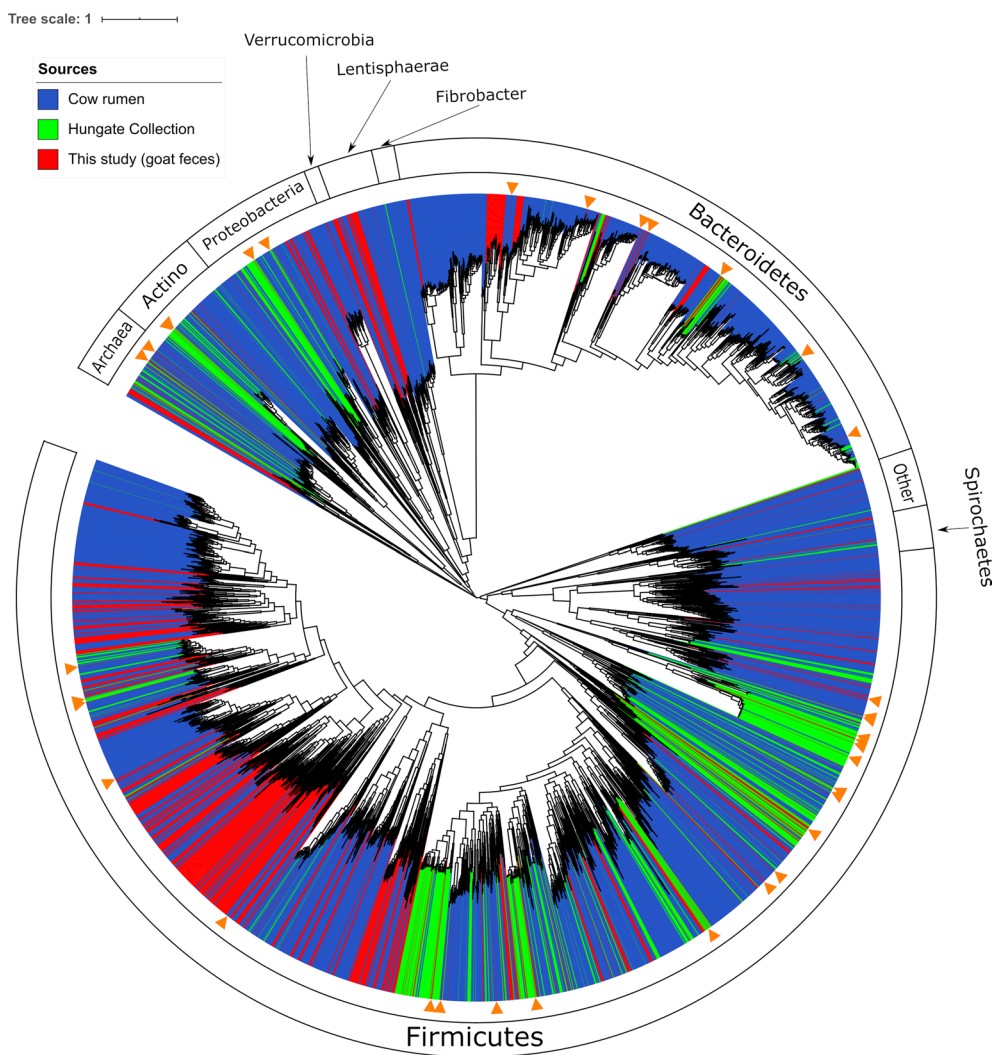

**Extended Data Fig. 2 | Phylogenomic tree including the 719 metagenome-assembled genomes (MAGs) reconstructed from the goat fecal microbiome and reference genomes and MAGs from the Hungate Collection and cow rumen.** The phylogeny was constructed from 400 broadly conserved proteins using PhyloPhlAn[65] and visualized using interactive Tree of Life[66]. Phylum-level classification is labeled at the outer ring of the tree. 'Actino' represents Actinobacteria. MAGs under 'Other' are primarily *Firmicutes*. The leaves of the tree were colored by the genome's source: red for the 719 MAGs from this study, bright green for the 493 genomes of cultivated rumen bacteria and archaea from the Hungate Collection[2], and blue for a dereplicated set of cow rumen MAGs[3,4]. The cow rumen MAGs were dereplicated at the species level (96.5% similarity and 60% alignment). The 35 MAGs from this study that are <u>not</u> novel at the species level compared to references are marked with orange triangles.

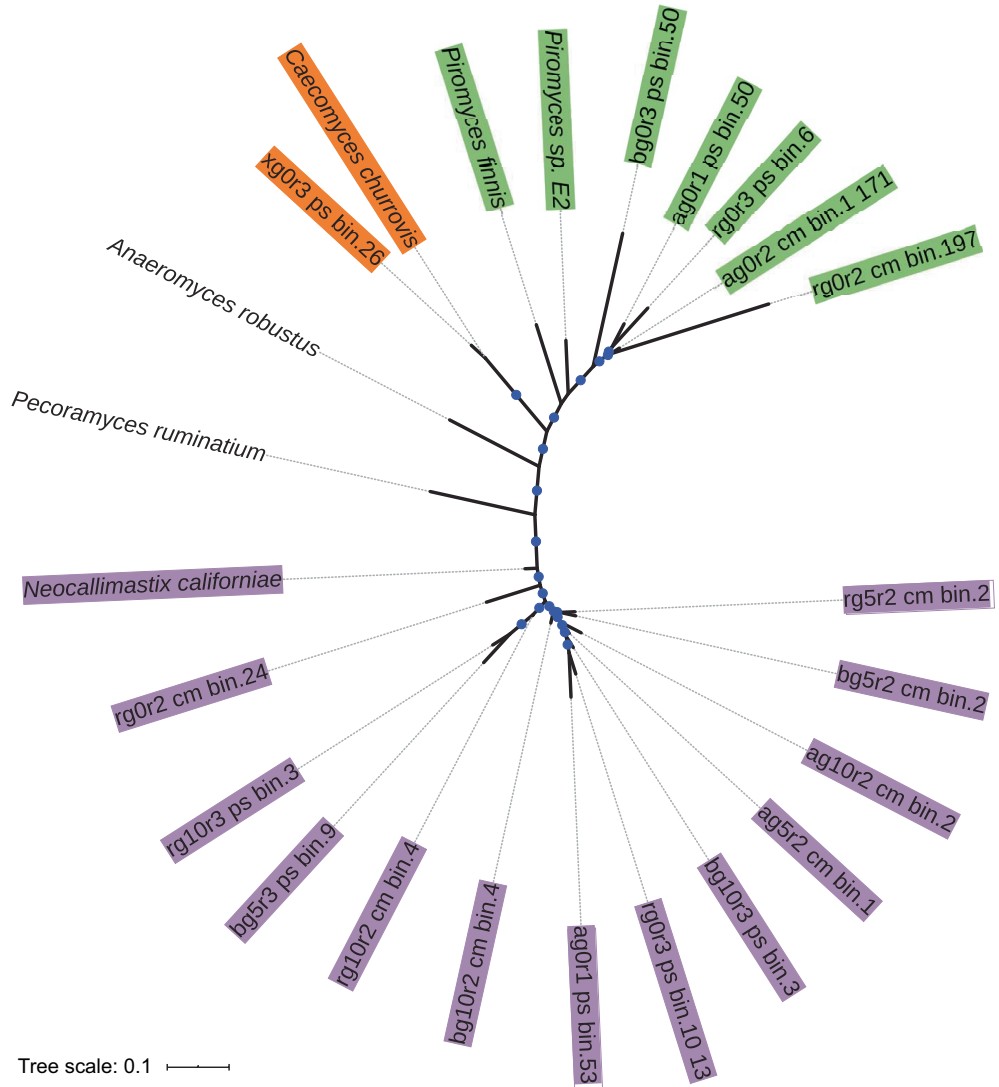

Tree scale: 0.1

**Extended Data Fig. 3 | Unrooted phylogenomic tree of 18 eukaryotic metagenome-assembled genomes (eukMAGs) and six genomes from Neocallimastigomycota.** To reconstruct the phylogeny, 60 single-copy orthologs retrieved using BUSCO[19] (v3.0.2) from OrthoDB[20] (v9.1) were individually aligned and subsequently concatenated before tree building using FastTree[83] (v2.1.1). Interactive tree of life[66] (iTOL v5.5.1) was used to generate the visuals. SH (Shimodaira-Hasegawa test resample size = 1000) values greater than 0.9 were marked with a blue circle on the corresponding branch. Light purple was used to highlight the genome of *Neocallmastix californiae* and eukMAGs from the same genus. Green was used to highlight the genome of *Piromyces finnis* and *Piromyces sp. E2* and eukMAGs from the same genus. Orange was used to highlight the genome of *Caecomyces churrovis* and the eukMAG from the same genus. Grey dashed lines were used to connect the tip of each leaf and its label.

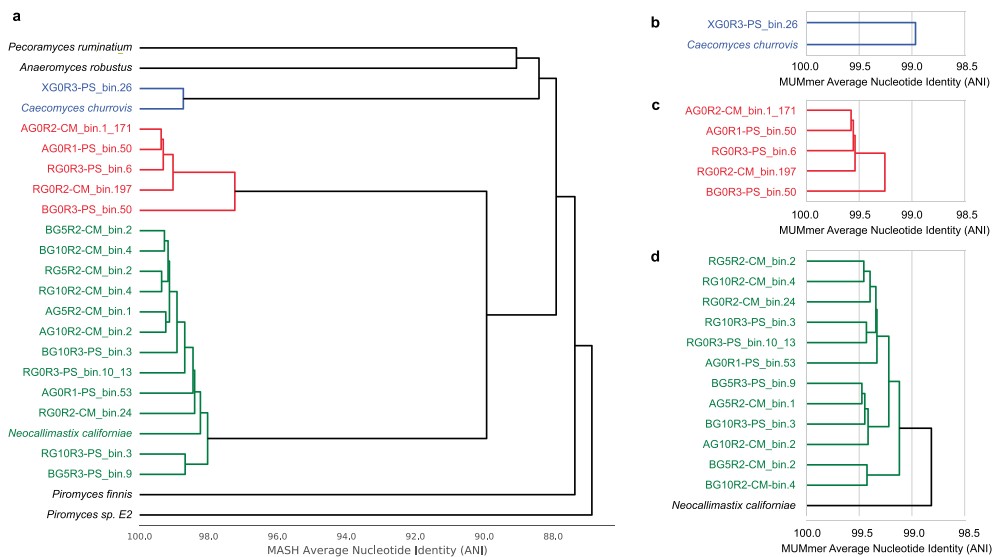

**Extended Data Fig. 4 | Genome-wide average nucleotide identity (ANI) between eukMAGs and previously sequenced genomes from the phylum Neocallimastigomycota. a**, Genome-wide ANI calculated by MASH[84] between the 18 eukMAGs recovered in this study and six previously sequenced genomes from the phylum *Neocallimastigomycota*. **b**, Genome-wide ANI calculated by MUMmer[85] between the the *Caecomyces* eukMAG and the genome of *Caecomyces churrovis*. **c**, Genome-wide ANI calculated by MUMmer[85] between the five *Piromyces* eukMAGs. **d**, Genome-wide ANI calculated by MUMmer[85] between the 12 *Neocallimastix* eukMAGs and the genome of *Neocallimastix californiae*. The tool dRep[10] was used to implement the ANI algorithms and construct the figures.

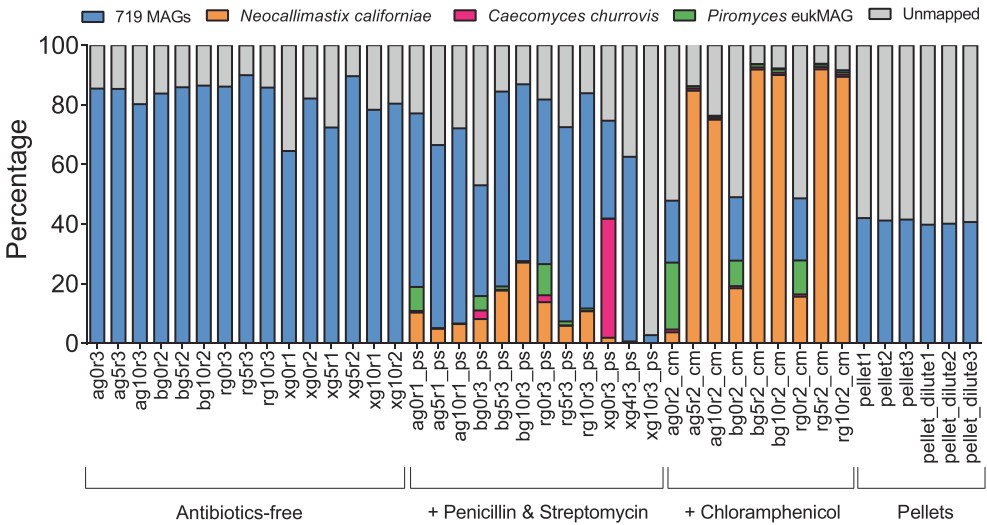

**Extended Data Fig. 5 | Percentage of raw metagenome reads mapped to MAGs, the genome of Neocallimastix californiae and Caecomyces churrovis, and the best Piromyces eukMAG.** Color bars were used to represent the percentage of reads from each sample that mapped to the collection of 719 MAGs (blue), the genome of *Neocallimastix californiae* (orange), the genome of *Caecomyces churrovis* (pink), and the best *Piromyces* eukMAG 'ag0r2_cm_bin.1_171' (green). Grey bars represent the percentage of unmapped reads. For each sample from each batch we have provided a unique identifier with the format 'SGxRy', where 'S' represent the carbon substrate ('A' for alfalfa stems, 'B' for bagasse, 'R' for reed canary grass, and 'X' for xylan), 'x' represents the batch number (0 through 10), and 'y' represents the replicate number (1, 2, or 3). For samples treated with penicillin and streptomycin, an additional '-PS' is appended to the end of the sample identifier. For samples treated with chloramphenicol, an additional '-CM' is appended to the end of the sample identifier.

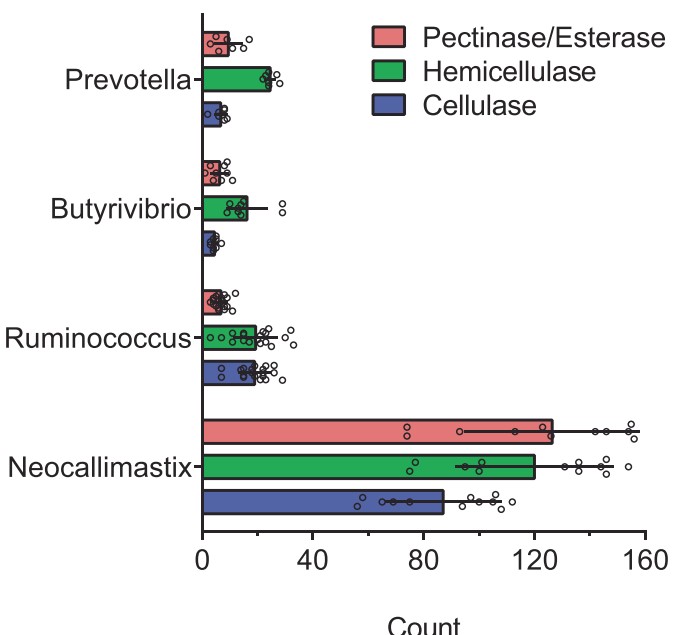

**Extended Data Fig. 6 | Cellulase, hemicellulase, and pectinase/esterase counts in MAGs from the genera Prevotella, Butyrivibrio, and Ruminococcus, and eukMAGs from the genus Neocallimastix.** The length of the bars represents the average value of counts and the error bars represent standard deviations (n = 7 for *Prevotella*; n = 9 for *Butyrivibrio*; n = 19 for *Ruminococcus*; n = 12 for *Neocallimastix*). The difference between *Neocallimastix* and each of the prokaryotic genera is significant for each type of CAZyme as confirmed by student's t-test (two-tailed p-value < 0.05). The p-value comparing the number of cellulases between *Neocallimastix* and any of the bacterial genera was $2.84 \times 10^{-8}$ for *Prevotella*, $2.78 \times 10^{-8}$ for *Butyrivibrio*, and $1.17 \times 10^{-7}$ for *Ruminococcus*. The p-value comparing the number of hemicellulases between *Neocallimastix* and any of the bacterial genera was $1.46 \times 10^{-7}$ for *Prevotella*, $2.04 \times 10^{-8}$ for *Butyrivibrio*, and $4.72 \times 10^{-8}$ for *Ruminococcus*. The p-value comparing the number of pectinases/esterases between *Neocallimastix* and any of the bacterial genera was $3.12 \times 10^{-8}$ for *Prevotella*, $3.72 \times 10^{-8}$ for *Butyrivibrio*, and $4.58 \times 10^{-8}$ for *Ruminococcus*.

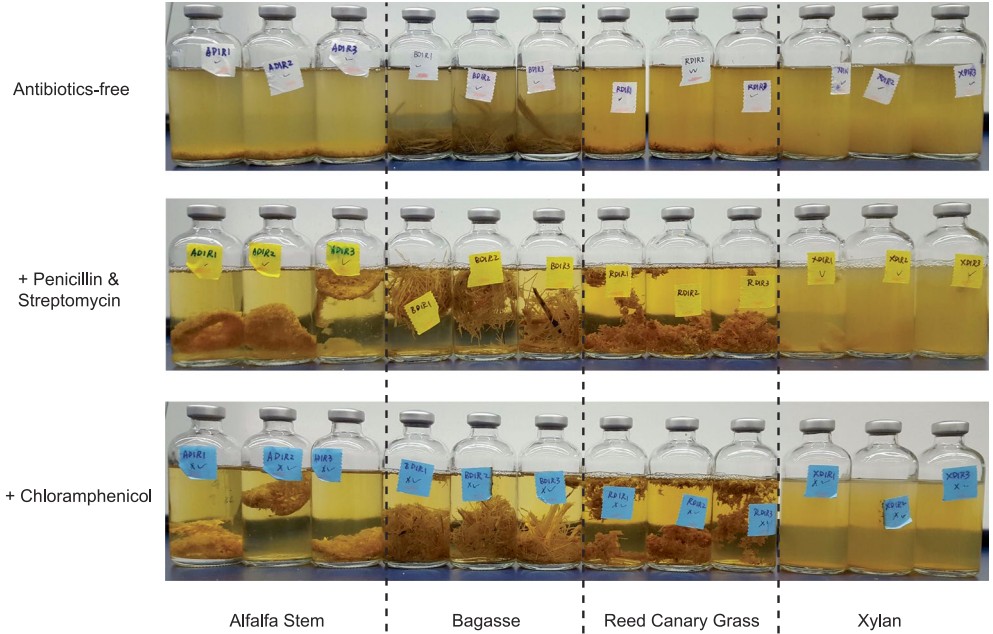

**Extended Data Fig. 7 | Photographs of representative microbial enrichment cultures after three days of growth.** Samples in the first row are antibiotics-free consortia; samples in the second row are penicillin and streptomycin-treated consortia; samples in the third row are chloramphenicol-treated consortia. Also visible is the difference in substrate (alfalfa stems, bagasse, reed canary grass, and xylan). There were three biological replicates per antibiotics treatment and carbon substrate. The coagulated plant material was a result of fungal growth, which penetrated and connected the plant material.

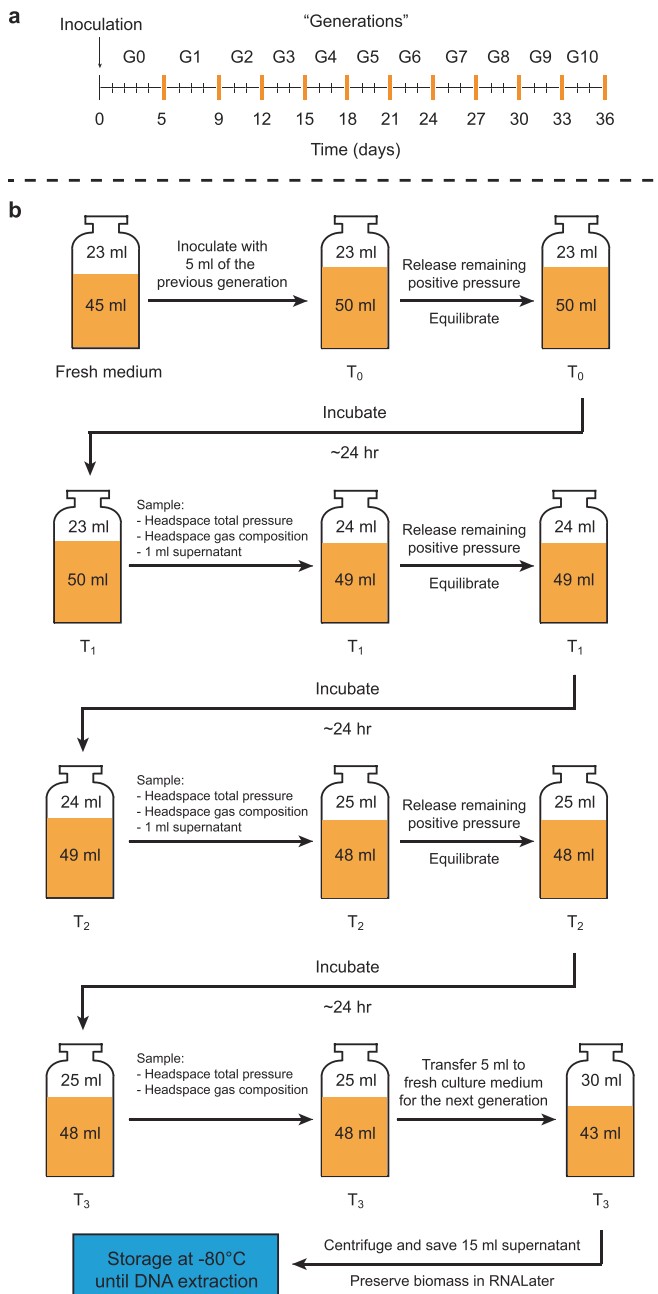

**Extended Data Fig. 8 | Timeline and workflow of all eleven batches of enrichment cultures characterized in this study. a**, The initial batch of enrichment cultures are referred to as 'G0' and the following batches are abbreviated from 'G1' to 'G10'. The thick orange marks on the timeline represent the time points when the previous batch was sampled for its chemical output, sub-cultured to fresh culture media to start the next batch, and preserved for nucleic acid extractions. The thin black marks between the thick orange marks on the timeline represent the time points when the enrichment cultures were sampled for their chemical output. **b**, Schematics showing headspace and supernatant sampling procedure during the maintenance of enrichment cultures. Briefly, 45 ml of fresh culture medium was inoculated with 5 ml of enrichment culture from the previous batch and allowed to incubate 24 hours until the first sampling time point. At each sampling time point, headspace gas pressure was measured with a pressure transducer and the hydrogen and methane concentration was determined with gas chromatography. After preserving 1 ml of supernatant for liquid metabolite analysis, the headspace pressure in each sample was equilibrated with the atmosphere by releasing any remaining positive pressure in the headspace. After the sampling time point after three days of incubation ($T_3$), 5 ml of the enrichment culture is transferred to fresh culture medium for starting the next batch. All biomass in the remaining samples from the current batch were harvested by centrifugation, preserved in RNA*later* RNA Stabilization Reagent (Qiagen Cat. No. 76106, Hilden, Germany), and stored at −80 °C until DNA extraction (see methods for additional details).

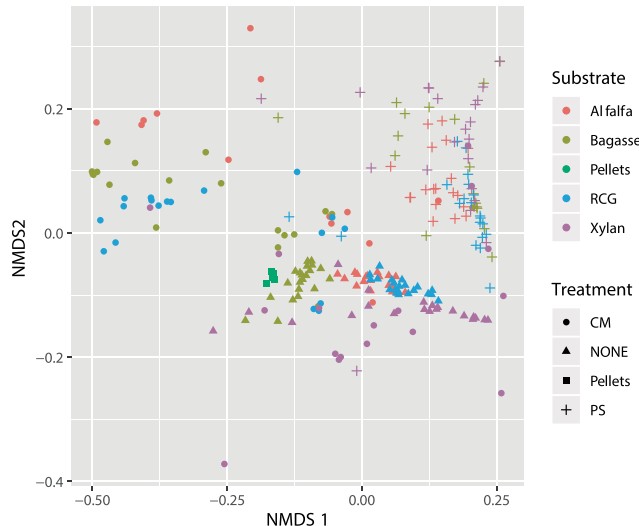

**Extended Data Fig. 9 | Biplot of non-metric multidimensional scaling (NMDS) showing the enrichment culture community composition evaluated by the amplicon sequence variant (ASV) clusters of the V4 region of the 16 S rRNA gene (16S-V4) as a function of antibiotics treatment and substrate.** Green squares represent pellets. Enrichment cultures grown on alfalfa are in red, on bagasse in olive, on reed canary grass (RCG) in blue, and on xylan in purple. Triangles represent antibiotics-free ('NONE') cultures; crosses represent penicillin and streptomycin-treated (PS) cultures; filled circles represent chloramphenicol-treated (CM) cultures. Stress of the NMDS was 0.11. Permutational multivariate analysis of variance (PERMANOVA)[58] was performed to test the hypothesis that the community composition of antibiotics-free consortia at generation 10 was the same regardless of the substrate type. The PERMANOVA was implemented by the R package vegan[59] with 9999 permutations and rejected the null hypothesis (pseudo-F = 12.797 and p = 0.0001).

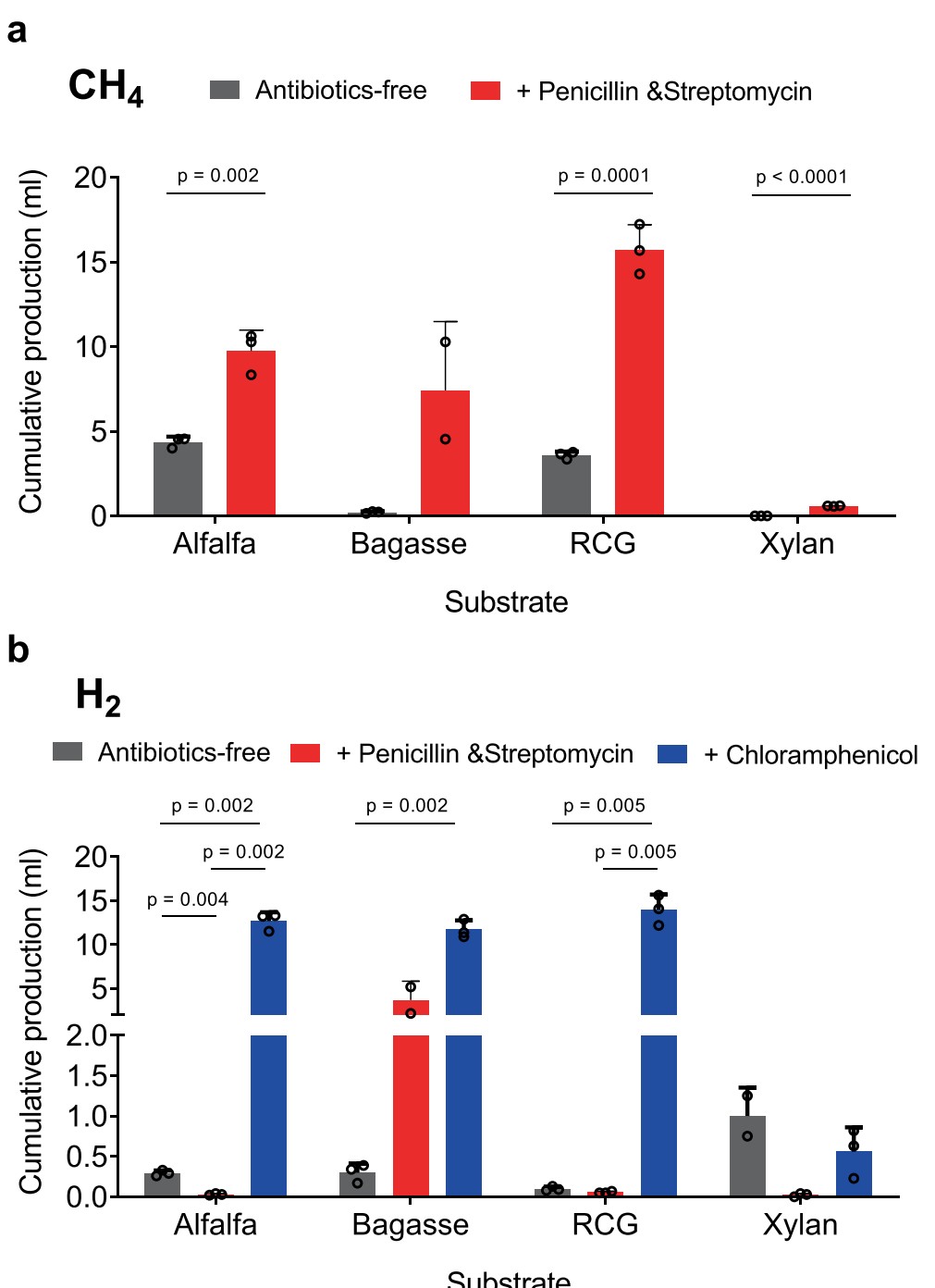

**Extended Data Fig. 10 | Average methane and hydrogen production from consortia at generation 10.** No methane ($CH_4$) was produced from chloramphenicol-treated (CM) consortia. The height of the bars represents the mean (n=3) and the error bars represent the standard deviations, except for PS consortia grown on bagasse, where only two replicates were available. 'RCG' represents the substrate reed canary grass. Only p-values smaller than 0.05 from two-tailed t-tests comparing different antibiotics treatment groups are shown.

# nature research

# Reporting Summary

Nature Research wishes to improve the reproducibility of the work that we publish. This form provides structure for consistency and transparency in reporting. For further information on Nature Research policies, see Authors & Referees and the Editorial Policy Checklist.

## Statistics

For all statistical analyses, confirm that the following items are present in the figure legend, table legend, main text, or Methods section.

| n/a | Confirmed | |
|---|---|---|
| ☐ | ☒ | The exact sample size (*n*) for each experimental group/condition, given as a discrete number and unit of measurement |
| ☐ | ☒ | A statement on whether measurements were taken from distinct samples or whether the same sample was measured repeatedly |
| ☐ | ☒ | The statistical test(s) used AND whether they are one- or two-sided<br>*Only common tests should be described solely by name; describe more complex techniques in the Methods section.* |
| ☒ | ☐ | A description of all covariates tested |
| ☒ | ☐ | A description of any assumptions or corrections, such as tests of normality and adjustment for multiple comparisons |
| ☐ | ☒ | A full description of the statistical parameters including central tendency (e.g. means) or other basic estimates (e.g. regression coefficient) AND variation (e.g. standard deviation) or associated estimates of uncertainty (e.g. confidence intervals) |
| ☐ | ☒ | For null hypothesis testing, the test statistic (e.g. *F*, *t*, *r*) with confidence intervals, effect sizes, degrees of freedom and *P* value noted<br>*Give P values as exact values whenever suitable.* |
| ☒ | ☐ | For Bayesian analysis, information on the choice of priors and Markov chain Monte Carlo settings |
| ☒ | ☐ | For hierarchical and complex designs, identification of the appropriate level for tests and full reporting of outcomes |
| ☒ | ☐ | Estimates of effect sizes (e.g. Cohen's *d*, Pearson's *r*), indicating how they were calculated |

*Our web collection on statistics for biologists contains articles on many of the points above.*

## Software and code

Policy information about availability of computer code

| Data collection | Chromeleon Chromatography Data System 7, OpenLab CDS ChemStation Edition C.01.02. |
|---|---|
| Data analysis | USEARCH v11, DADA2 v1.8.0, blast v2.7.1, bowtie2 v2.3..4..1, bbmap v38.00, Trimmomatic v0.36, SPAdes 3.11.1, MetaBat2 v2.12.1, CheckM v1.0.7, dRep v2.2.2, BUSCO v3, PhyloPhlAn v1.10, iTOL 4.4.2, IMG/M v5.0. |

For manuscripts utilizing custom algorithms or software that are central to the research but not yet described in published literature, software must be made available to editors/reviewers. We strongly encourage code deposition in a community repository (e.g. GitHub). See the Nature Research guidelines for submitting code & software for further information.

## Data

Policy information about availability of data

All manuscripts must include a data availability statement. This statement should provide the following information, where applicable:
- Accession codes, unique identifiers, or web links for publicly available datasets
- A list of figures that have associated raw data
- A description of any restrictions on data availability

The metagenome sequencing reads can be accessed at the Joint Genome Institute under JGI Project ID's and at the NCBI Sequence Read Archive under the SRA ID's listed in Supplementary Data 13. Contigs for each MAG are available at NCBI's Whole Genome Shotgun database under accession numbers SAMN11294286 - SAMN11295004 (Supplementary Table 3) and project number PRJNA530070 (https://www.ncbi.nlm.nih.gov/bioproject/?term=prjna530070).

# Field-specific reporting

Please select the one below that is the best fit for your research. If you are not sure, read the appropriate sections before making your selection.

☒ Life sciences          ☐ Behavioural & social sciences          ☐ Ecological, evolutionary & environmental sciences

For a reference copy of the document with all sections, see nature.com/documents/nr-reporting-summary-flat.pdf

# Life sciences study design

All studies must disclose on these points even when the disclosure is negative.

| | |
|---|---|
| Sample size | In our experiment, we included enrichment cultures based on four different types of substrates in order to investigate the influence of substrate composition on the membership of microbial consortia. We included three antibiotics treatments to separate the fungal and bacterial parts of the source microbiome. We chose to sample from eleven culture generations because previous enrichment cultures in the lab indicate that should be sufficient to allow the community composition of the enrichment cultures to stabilize. |
| Data exclusions | No data were excluded. |
| Replication | This study is a large-scale enrichment experiment and it is not feasible for us to perform an experiment that is exactly the same. However, the experimental design of this study itself provides evidence of reproducibility, because the similar types of microbial memberships were enriched in different types of antibiotics treatment, regardless of the type of carbon substrate used in the enrichment culture. Bacteria from phylum Firmicutes dominated antibiotics-free cultures; anaerobic fungi and methanogenic archaea dominated penicillin & streptomycin-treated cultures; anaerobic fungi alone dominated chloramphenicol-treated cultures. |
| Randomization | This is not relevant to our study because the initial inoculum used for all enrichment cultures was generated by homogenizing multiple fecal pellets before inoculation. |
| Blinding | Blinding was not relevant to our study because we intended to study the fecal microbiome of an individual goat. |

# Reporting for specific materials, systems and methods

We require information from authors about some types of materials, experimental systems and methods used in many studies. Here, indicate whether each material, system or method listed is relevant to your study. If you are not sure if a list item applies to your research, read the appropriate section before selecting a response.

## Materials & experimental systems

| n/a | Involved in the study |
|---|---|
| ☒ ☐ | Antibodies |
| ☒ ☐ | Eukaryotic cell lines |
| ☒ ☐ | Palaeontology |
| ☒ ☐ | Animals and other organisms |
| ☒ ☐ | Human research participants |
| ☒ ☐ | Clinical data |

## Methods

| n/a | Involved in the study |
|---|---|
| ☒ ☐ | ChIP-seq |
| ☒ ☐ | Flow cytometry |
| ☒ ☐ | MRI-based neuroimaging |

