## [Peer Review File · Nature Microbiology]

Peer Review Information

Journal: Michelle O Malley

Manuscript Title: Genomic and functional analyses of fungal and bacterial consortia that enable lignocellulose breakdown in goat gut microbiomes

Corresponding author name(s): Michelle O Malley

Reviewer Comments & Decisions:

Decision Letter, initial version:
--

Dear Michelle,

Thank you for your patience while your manuscript "Sculpting gut microbial communities alters fermentation products and methane release" was under peer-review at Nature Microbiology. It has now been seen by 4 referees, whose expertise and comments you will find at the of this email. Although they find your work of some potential interest, they have raised a number of concerns that will need to be addressed before we can consider publication of the work in Nature Microbiology.

In particular, referees #2, #3 and #4 all ask for further methods details, especially for the approaches used to recover and analyse the fungal MAGs as well as details of their completeness and contamination levels. Referee #3 requests that statistics are added throughout, that the paper be written in a clearer and more concise way so that the key findings are clear to the reader, and additional details on the enrichment procedures used. Referee #4 asks for additional analyses including fungal ITS2 sequences under 360 bases, and has other concerns with the fungal methods including the barcoding, primer specificity and the need for more specific discussion on the fungal enzymes. This referee also requests that you justify the model used, including the limitations of the study, which should be clear and upfront at the beginning of the discussion. Referee #1 also requests that you put the findings in the context of previous work and identify the most efficient combination of organisms for lignocellulose degradation. While referee #4 asks that you use the recovered RNA for further analyses such as metatranscriptomics, and sampling of the hindgut, we are willing to overrule this request.

Should further experimental data allow you to address these criticisms, we would be happy to look at a revised manuscript.

We are committed to providing a fair and constructive peer-review process. Please do not hesitate to contact us if there are specific requests from the reviewers that you believe are technically impossible

or unlikely to yield a meaningful outcome.

Please include a data availability statement as a separate section after Methods but before references, under the heading "Data Availability". This section should inform readers about the availability of the data used to support the conclusions of your study. This information includes accession codes to public repositories (data banks for protein, DNA or RNA sequences, microarray, proteomics data etc...), references to source data published alongside the paper, unique identifiers such as URLs to data repository entries, or data set DOIs, and any other statement about data availability. At a minimum, you should include the following statement: "The data that support the findings of this study are available from the corresponding author upon request", mentioning any restrictions on availability. If DOIs are provided, we also strongly encourage including these in the Reference list (authors, title, publisher (repository name), identifier, year). For more guidance on how to write this section please see:

<http://www.nature.com/authors/policies/data/data-availability-statements-data-citations.pdf>

* If you have not done so already we suggest that you begin to revise your manuscript so that it conforms to our Article format instructions at <http://www.nature.com/nmicrobiol/info/final-submission>. Refer also to any guidelines provided in this letter.

{REDACTED}

Note: This url links to your confidential homepage and associated information about manuscripts you may have submitted or be reviewing for us. If you wish to forward this e-mail to co-authors, please delete this link to your homepage first.

Nature Microbiology is committed to improving transparency in authorship. As part of our efforts in this direction, we are now requesting that all authors identified as 'corresponding author' on published papers create and link their Open Researcher and Contributor Identifier (ORCID) with their account on

the Manuscript Tracking System (MTS), prior to acceptance. This applies to primary research papers only. ORCID helps the scientific community achieve unambiguous attribution of all scholarly contributions. You can create and link your ORCID from the home page of the MTS by clicking on 'Modify my Springer Nature account'. For more information please visit www.springernature.com/orcid.

If you wish to submit a suitably revised manuscript we would hope to receive it within 6 months. If you cannot send it within this time, please let us know. We will be happy to consider your revision, even if a similar study has been accepted for publication at Nature Microbiology or published elsewhere (up to a maximum of 6 months). Should your manuscript be substantially delayed without notifying us in advance and your article is eventually published, the received date would be that of the revised, not the original, version.

Reviewer Comments:

Reviewer #1 (Remarks to the Author):

1. General comments:

The experimental design, the metagenomics analysis (except for not including the latest MAGs of the publication of Stewart et al. (2019) Nature Biotechnology), the enrichment analysis, etc. are very well carried out and the results provide novel insight particularly into the fungal community and its connectivity to the bacteria community. However, the abstract sounds like that only fungi would be needed for efficient lignocellulose degradation. But even your results indicate that fungi do not contain all important glycoside hydrolase and therefore a combination of bacteria and fungal community would probably be the most efficient way to produce methane from lignocellulose. Therefore, the authors should investigate whether their data provides an opportunity to suggest an optimum combination of specific bacteria and fungi to maximize the production of methane. In addition the following specific comments have to be addressed:

2. Specific comments:

Line Comment

L99-100 Could you provide a reference about the amount of microbes located in the hindgut often originate from and were active in the rumen, because many are digested and therefore probably not in the feces. Is the composition of the rumen microbiota not very different from the feces microbiota?
L114 Please correct to Verrucomicrobia.

Figure 3 In this figure, methonogenic archaea seem to use formate as main substrates to produce methane. Please note that Grennings et al. (2019) found that formate is not the main substrate for rumen methanogenesis.

L153-155 This comparison does result in a bias prediction because there are probably many prokaryotes, which are not or in a minor extent involved in fermentation of lignocellulose. Are you including those MAGs having no lignocellulose-active genes in your calculation of the average MAG? It would be more interesting to compare the number of lignocellulose-active genes in the top three eukMAG with *Ruminococcus*, *Butyrivibrio* and *Prevotella*. Moreover, it is questionable to determine hydrolytic potential of *Neocallimastigomycota* by the number of lignocellulose-active genes. Of more interest would be the efficiency of these lignocellulose-active genes to provide substrates for the methanogenic archaea to produce methane.

L184-186 Are the in vitro conditions not suitable for the *Ruminococcus* that its relative abundance was similar to the source microbiome? Is this an indication that in vitro studies do not resemble in vivo studies in animals?

L208-209 This could be written more clearly in order to indicate that these microbes were available in the source microbiome but at a level that was not detectable.

L300-302 What kind of functional evidence you have that PS-resistant Firmicutes preventing catabolite repression of anaerobic fungi?

L370-372 This is well known from animal trials comparing forage and concentrate diets, e.g. Roehe et al. (2016), PLOS Genetics.

L370-372 How would be the comparison considering the mass of bacteria and fungi?

L394-395 "whereas complex lignocellulose substrates

395 enriched for many functionally redundant lignocellulolytic bacteria" please clarify what you mean with an example.

L412-414 You did not include in your analysis the MAGs identified by Stewart et al. (2019) Nature Biotechnology.

.

Reviewer #2 (Remarks to the Author):

The authors present a culture-independent analysis of gut microbes that have been enriched from goat feces. They use deep metagenomic sequencing and a standard bioinformatic workflow to recover high-quality metagenome assembled genomes. What makes this work particularly stand out is the authors use of selective enrichments to increase the concentration of anaerobic fungi to levels that make them amenable to genome reconstruction. In the process they report the first genomes sequences representative for as-yet uncultured fungi. This work would have really benefited from some functional information (such as transcriptomes/proteomes) to visualize the path of these carbon substrates in the various culture conditions. Overall this is a well written and presented study, I have the following comments to consider:

1. Title: For me, the novelty of this work lies heavily in the genome recovery of the as-yet uncultured

Fungi. I think the title should perhaps better reflect that.

2. Abstract: When reading the main text, it is not clear to me where the 400+ enrichments come from, is this calculated via the different substrates x different chemical conditions x different generations? In any case, I do not think it is overly important. Again, for me the impact of this paper is Fungal MAGs! On this point, it was not clear to me if these MAGs were found across the different fecal samples used to create the enrichments. Is there redundancy of these populations or is there individual variation?

3. Line 117: I am curious as to whether the eukMAGs were recovered using the same binning process as what was used for the prokaryotic MAGs? In the methods it appears as if a standard approach was used across both domains and that eukaryotic-specific binning tools were not used i.e. EukRep? If that is the case, I am curious in the authors opinion why others have not come across fungal MAGs previously, is it simply a matter of their relative DNA levels in endogenous microbiomes not being of sufficient quantity?

4. Line 125: What is BUSCO? How was completeness for the eukMAGs estimated? I appreciate that this information is provided in the supp methods, but given that this is the first time that fungal MAGs have been reported, it might be prudent to include a bit more detail as to how they were constructed in the main text.

5. Line 182: It is a bit unclear what is meant by source microbiota? Do you the authors mean compared to the original fecal material? What does G0 represent?

6. Line 266: How does the diversity of the fungal MAGs match the amplicon-based analysis? Do the authors have an idea of what proportion of the fungal populations in the fecal samples or even their enrichments are represented by their MAGs? How much work remains to gain genomic sampling of the rest of the diversity?

7. Line 275: The role of lactate as an intermediate to butyrate (i.e. by lactate utilizers Megasphaera and others) has been demonstrated in low methane producing ruminants (see work in sheep by Greame Attwood and colleagues). It is interesting that lactate producers were detected in Antibiotic-free enrichments, but I do not see any evidence of lactate utilizers (such as Megasphaera). It was also interesting that in the antibiotic enrichments, the authors also observe lactate accumulation (line 291). Was lactate also detected in the antibiotic free enrichments?

8. Line 320: saccharolytic mechanisms are mentioned, but only cellulosomes. What about polysaccharide utilization loci in both gram negatives (i.e. Bacteroidetes: CAZymes in clusters with SusC/D) and positives (i.e. butyrovibrios: CAZymes in clusters with ABC transporters)

9. Line 335: Do the authors have any thoughts as to what the possible metabolic contributions of Erysipelotrichaceae and Ruminococcus in the Fungi dominated enrichments?

Reviewer #3 (Remarks to the Author):

The authors describe a culture enrichment procedure to enrich for different microbial components from goat faecal samples. They then construct MAGs from the sequenced metagenomes and use the MAGs to assess the metabolic cross-feeding occurring in the goat intestinal microbiota. HPLC is used to determine the metabolic products from the enrichment cultures.

The authors have used well-designed culture enrichment strategies to profile the metabolic capabilities of bacteria, archaea and fungi and have generated considerable amount of interesting data. They highlight taxonomic novelty in the goat intestinal microbiota and show how bacteria, fungi and archaea can be separated to study the biology of each component separately.

However, it is currently difficult to assess the validity of the results as there are no statistics, some of the figures are either far too long or not concise in message and the results are not presented in a clear sequential order in the text. It is also not clear if fungi produce more methane than bacteria as stated in the text as Figures 4 and 5 suggest the opposite.

Major comments :

1. The results are not presented in a clear sequential order. This distracts from the interesting results presented. For example, the results jumps straight into Figure 2 and the MAGs. However, the process to generate the different microbial consortia in Figure 1 is only briefly mentioned before this (line 59) and is discussed later in two separate sections (line 97-105 and 171-178) in more detail which disrupts the flow of the paper. The experiments from Figure 1 to enrich for different members of the goat microbiota are clever and interesting and underpin the rest of the paper so they should be incorporated into the results section together at the start and described in more detail before discussing MAGs.
2. What was the relative abundance of the MAGS in the entire metagenome? What is the relative abundance of the fungi? These are important questions that (as far as I can tell) are not answered.
3. It is difficult to assess the quality of the fungal MAGs. There is no description in the main methods as to how the fungal genome analysis was done including the BUSCO analysis to determine fungal MAG quality. The authors say the eukMAGS belong to the *Neocallimastix* genus but do not describe how this was determined in the main text. The genome of *N. californiae* is 193 Mb long (GCA_002104975.1) but all of the eukMAGs (apart from the largest) are <100Mb so are less than half the size of *N. californiae* suggesting very incomplete assemblies which would be unsuitable for further analysis. If the authors map reads from the eukMAGs to *N. californiae*, what % of the *N. californiae* genome is covered? To include eukMAGs in subsequent analysis I would expect at least 60% of the *N. californiae* genome to be aligned. Similarly, it is difficult to gauge the quality of the genomes based on BUSCO marker genes as there is no indication what % of these genes a good assembly would contain.
4. Some of the supplemental figures are too long and difficult to understand. For example Supplemental Figures 3 (described as Supplementary Figure 5 in text) is 501 pages long, Supp. Figure 6 is 22 pages long, Supp. Figure 7 is 9 pages long, Supp. Figure 9 is 10 pages long and Supp. Figure 10 is 33 pages long. This distracts from the message, for example the authors discuss how 'rare' MAGs have a different metabolic capacity but this cannot be determined from Supplementary Figure 9.
5. Line 171-245 describes the strategies to enrich for different members of the microbiota but it is not clear which members of the microbiota were enriched and by what method. As a result, it is not clear if each enrichment strategy worked. Supplementary Figure 6 could be made much more compact and concise and even moved to the main figures as it describes an important part of the study.
6. Can the authors please expand the text related to Figures 3, 4 and 5. It is not clear how the analysis was done, what are the main results and what the take-home messages is. For example, Figure 4 title states day 0-3 but the legend describes three different generations. Is the data in Figure 5 derived from Figure 4? If so, it should be made clear why this was done or else combine the figures.
7. One of the main results the author describe (in the abstract, line 71-73, line 247, line 307) is fungi producing more methane than bacteria. However, Figures 4 and 5 show the fungi and archaea consortium (PS) producing methane but not the fungal consortium alone (CM). In fact, Figure 4 shows the bacterial component (antibiotic-free) producing more methane than the fungi in G10.
8. There are no statistics in the paper, therefore it is difficult to assess the validity of some of the results.

Minor:

Title- Use of word 'sculpting' is ambiguous. Title could include terms 'targeted enrichment' or something similar to be more descriptive

The abstract and introduction gives the impression that these microbes were cultured. However no isolation of pure cultures took place. The text should be modified accordingly. Also (line 22) states rumen bacteria are difficult to culture but this study does not examine rumen bacteria, please rephrase.

Line 28- please rephrase, '95% of which were previously uncultured' - the MAGs were not cultured. Also, this comparison includes human gut bacteria so it is incorrect to say 95% were unique in herbivore digestive tract.

Line 55 and line 76- what does 'tune' mean? Please be more specific

Line 83- why were these cut-offs chosen?

Line 85- how did you compare these? Please provide more details, perhaps in a Supplementary Table. Supplementary Table 3 is referenced but does not compare against other data-sets. Please provide the range for completeness and contamination along with the average. Also, studies generating MAGs from human gut datasets have been considerably larger and used higher thresholds for completeness and contamination (>90% completeness <5% contamination) so it is incorrect to say the collection described here is among the largest and most complete of all anaerobic microbiomes.

Line 87- No mention of supplementary table 1 and 2

Line 93- It is difficult to assess the novelty of the MAGs in this study using Figure 2 or Supplementary Figures 3-5. The authors make comparisons including human derived genomes and rumen genomes which will inflate the novelty without giving an insight to the novelty compared to studies from faeces of other herbivores. A comparison against just herbivores (faeces) should be done. Regarding Figure 2: it is not clear where the 3237 reference genomes are derived from. To properly assess the novelty and the diversity of the MAGs the tree would be more informative presenting just genomes from studies of microbiomes from other herbivores. The novel and non-novel MAGS from this study should be highlighted on the tree.

The taxonomy of the tree in the outer ring includes different taxonomic levels but this is not described in the legend. Either describe these taxonomic descriptions in the legend or consider presenting at just family level to be more consistent. Please give details in the main methods describing how this tree was constructed.

What is Supplementary Figure 2 representing? Is the legend for this described as Supplementary Figure 4 in text? The figure didn't load properly, the top of the phylogeny is missing. I am not sure what this figure adds to the story, consider removing or presenting in a different format.

Supplementary Figure 3 is 501 pages long and is described as Supplementary Figure 5 in text. I am not sure what this figure adds to the story, consider removing or presenting in a different format.

Line 199- please expand and provide context for reader

Line 208- this should be presented in a figure.

Line 248-252-Please re-phrase, meaning is not clear. Also cannot tell metabolic potential or rare taxa from Supp Figure 9.

Line 309- What was the rationale for cryopreservation?

Reviewer #4 (Remarks to the Author):

This is an interesting paper that combines the metagenome exploration and metabolomics to study the microbial consortia containing bacteria, archaea and fungi in terms of composition as well as function. Moreover, the paper uses cultivation approaches to derive stable microbial consortia that transform various types of plant lignocellulose and may be of importance for biotechnology. Undoubtedly, the paper increases our understanding of the hindgut microbiome function that was so far much less explored compared to the rumen microbiome. In my opinion, the authors made a very good job in addressing the structure and function of all domains of life and did their best to show metabolic dependencies between their members. Still, I have a number of points that should be addressed. First, I suggest that the authors should more clearly justify why they have chosen a study system using an animal that is not at all common (San Clemente Island Goat). Furthermore, I lack more clear discussion of the limitations of exploring hindgut by analyzing goat feces. I would recommend to sample the hindgut itself (if possible) and to compare its composition and function to the feces. Also, the authors should clarify why RNA was isolated (as stated in methods) but not utilized, for example by metatranscriptome sequencing. Would it still be possible to add the metatranscriptome analysis? Also, the barcoding methods utilized by the authors seem to be suboptimal (see below). Why not to use total RNA sequencing and rRNA classification to obtain a better view of the eukaryotic community composition and the prokaryote/eukaryote ratios? This would be a worthwhile addition. From the formal view, I believe that the paper requires some reduction since several points are mentioned repeatedly. The Discussion largely re-iterates the results, having only 6 references (!) in the text. I believe that combining the Discussion with Results would do a much better job and decrease the level of repetition.

Specific comments:

Line 80: "cultured" should be "uncultured"

Line 127: the level of contamination of the eukMAGs should be noted briefly in the text and in detail in the Supplementary Table. How does the (in)completeness of eukMAGs affect the results?

Line 189: There is some trouble with the citations to references

Line 414-417: Indeed, the differences of the rumen and fecal consortia are interesting, but the authors should clarify how fecal microbiomes correspond to hindgut ones

Line 425-427: I agree that the biotechnological use of the consortia is one of the interesting applications of this research. However, the authors should be more clear about the complexity of the hindgut microbiomes where subculturing and enrichments did not provide the best possible model system due to forced selection of certain taxa. This paper is rather about simplified (nonnatural) consortia and their (novel) members, but not so much about hindgut function. If the latter is the goal, metatranscriptomics/metabolomics of the real hindgut samples would perhaps be the way to go.

Line 514-514: References (2) and (4) are the same.

Supplementary Figure 6-I: 18S sequencing here revealed many plant taxa (the orders Fagales, Myrtales, Poales, Rosales and Zingiberales are plants). It should be made clear what is the reason. Most probably, fecal pellets still contained DNA from plant material, but then the title "Microbial community composition" is misleading.

Supplementary Figure 6-J: The same is visible here, plant orders are present. But how is it possible that Fagales DNA of plant origin is also present in consortia on xylan (I suppose DNA-free chemical)?

Legends of some Supplementary Figures are too small to read.

Supplementary Figure 9-K: Are the bacterial taxa *Stap* and *Ery* really unable to utilize glucose and other hexoses? This is difficult to believe.

Supplementary Figure 13 mention "Green dotted patterns" but there are none in the figure.

Online methods:

The authors mention RNA extraction but not its subsequent analysis and its utilization. RNA can be highly valuable for analyzing the composition of the microbiome and tracking activity, so its use should be better explained and, if possible, additional data provided (see above).

ITS2 sequencing: the authors mention that merged reads <360 bases were discarded, but there are many fungi with short ITS2 (as short as 40 bases). This for sure needs to be re-done. It is best to run the whole analysis and use ITS extraction afterwards, only excluding ITS2 sequences below 40 bases. Primer specificity, choice and biases need to be discussed here. It is difficult to believe that there are 143 species-level fungal taxa and only 24 species-level eukaryotes (that include fungi). The authors should mention that SSU does not have discrimination power in certain fungi where whole families may have exactly the same sequence, so any mentions of "species or genus" level divisions should be removed. These are simply similarity clusters.

The authors should provide the analysis of all MG reads and specifically mention what was the identity of those not mapping to MAGs and eukMAGs.

AA10 is missing among cellulolytic CAZymes; GH5 is mentioned for both cellulose and hemicellulose: this should be more clear. Groups within the GH5 family have different functions and the authors should list those cellulolytic ones and hemicellulolytic ones that they considered. For sure, whole GH5 can neither be cellulolytic or hemicellulolytic.

Author Rebuttal to Initial comments

1. General comments:

The experimental design, the metagenomics analysis (except for not including the latest MAGs of the publication of Stewart et al. (2019) *Nature Biotechnology*), the enrichment analysis, etc. are very well carried out and the results provide novel insight particularly into the fungal community and its connectivity to the bacteria community. *However, the abstract sounds like that only fungi would be needed for efficient lignocellulose degradation. But even your results indicate that fungi do not contain all important glycoside hydrolase and therefore a combination of bacteria and fungal community would probably be the most efficient way to produce methane from lignocellulose. **Therefore, the authors should investigate whether their data provides an opportunity to suggest an optimum combination of specific bacteria and fungi to maximize the production of methane.*** In addition the following specific comments have to be addressed:

Thank you for the comments, and we very much appreciate the reviewer's insights. We do not intend to suggest that only fungi would be needed for efficient lignocellulose degradation and have revised the abstract to avoid any confusion. The partially complementary sets of CAZymes possessed by bacteria and fungi certainly suggest that a combination of these communities will be the more efficient in biomass

degradation than using only bacteria or fungi. However, our data indicates that bacteria channels a significant portion of the fermentation products towards propionate and butyrate, which cannot be used for methanogenesis by the gut microbial community. In contrast, anaerobic fungi do not produce propionate and butyrate, and provide more substrate (e.g. H₂ and formate) available for methanogenesis. Therefore, we suggest a microbial community dominated by (or at least supplemented with) anaerobic fungi would maximize methane production.

Additionally, we have in fact included the latest MAGs published in Stewart et al, (2019), as detailed in the comments below. We agree with the reviewer that this is an extremely important dataset to benchmark our MAGs from this study against.

2. Specific comments:

Line Comment

L99-100 Could you provide a reference about the amount of microbes located in the hindgut often originate from and were active in the rumen, because many are digested and therefore probably not in the feces. Is the composition of the rumen microbiota not very different from the feces microbiota?

Studies investigating the microbial diversity in different parts of the digestive tract of ruminants have found that the rumen microbiota are quite different from the fecal microbiota. In most cases, the main difference is that the relative abundance of *Bacteroidetes* is higher in the rumen than in the hindgut or feces. In the initial version of the manuscript we already referenced two studies comparing the rumen and fecal microbiota in this way (Kim and Wells 2016; Godoy-Vitorino et al. 2012).

L114 Please correct to Verrucomicrobia.

Thank you, we have made this correction.

Figure 3 In this figure, methonogenic archaea seem to use formate as main substrates to produce methane. Please note that Grennings et al. (2019) found that formate is not the main substrate for rumen methanogenesis.

In Figure 3 the lines between a microbe and a substrate indicate the presence of the corresponding metabolic pathway for the microbe to consume or produce the substrate. The thickness of the lines represents the relative abundance of the corresponding microbes. We do not intend to suggest that formate was the main substrate for methane production in our experiments, simply that this metabolic route is possible. We have included a reference to Greening et al. (2019) in the main text (lines ~ 387-389).

L153-155 This comparison does result in a bias prediction because there are probably many prokaryotes, which are not or in a minor extent involved in fermentation of lignocellulose. Are you including those MAGs having no lignocellulose-active genes in your calculation of the average MAG? It would be more interesting to compare the number of lignocellulose-active genes in the top three eukMAG with Ruminococcus, Butyrivibrio and Prevotella. Moreover, it is questionable to determine hydrolytic potential of Neocallimastigomycota by the number of lignocellulose-active genes. Of more interest would be the efficiency of these lignocellulose-active genes to provide substrates for the methanogenic archaea to produce methane.

Thank you for pointing this out. Of all 719 MAGs, only 27 of them (<4%) do not have a lignocellulose-active gene, and 14 of these 27 MAGs are methanogenic archaea, so we think the comparison made here is grounded. We do agree with the reviewer that a comparison of the number of lignocellulose-active genes in MAGs from the genera *Ruminococcus*, *Butyrivibrio* and *Prevotella*, and eukMAGs from *Neocallimastigomycota* is of interest. Hence, we have now included an additional supplementary figure (Supplementary Figure 7; also see below), which shows that anaerobic fungi possess up to an order of magnitude higher number of lignocellulose-active genes than the key lignocellulolytic bacteria. We also note that this finding is consistent with previously published work (Solomon, et al, *Science*, 2016; Haitjema et al, *Nature Microbiology*, 2017) that showed that anaerobic fungi have an extremely high amount of CAZymes in their transcriptomes/genomes compared to higher fungi and anaerobic bacteria.

We agree that these numbers alone cannot be the basis of strong inferences and have included experimental data such as those in Figure 6, which demonstrate that as a consortium, bacteria and anaerobic fungi are similar in their potential to degrade plant biomass, while anaerobic fungi can degrade about twice as much cellulose paper as the bacterial consortium. It is difficult to assess the efficiency of these lignocellulose-active genes to provide substrates for the methanogenic archaea to produce methane because the products of the reactions they catalyze are monomeric and oligomeric sugars, which cannot be used by methanogenic archaea as substrates for methanogenesis. Fermentation of the sugars that produces H₂ or/and formate is carried out by *Neocallimastigomycota* in antibiotics-treated consortia and a variety of bacteria in antibiotics-free consortia. The efficiency of these two types of consortia in providing substrates for methanogenic archaea is indeed one of the focal points of this study and discussed in details in lines ~383-395.

Supplementary Figure 7. Cellulase, hemicellulase, and pectinase/esterase counts in MAGs from the genera *Prevotella*, *Butyrivibrio*, and *Ruminococcus*, and eukMAGs from the genus *Neocallimastix*.

L184-186 Are the *in vitro* conditions not suitable for the *Ruminococcus* that its relative abundance was similar to the source microbiome? Is this an indication that *in vitro* studies do not resemble *in vivo* studies in animals?

This is an interesting point. On average <3% of the source microbiota were cultivated. The fact that *Ruminococcus* was one of the cultivated lineages indicates that the *in vitro* conditions were suitable for them. On the other hand, *in vitro* studies typically do not preserve the full microbial diversity observed *in vivo*, so it is expected that there are differences between these two types of studies.

The culture medium used in our study was developed to mimic *in vivo* conditions to the best of our ability, which enabled us to successfully enrich for and isolate many anaerobes from herbivores' gut that have evaded prior characterization. Nevertheless, as far as we know, no culture medium can perfectly reproduce *in vivo* conditions, and any culture medium has its own bias for or against different microbial lineages. We are aware of the difficulty to mimic many *in vivo* conditions such as gastrointestinal movement and intermittent feeding, and we appreciate the contribution by *in vivo* studies to the field.

L208-209 This could be written more clearly in order to indicate that these microbes were available in the

source microbiome but at a level that was not detectable.

Thank you for the suggestion. We have edited the text here (see lines 218-219 in the updated manuscript).

L300-302 What kind of functional evidence you have that PS-resistant Firmicutes preventing catabolite repression of anaerobic fungi?

These Firmicutes possess metabolic pathways to use hexoses and pentoses which are fungal hydrolytic products (symbols “9”, “10”, and “11” in Figure 3).

L370-372 This is well known from animal trials comparing forage and concentrate diets, e.g. Roehe et al. (2016), PLOS Genetics.

Thanks for pointing out this study. The findings by Roehe et al. (2016) are certainly highly relevant to our study and we have now included a reference to this work (lines 387-389). Both our manuscript and Roehe et al. (2016) report that higher propionate formation is responsible for reduced methane production.

However, the different metabolic profile reported in Roehe et al. (2016) was a result of different diet acting on the same microbial population, while our experiments highlight the different metabolic profile as a result of different microbial population (with or without antibiotics) acting on the same substrate. Therefore, we feel that our findings are a novel contribution to understanding the roles played by different parts of the gut microbiota.

L370-372 How would be the comparison considering the mass of bacteria and fungi?

We believe that the reviewer is referring to the statement in lines 387 – 390 in the initial version of the manuscript: “These findings highlight the disproportionately large role that rare microbes such as anaerobic fungi play in lignocellulosic biomass degradation despite orders of magnitude lower abundances of fungi than of gut bacteria in the rumen”. In short, this statement holds even when the comparison is made considering the mass of bacteria and fungi.

Assuming the radius of a typical anaerobic fungal cell is about 10 times that of a typical rumen bacterium, the volume of a typical anaerobic fungal cell is then about 1,000 times that of a typical rumen bacterium. A precise comparison between the density of fungal and bacterial cells is not well established, but it is safe to assume that the density of fungal cells is not more than 10 times that of bacterial cells. Therefore, the mass of a fungal cell should be no more than 10,000 times that of a bacterial cell. Because the number of bacterial cells in the rumen are about six to seven orders of magnitude higher than that of fungal cells (Lee et al. 2000), the mass of bacterial cells are at least two to three orders of magnitude higher than that of fungal cells.

L394-395 “whereas complex lignocellulose substrates enriched for many functionally redundant lignocellulolytic bacteria” please clarify what you mean with an example.

Thank you. We have included examples illustrating for this argument in the section under the subheading “A high degree of functional redundancy is observed in antibiotics-free consortia” (lines 284-302).

L412-414 *You did not include in your analysis the MAGs identified by Stewart et al. (2019) Nature Biotechnology.*

Thank you for the comment, but we would like to point out that we did in fact include the MAGs identified by Stewart et al. (2019) Nature Biotechnology in the dereplication analysis to identify novel lineages found in our study. The comparisons were shown in Supplementary Figures 4 and 5 in the initial version of the manuscript, which were correctly referenced in the main text and indexed in the supplementary material, but the two corresponding pdf files were mislabeled as “Supplementary Figures 2 and 3”. Considering suggestions from Reviewer #3, we have replaced these two supplementary figures that are difficult to read with two supplementary datasets containing the same information but more accessible for readers to explore (Supplementary Data 2 and 3). The Stewart et al. (2019) paper was also referenced (reference #6 at line 85 in the initial version of the manuscript). Nevertheless, we do recognize that the Stewart et al. (2019) MAGs were not included in Supplementary Figure 3, and we have updated that figure to include all 4941 MAGs from Stewart et al. (2019).

Reviewer #2 (Remarks to the Author):

The authors present a culture-independent analysis of gut microbes that have been enriched from goat feces. They use deep metagenomic sequencing and a standard bioinformatic workflow to recover high-quality metagenome assembled genomes. What makes this work particularly stand out is the authors use of selective enrichments to increase the concentration of anaerobic fungi to levels that make them

amendable to genome reconstruction. In the process they report the first genomes sequences representative for as-yet uncultured fungi. This work would have really benefited from some functional information (such as transcriptomes/proteomes) to visualize the path of these carbon substrates in the various culture conditions. Overall this is a well written and presented study, I have the following comments to consider:

1. Title: For me, the novelty of this work lies heavily in the genome recovery of the as-yet uncultured Fungi. I think the title should perhaps better reflect that.

We agree that the recovery and metagenomic characterization of fungi is a key contribution of this work. And while we appreciate the reviewer's suggestion very much, the scale and scope of this work prevents us from adequately capturing all aspects of the study in the manuscript title. For this reason, we intend to keep the original title for the manuscript, as it emphasizes what we see as the largest "take home message" of the work. Importantly, the abstract does stress that we recovered anaerobic fungi as part of this study, and should be a suitable avenue to communicate the content of the paper to potential readers. That said, in light of Reviewer #3's suggestion, we have edited the title of the manuscript to be "Targeted enrichment of gut microbial communities alters fermentation products and methane release."

2. Abstract: When reading the main text, it is not clear to me where the 400+ enrichments come from, is this calculated via the different substrates x different chemical conditions x different generations? In any case, I do not think it is overly important. Again, for me the impact of this paper is Fungal MAGs! On this point, it was not clear to me if these MAGs were found across the different fecal samples used to create the enrichments. Is there redundancy of these populations or is there individual variation?

Yes, the 400+ enrichments were from the different substrates (4) x different antibiotics treatments (3) x different generations (11) x replicates (3) (= 396), as well as those enrichments (> 50) used in follow-up experiments examining the long-term stability of the enriched cultures.

We appreciate the interest in the fungal MAGs and the reviewer's suggestion to clarify the redundancy of our eukMAGs. We would like to clarify that the fecal pellets used for enrichment culturing were all from the same goat (Figure 1), and the microbial community composition was remarkably similar between individual pellets as determined by 16S, 18S, and ITS amplicon sequencing (Supplementary Figure 11). As described in Figure 1, we encouraged the cultivation of different taxa with the addition of antibiotics (chloramphenicol, or pen/strep) – in general the cultures with these antibiotic treatments enabled the cultivation of fungi, and therefore the sequencing and assembly of fungal MAGs (lines 373-375).

The ITS sequencing results showed that anaerobic fungi from three genera were present in the fecal pellets (Supplementary Figure 11; Supplementary Table 7), and we have recovered eukMAG(s) from each of the genus: 12 from *Neocallimastix*, 5 from *Piriomyces*, and 1 from *Caecomyces*. (Note that given comments from Reviewer #3, we excluded two *Neocallimastix* eukMAGs that had an alignment coverage of < 80% to the genome of *Neocallimastix californiae*.) Genome-wide average nucleotide identity comparison between the eukMAGs showed that all twelve *Neocallimastix* eukMAGs were >99% similar to each other and all five *Piriomyces* eukMAGs were >99% similar to each other, indicating that they each represent one species (Supplementary Figure 4 and see below).

Supplementary Figure 4. A) Genome-wide average nucleotide identity (ANI) calculated by MASH(Ondov et al. 2016) between the 18 eukMAGs recovered in this study and six previously sequenced genomes from the phylum *Neocallimastigomycota*. B) Genome-wide ANI calculated by MUMmer(Kurtz et al. 2004) between the the *Caecomyces* eukMAG and the genome of *Caecomyces churrovis*. C) Genome-wide ANI calculated by MUMmer(Kurtz et al. 2004) between the five *Piriomyces* eukMAGs. D) Genome-wide ANI calculated by MUMmer(Kurtz et al. 2004) between the 12 *Neocallimastix* eukMAGs and the genome of *Neocallimastix*

californiae. The tool dRep(Olm et al. 2017) was used to implement the ANI algorithms and construct the figures.

3. Line 117: *I am curious as to whether the eukMAGs were recovered using the same binning process as what was used for the prokaryotic MAGs? In the methods it appears as if a standard approach was used across both domains and that eukaryotic-specific binning tools were not used i.e. EukRep? If that is the case, I am curious in the authors opinion why others have not come across fungal MAGs previously, is it simply a matter of their relative DNA levels in endogenous microbiomes not being of sufficient quantity?*

The eukMAGs were indeed recovered using the same binning process as what was used for the prokaryotic MAGs (based on coverage and tetranucleotide frequency). We were able to recover eukMAGs only from our enrichment cultures treated with antibiotics where anaerobic fungi constitute a major part of the microbial community. In fact, we were not able to recover any eukMAGs from the fecal pellets or the antibiotics-free consortia. Therefore, we think the low relative amount of eukaryotic DNA present in most microbiome studies made it extremely challenging to reconstruct eukMAGs, as the reviewer has pointed out.

We did test EukRep on our dataset but the results were very poor (nearly all false negatives in a sample dominated by anaerobic fungi), and this is not surprising because EukRep was built and trained on almost exclusively *Ascomycota* genomes (24 out of 27 fungal genomes used for training). The non-*Ascomycota* genomes used to train EukRep belong to Microsporidia and Basidiomycota, so EukRep is not appropriate for classifying communities dominated by Neocallimastigomycota that have distinct genomic properties compared to Dikarya (e.g. low GC content at ~15%; see additional details in (Wilken et al. 2020)).

4. Line 125: *What is BUSCO? How was completeness for the eukMAGs estimated? I appreciate that this information is provided in the supp methods, but given that this is the first time that fungal MAGs have been reported, it might be prudent to include a bit more detail as to how they were constructed in the main text.*

BUSCO represents “Benchmarking Universal Single-Copy Orthologs” and is widely used for assessing the quality of eukaryotic genomes (Waterhouse et al. 2018). The completeness for the eukMAGs were estimated by identifying the presence and absence of the near-universal single-copy orthologs specific to their lineage (fungi in our case). Additional information related to the quality assessment of eukMAGs has now been included in the main text (lines 129-132 and lines 694-696 in Methods).

5. Line 182: It is a bit unclear what is meant by source microbiota? Do you the authors mean compared to the original fecal material? What does G0 represent?

Yes, the original fecal material is the source microbiota. We have revised the text here (“source fecal microbiota”) for clarification. “G0” represents the initial batch of cultures inoculated by the fecal pellets (see Methods section).

6. Line 266: How does the diversity of the fungal MAGs match the amplicon-based analysis? Do the authors have an idea of what proportion of the fungal populations in the fecal samples or even their enrichments are represented by their MAGs? How much work remains to gain genomic sampling of the rest of the diversity?

ITS2 sequencing showed that the three genera *Neocallimastix*, *Piromyces*, and *Caecomyces* accounted for the majority (85%) of the fungal community (Supplementary Table 7). There were three *Neocallimastix* ASV_Clusters (at 97% similarity level) detected in the fecal pellets, and the more abundant two of them (ASV_Clusters 1 and 2) were enriched with antibiotics treatments at G5 and G10. Because all 12 *Neocallimastix* eukMAGs have > 99% ANI when compared to each other (Supplementary Figure 4; Supplementary Data 5), it is probable that the *Neocallimastix* eukMAGs we reconstructed represent just one of the three *Neocallimastix* ASV_Clusters from the fecal pellets.

Similarly, there were two *Piromyces* ASV_Clusters (at 97% similarity level) detected in the fecal pellets, and enrichment cultures at G0 enabled the reconstruction of five *Piromyces* eukMAGs that are likely the same species (>99% ANI). There were four *Caecomyces* ASV_Clusters (at 97% similarity level) detected in the fecal pellets, and one *Caecomyces* eukMAG was reconstructed.

Summary statistics show that the *Neocallimastix* eukMAGs correspond to an ASV_Cluster that accounts for ~5% of the ITS2 relative abundance in the fecal pellets (Supplementary Table 7). *Piromyces* eukMAGs correspond to an ASV_Cluster that accounts for 32% of the ITS2 relative abundance in fecal pellets. The *Caecomyces* eukMAG correspond to the ASV_Cluster that accounts for 26% of the ITS2 relative abundance in fecal pellets. Together our eukMAG collection represents over 60% of the fungal community (assessed by ITS2 relative abundance) in the fecal pellets.

Due to the variable nature of the ITS regions and the limitation of short-read sequencing, none of the eukMAGs we reconstructed included a full-length ITS2 region, so we are unable to match an ASV to a eukMAG. Nevertheless, our data indicate that the reconstruction of high-quality eukMAGs requires the organisms they represent to be one of the most abundant ones in the enrichment culture so that there would be sufficient amount of genetic material. Therefore, the fungal eukMAGs represent the most abundant ASV_Cluster within each genus. In order to sample the remainder of the genomic diversity of the anaerobic fungi, different culturing conditions are likely necessary so that lineages different from those enriched in our study can be enriched.

7. Line 275: The role of lactate as an intermediate to butyrate (i.e. by lactate utilizers Megasphaera and others) has been demonstrated in low methane producing ruminants (see work in sheep by Graeme Attwood and colleagues). It is interesting that lactate producers were detected in Antibiotic-free enrichments, but I do not see any evidence of lactate utilizers (such as Megasphaera). It was also interesting that in the antibiotic enrichments, the authors also observe lactate accumulation (line 291). Was lactate also detected in the antibiotic free enrichments?

Lactate is a known fermentation product from anaerobic fungi, so its accumulation in antibiotic-treated cultures was expected. Lactate was not detected in antibiotics-free consortia enriched on plant substrates. Lactate was a major metabolic product in antibiotics-free consortia enriched on xylan, primarily as a fermentation product from the dominant bacteria *Selenomonas*.

Megasphaera were not present in our samples, including the source fecal microbiota (Supplementary Data 7). This is likely a result of different bacterial community composition between the rumen and the feces. We have noted this difference and referenced Graeme Attwood's work (Kamke et al. 2016) at line 390.

8. Line320: saccharolytic mechanisms are mentioned, but only cellulosomes. What about polysaccharide utilization loci in both gram negatives (i.e. Bacteroidetes: CAZymes in clusters with SusC/D) and positives (i.e. butyrovibrios: CAZymes in clusters with ABC transporters)

Thank you for the suggestion. In the updated manuscript, we have included the identification and analysis of both polysaccharide utilization loci (PUL) and other types of CAZyme gene clusters (CGC) (see lines 348-354). In sum, among the MAGs enriched in antibiotics-free consortia, there were on average around eight CAZyme gene clusters (CGC) defined by the presence of at least one CAZyme gene, one transporter gene, and one transcription factor gene (Huang et al. 2018) (Supplementary

Table 11 and Data 3). All CGC that include ABC transporters have also been identified (Supplementary Data 3). Among the Bacteroidetes MAGs enriched in antibiotics-free consortia with plant substrates, there were on average 17 to 22.5 polysaccharide utilization loci (PUL) defined by the tandem presence of SusC and SusD genes (Terrapon et al. 2015; Rob D. Stewart et al. 2018) (Supplementary Table 11 and Data 4).

9. Line 335: Do the authors have any thoughts as to what the possible metabolic contributions of *Erysipelotrichaceae* and *Ruminococcus* in the Fungi dominated enrichments?

Erysipelotrichaceae were found in both penicillin & streptomycin-treated and chloramphenicol-treated cultures, and we think they might be active in consuming monomeric sugars and hence contribute to reducing catabolite repression (see lines ~315-318). One MAG (“Rum3”) from the family *Ruminococcaceae* was detected at very low abundance in the consortia AG10R1-PS and XG10R3-PS, and it may have a similar role as the *Erysipelotrichaceae*.

Reviewer #3 (Remarks to the Author):

The authors describe a culture enrichment procedure to enrich for different microbial components from goat faecal samples. They then construct MAGS from the sequenced metagenomes and use the MAGs to assess the metabolic cross-feeding occurring in the goat intestinal microbiota. HPLC is used to determine the metabolic products from the enrichment cultures.

The authors have used well-designed culture enrichment strategies to profile the metabolic capabilities of bacteria, archaea and fungi and have generated considerable amount of interesting data. They highlight taxonomic novelty in the goat intestinal microbiota and show how bacteria, fungi and archaea can be separated to study the biology of each component separately.

However, it is currently difficult to assess the validity of the results as there are no statistics, some of the figures are either far too long or not concise in message and the results are not presented in a clear sequential order in the text. It is also not clear if fungi produce more methane than bacteria as stated in the text as Figures 4 and 5 suggest the opposite.

Thank you for the comments and for the suggestion. We have included references to supplementary tables presenting statistical test results in all relevant sections of the main text. We would like to clarify that neither fungi nor bacteria produce methane, rather they partner with archaea (methanogens) to do so. When the enrichment cultures in our study were treated with chloramphenicol (CM) that kills

methanogenic archaea, no methane production was detected (as seen in Figures 4 and 5). It is the consortium including methanogenic archaea and fungi/bacteria that can produce methane due to the metabolic exchanges that occur between fungi/bacteria and methanogen partners in the microbial community. When the enrichment cultures were treated with penicillin and streptomycin (PS) which kills most bacteria and not archaea, the culture became dominated by a consortium of anaerobic fungi and methanogenic archaea, which produced more methane than any antibiotics-free cultures that include both bacteria and methanogenic archaea.

Major comments:

1. The results are not presented in a clear sequential order. This distracts from the interesting results presented. For example, the results jumps straight into Figure 2 and the MAGs. However, the process to generate the different microbial consortia in Figure 1 is only briefly mentioned before this (line 59) and is discussed later in two separate sections (line 97-105 and 171-178) in more detail which disrupts the flow of the paper. The experiments from Figure 1 to enrich for different members of the goat microbiota are clever and interesting and underpin the rest of the paper so they should be incorporated into the results section together at the start and described in more detail before discussing MAGs.

Thanks for the suggestion. We have included an additional panel to Figure 1 to demonstrate the outcome of the enrichment culture experiment as assessed by 16S- and ITS2-based community composition, and this is mentioned before presenting results related to MAGs (Figure 1B). We believe this makes the presentation of results more cohesive in the context of the text and figures.

2. What was the relative abundance of the MAGs in the entire metagenome? What is the relative abundance of the fungi? These are important questions that (as far as I can tell) are not answered.

Thank you for the suggestion. In the initial version of the manuscript there was a supplementary figure showing the relative abundance of MAGs and fungi, which we have now updated as Supplementary Figure 5 and referenced in the main text (lines ~134). The relative abundance of each MAG in each enrichment culture is presented in Supplementary Data 4. In most antibiotics-free consortia, the 719 MAGs accounted for > 80% of the metagenomic reads. In penicillin and streptomycin-treated consortia, the 719 MAGs accounted for 50 – 60% of the metagenomic reads, while ~5 – 25% of the metagenomics reads mapped to one of the three fungal genome/eukMAG representative of the taxa found in this study. In chloramphenicol-treated consortia, > 80% of the reads mapped to the genome of *Neocallimastix californiae* after the initial generation. The pellet samples harbor a greater diversity as the source microbiota and only ~40% of the reads from them were mapped to our collection of MAGs and fungal eukMAG/genome. Additionally, the relative abundance of prokaryotic MAGs and fungi is represented in Figure 3 by the thickness of the lines connecting MAGs/fungi with substrates/products.

3. It is difficult to assess the quality of the fungal MAGs. There is no description in the main methods as to how the fungal genome analysis was done including the BUSCO analysis to determine fungal MAG quality. The authors say the eukMAGs belong to the *Neocallimastix* genus but do not describe how this was determined in the main text. The genome of *N. californiae* is 193 Mb long (GCA_002104975.1) but all of the eukMAGs (apart from the largest) are <100Mb so are less than half the size of *N. californiae* suggesting very incomplete assemblies which would be unsuitable for further analysis. If the authors map reads from the eukMAGs to *N. californiae*, what % of the *N. californiae* genome is covered? To include eukMAGs in subsequent analysis I would expect at least 60% of the *N. californiae* genome to be aligned. Similarly, it is difficult to gauge the quality of the genomes based on BUSCO marker genes as there is no indication what % of these genes a good assembly would contain.

The eukMAGs' taxonomy was determined by performing whole-genome average nucleotide comparison with the isolated genomes of anaerobic fungi (Supplementary Figure 4) and constructing a phylogenomic tree based on 60 single-copy orthologs that were present in over 60% of the eukMAGs (Supplementary Figure 3 and see below). Additional descriptions are now included in the main methods (lines 690-697).

Supplementary Figure 3. Unrooted phylogenomic tree of 18 eukaryotic metagenome-assembled genomes (eukMAGs) and six genomes from *Neocallimastigomycota*. To reconstruct the phylogeny, 60 single-copy orthologs retrieved using BUSCO(Simão et al. 2015) (v3.0.2) from OrthoDB(Zdobnov et al. 2017) (v9.1) were individually aligned and subsequently concatenated before tree building using FastTree(Price, Dehal, and Arkin 2010) (v2.1.1). Interactive tree of life(Letunic and Bork 2016) (iTOL v5.5.1) was used to generate the visuals.

SH (Shimodaira-Hasegawa test resample size = 1000) values greater than 0.9 were marked with a blue circle on the corresponding branch. Light purple was used to highlight the genome of *Neocallimastix californiae* and eukMAGs from the same genus. Green was used to highlight the genome of *Piromyces finnis* and *Piromyces sp. E2* and eukMAGs from the same genus. Orange was used to highlight the genome of *Caecomyces churrovis* and the eukMAG from the same genus. Grey dashed lines were used to connect the tip of each leaf and its label.

We agree with the reviewer's suggestion that at least 60% of a eukMAGs should align to the genome of *N. californiae* for it to be considered suitable for further analysis. Twelve out of the 14 *Neocallimastix* eukMAGs were aligned at > 83% to *N. californiae* genome (using nucmer), with an average alignment coverage of 95% (Supplementary Data 5). The remaining two *Neocallimastix* eukMAGs had an alignment rate of 54% and 42% and are excluded from further analysis in the updated manuscript.

Indeed, the genome size of *N. californiae* (193 Mb) is more than twice the size of all *Neocallimastix* eukMAGs. However, this is because *N. californiae* has some elevated ploidy (diploid, hybrid, etc), and >40% of its gene models were duplicated. We can use BUSCO to gauge the quality of eukMAGs because the same analysis was performed on the genomes of isolated anaerobic fungi for comparisons (Supplementary Data 5).

4. Some of the supplemental figures are too long and difficult to understand. For example Supplemental Figures 3 (described as Supplementary Figure 5 in text) is 501 pages long, Supp. Figure 6 is 22 pages long, Supp. Figure 7 is 9 pages long, Supp. Figure 9 is 10 pages long and Supp. Figure 10 is 33 pages long. This distracts from the message, for example the authors discuss how 'rare' MAGs have a different metabolic capacity but this cannot be determined from Supplementary Figure 9.

Thank you for the suggestion. We have replaced the 501-page Supplementary Figure 5 (the separate pdf file was mislabeled as "Supplementary Figure 3") with a Supplementary dataset that is more accessible for readers to explore (Supplementary Data 2). However, for the other supplementary figures, it is infeasible to compress all the supplementary information without losing the important details due to the extensiveness of this study. For the readers to have a comprehensive view of the results from this study, we think that it is best to present the supplementary figures as is (except for the overly long Supplementary Figures 4 and 5 in the initial submission, which have been replaced by Supplementary Tables). The main figures alone support our major conclusions, and the supplementary material has been edited and improved after incorporating comments from all four reviewers.

Thanks for pointing out that in Supplementary Figure 9 it is difficult to distinguish which MAGs were “rare”. In the revised submission, we have highlighted the relatively abundant MAGs (>1% relative abundance) with red boxes and now it is straightforward to tell which MAGs are rare in each consortium (in Supplementary Figure 13 in the updated manuscript). We have also added explanations specifying the additional metabolic capacity provided by the rare MAGs in the caption of each sub-figures of Supplementary Figure 13.

5. Line 171-245 describes the strategies to enrich for different members of the microbiota but it is not clear which members of the microbiota were enriched and by what method. As a result, it is not clear if each enrichment strategy worked. Supplementary Figure 6 could be made much more compact and concise and even moved to the main figures as it describes an important part of the study.

Thanks for the suggestion. We agree that Supplementary Figure 6 (in the initial manuscript) describes an important part of the study, and in fact we invested a significant amount of time and resources to present the development of the microbial communities in all 36 enrichment cultures at six different generations. As mentioned above, it is infeasible to compress all the amplicon sequencing results without losing valuable details. Nevertheless, we decided to present part of the results by including the community composition analysis based on amplicons (16S and ITS2) as a second panel of Figure 1. The additional panel in Figure 1 clearly shows which part of the microbiota were enriched by what method. Additionally, Figure 3, Supplementary Table 6, and Supplementary Figure 13 shows which members of the microbiota were enriched under each culturing condition.

6. Can the authors please expand the text related to Figures 3, 4 and 5. It is not clear how the analysis was done, what are the main results and what the take-home messages is. For example, Figure 4 title states day 0-3 but the legend describes three different generations. Is the data in Figure 5 derived from Figure 4? If so, it should be made clear why this was done or else combine the figures.

Thanks for the suggestion. We have moved the details on how the analysis was performed from the Supplementary material to the main Methods section. We have included additional explanation for Figure 3 at lines 240-245. The main results for Figures 4 and 5 have been summarized in the sections “Methane production is elevated in fungi-dominated consortia” and “Streamlined fermentation in selective consortia lead to higher methane production”. One of the main take-home messages from these three figures is that consortia dominated by bacteria channel much of the fermentation product towards short-chain fatty acids and reduce the substrate availability for hydrogenotrophic methanogens, while consortia dominated by anaerobic fungi do not produce propionate and butyrate and instead produce a greater amount of methane

than consortia dominated by bacteria. This take-home message was stated and discussed in the main text at lines 255 – 257, 288-292, 300-302, and 332 – 334.

Figure 4 presents the net change in primary metabolic products from day 0 to day 3 in anaerobic consortia during the course of parallel enrichments. This “net change” was measured for each of the three generations presented in Figure 4, and the details has been included in Supplementary Figure 2 and the updated Methods section.

While Figure 4 does include the net cumulative production of CH₄ and H₂ from different consortia, it does not show the trend of CH₄ and H₂ production over the course of three days, which is presented in Figure 5. Moreover, the difference in CH₄ production between different consortia is also better illustrated in Figure 5 than in Figure 4, so we think it is necessary to include both figures.

7. One of the main results the author describe (in the abstract, line 71-73, line 247, line 307) is fungi producing more methane than bacteria. However, Figures 4 and 5 show the fungi and archaea consortium (PS) producing methane but not the fungal consortium alone (CM). In fact, Figure 4 shows the bacterial component (antibiotic-free) producing more methane than the fungi in G10.

As mentioned above, we would like to clarify that neither fungi nor bacteria produce methane on their own. It is the consortium that include both fungi or bacteria and methanogenic archaea that can produce methane. In such a consortium, fungi/bacteria are responsible for hydrolysis and fermentation, which produces substrates (H₂ and formate) for methanogenesis by archaea.

When the enrichment cultures were treated with chloramphenicol (CM) that kills methanogenic archaea, no methane production was detected (Figures 4 and 5). When the enrichment cultures were treated with penicillin and streptomycin (PS) which kills most bacteria and not archaea, the culture became dominated by a consortium of anaerobic fungi and methanogenic archaea, which produced more methane than any antibiotics-free cultures that include both bacteria and methanogenic archaea.

8. There are no statistics in the paper, therefore it is difficult to assess the validity of some of the results.

Thanks for the suggestion. We have included additional statistical analysis to validate our arguments in the manuscript. As a result, the following figures and tables have been updated or added with additional

statistical analysis: Supplementary Figures 6, 7, 8, 13, 16, and 17; Supplementary Tables 2, 4, 6, 7, 10, and 11.

Minor:

Title- Use of word 'sculpting' is ambiguous. Title could include terms 'targeted enrichment' or something similar to be more descriptive

Thank you for the suggestion. We have edited the title of the manuscript to be “Targeted enrichment of gut microbial communities alters fermentation products and methane release.”

The abstract and introduction gives the impression that these microbes were cultured. However no isolation of pure cultures took place. The text should be modified accordingly. Also (line 22) states rumen bacteria are difficult to culture but this study does not examine rumen bacteria, please re-phrase.

Our experiments used enrichment cultures. The abstract and the entire main text accurately reflect this by using the words “consortia” and “enrichment” throughout and no mention of pure cultures. We have changed the wording in the abstract (line 22) to use “gut bacteria” instead of “rumen bacteria” to minimize ambiguities.

Line 28- please rephrase, '95% of which were previously uncultured' - the MAGs were not cultured. Also, this comparison includes human gut bacteria so it is incorrect to say 95% were unique in herbivore digestive tract.

The clause “95% of which were previously uncultured” refers to the 719 metagenome-assembled genomes (MAGs). MAGs are by definition uncultured and we did not in anywhere in the text indicate that we have obtained pure cultures of these MAGs, so we do not think the statement here is inaccurate. In response to the reviewer’s request, we have generated an herbivore-only tree/comparison, which shows that 95% of our MAGs were unique in the herbivore digestive tract (Supplementary Data 3). When the comparison includes genomes from non-herbivore sources (i.e. human gut collection, Genome Encyclopedia of Bacteria and Archaea, and 221 additional reference genomes from the NCBI), 94% of our MAGs are still unique (Supplementary Data 2).

Line 55 and line 76- what does 'tune' mean? Please be more specific

With this terminology, we mean to suggest that we are selectively altering the membership of the microbial community.

Line 83- why were these cut-offs chosen?

These cut-offs were chosen to be consistent and hence comparable to the largest collections of MAGs from herbivores' digestive tracts (Robert D. Stewart et al. 2018; 2019).

Line 85- how did you compare these? Please provide more details, perhaps in a Supplementary Table.

Whole-genome average nucleotide identity (ANI) comparisons were made using the tool dRep (Olm et al. 2017). Specifically, the tool first performs primary clustering using a MinHash-based method Mash (Ondov et al. 2016). The average nucleotide identity and alignment coverage between each pair ("Query" and "Reference") of genomes and MAGs within a primary cluster were computed using the Joint Genome Institute's ANIcalculator. All MAGs from this study that has an ANI less than 96.5% compared to any of the reference genomes are considered novel at the species level (Varghese et al. 2015). These methodological details are now included in the main methods section and Supplementary Data 2 and 3 are included to show all comparisons.

Supplementary Table 3 is referenced but does not compare against other data-sets. Please provide the range for completeness and contamination along with the average. Also, studies generating MAGs from human gut datasets have been considerably larger and used higher thresholds for completeness and contamination (>90% completeness <5% contamination) so it is incorrect to say the collection described here is among the largest and most complete of all anaerobic microbiomes.

Thanks for the suggestion. We have now provided Supplementary Table 2 summarizing the range and average of the completeness of all MAGs, as well as those from the reference collections. We have revised the statement that this collection is among the largest and most complete (lines 83-85).

Line 87- No mention of supplementary table 1 and 2

In the initial version of the manuscript, Supplementary Tables 1 and 2 were referenced along with Figure 2. In the revised manuscript, Supplementary Table 1 is referenced along Figure 2 at line 83.

Line 93- It is difficult to assess the novelty of the MAGs in this study using Figure 2 or Supplementary

We performed whole-genome comparisons based on average nucleotide identity, and considered a MAG novel (representing a new species) if it is less than 96.5% similar to the reference genomes. Following the reviewer's suggestion below, we have constructed an additional phylogenomic tree highlighting the novel MAGs from this study (Supplementary Figures 1 and 2).

Figures 3-5. The authors make comparisons including human derived genomes and rumen genomes which will inflate the novelty without giving an insight to the novelty compared to studies from faeces of other herbivores. A comparison against just herbivores (faeces) should be done. Regarding Figure 2: it is not clear where the 3237 reference genomes are derived from. To properly assess the novelty and the diversity of the MAGs the tree would be more informative presenting just genomes from studies of microbiomes from other herbivores. The novel and non-novel MAGS from this study should be highlighted on the tree.

Thanks for the suggestions. We respectfully disagree that including human gut bacterial genomes as part of the comparison will inflate the novelty of our collection of MAGs. Upon the reviewer's request, we have performed two comparisons: one including all reference genomes as mentioned in the text originally (Supplementary Data 2), and the other one including reference genomes and MAGs from herbivores only (Supplementary Data 3). The comparisons show that when human gut bacterial genomes are included, the percentage of novel MAGs from our study (94.2%) is slightly lower than when human gut bacterial genomes and other reference genomes are excluded (95.1%). The second comparison we performed (Supplementary Data 3) includes all genomes from the Hungate Collection, many of which belong to bacteria cultivated from fecal sources. Additionally, we have generated a tree including all genomes and MAGs used in the second comparison, highlighting the novel and non-novel MAGs from this study (Supplementary Figure 2).

Regarding Figure 2: we have now included a description of where all the reference genomes are derived from in the figure caption and in Supplementary Table 2. Note that the source of the reference genomes was already described in the main text of the initially version of this manuscript.

The taxonomy of the tree in the outer ring includes different taxonomic levels but this is not described in the legend. Either describe these taxonomic descriptions in the legend or consider presenting at just family level to be more consistent. Please give details in the main methods describing how this tree was constructed.

Thanks for the suggestion. The taxonomy of the tree in the outer ring is described in the figure caption.

We have provided details in the main methods describing how the phylogenomic tree was constructed (lines 697-699).

What is Supplementary Figure 2 representing? Is the legend for this described as Supplementary Figure 4 in text? The figure didn't load properly, the top of the phylogeny is missing. I am not sure what this figure adds to the story, consider removing or presenting in a different format.

Supplementary Figure 3 is 501 pages long and is described as Supplementary Figure 5 in text. I am not sure what this figure adds to the story, consider removing or presenting in a different format.

The pdf file for Supplementary Figure 4 and 5 was mistakenly named as “Supplementary Figure 2” and “Supplementary Figure 3”, respectively. It was intended to show the results for primary clustering (at 90% average nucleotide identity by MASH) and the secondary clustering (at 96.5% average nucleotide identity by gANI) of all MAGs and genomes included for comparison (implemented by the tool dRep), so that we can determine which MAGs from our collection represent novel species. It is necessary to include this information because it supports our argument about the novelty of our 719 MAGs.

Given the reviewer's suggestion, we have replaced these two supplementary figures with Supplementary Data 2, which presents the same information but more accessible to navigate.

Line 199- please expand and provide context for reader

Thanks for the suggestion. We have included additional details here (now at lines 208-210).

Line 208- this should be presented in a figure.

This was presented in Supplementary Figures 6B and 6C. In the revised manuscript, we have now included a reference to Supplementary Figures 11B and 11C here.

Line 248-252-Please re-phrase, meaning is not clear. Also cannot tell metabolic potential or rare taxa from Supp Figure 9.

Thanks for the suggestion. We have rephrased this part and included an example where rare taxa harbor metabolic potentials that are unique compared to the more abundant taxa (now at lines 264-268). Metabolic potentials in Supplementary Figure 9 are represented by lines connecting MAGs and substrates/products. In the revised submission, we have highlighted the relatively abundant MAGs (>1% relative abundance) with red boxes and now it is straightforward to tell which MAGs are rare in each consortium. We have also added explanations specifying the additional metabolic capacity provided by the rare MAGs in the caption of each sub-figures of this supplementary figure (Supplementary Figure 14 in the revised manuscript).

Line 309- What was the rationale for cryopreservation?

We showed that some of the enriched consortia are stable after cryopreservation because this is of interest to the biotechnological community that may explore scaling up our experiments for industrial applications.

Reviewer #4 (Remarks to the Author):

This is an interesting paper that combines the metagenome exploration and metabolomics to study the microbial consortia containing bacteria, archaea and fungi in terms of composition as well as function. Moreover, the paper uses cultivation approaches to derive stable microbial consortia that transform various types of plant lignocellulose and may be of importance for biotechnology. Undoubtedly, the paper increases our understanding of the hindgut microbiome function that was so far much less explored compared to the rumen microbiome. In my opinion, the authors made a very good job in addressing the structure and function of all domains of life and did their best to show metabolic dependencies between their members. Still, I have a number of points that should be addressed.

First, I suggest that the authors should more clearly justify why they have chosen a study system using an animal that is not at all common (San Clemente Island Goat).

Goats from different species share a broad spectrum of diet to which grasses are central. The San Clemente Island Goat belongs to the species *Capra hircus* and its fresh feces are readily accessible for

collection at the local zoo. Importantly, we do expect that the methodology we present to selectively enrich for different microbial consortia would apply broadly across herbivores.

Furthermore, I lack more clear discussion of the limitations of exploring hindgut by analyzing goat feces. I would recommend to sample the hindgut itself (if possible) and to compare its composition and function to the feces.

As the reviewer pointed out in a comment below, this study is not about the functioning of hindgut microbiome in its native host environment. We are interested in comparing the metabolic potentials of the prokaryotic and eukaryotic parts of the microbiome in herbivore's digestive tract, and the least invasive way to sample is to collect freshly produced fecal pellets from herbivores. Hence for this study we do not consider it necessary to sample the hindgut itself, which typically requires procedures invasive (or killing) to animals.

Also, the authors should clarify why RNA was isolated (as stated in methods) but not utilized, for example by metatranscriptome sequencing. Would it still be possible to add the metatranscriptome analysis?

Indeed, metatranscriptome sequencing was part of our experimental design, and we performed extensive testing on practice samples to optimize our protocols. Unfortunately, the quality of all RNA from the samples where the DNA were co-extracted with the QIAGEN AllPrep kit was very low and none of them passed the quality control at the Joint Genome Institute. Therefore, we do not have metatranscriptome data available and we have clarified this at lines 588-589. Nonetheless, the experimental data included in the manuscript are very extensive and support all major conclusions.

Also, the barcoding methods utilized by the authors seem to be suboptimal (see below). Why not to use total RNA sequencing and rRNA classification to obtain a better view of the eukaryotic community composition and the prokaryote/eukaryote ratios? This would be a worthwhile addition.

Thanks for the advice on ITS2 amplicon analysis. We have re-analyzed the ITS2 dataset without excluding amplicons < 360 bp (instead using a 40 bp filter as suggested by the reviewer) and updated the figures. As mentioned above, we unfortunately do not have direct metatranscriptome data available to be included in this study.

From the formal view, I believe that the paper requires some reduction since several points are mentioned repeatedly. The Discussion largely re-iterates the results, having only 6 references (!) in the text. I believe that combining the Discussion with Results would do a much better job and decrease the level of repetition.

Thanks for the suggestion. We have streamlined both the Results section and Discussion section to minimize redundancy.

Specific comments:

Line 80: "cultured" should be "uncultured"

Corrected.

Line 127: the level of contamination of the eukMAGs should be noted briefly in the text and in detail in the Supplementary Table. How does the (in)completeness of eukMAGs affect the results?

It is difficult to assess the level of contamination of the eukMAGs, although the percentage of duplicated BUSCOs provides a very rough estimate. We have included at lines 131-132 the average percentage of duplicated BUSCOs among eukMAGs, and the details are reported in Supplementary Data 5. The main consequence of the (in)completeness of eukMAGs in this study is the potential underestimation of the number of CAZymes within each of them.

Line 189: There is some trouble with the citations to references

Thanks for pointing this out. This has been corrected.

Line 414-417: Indeed, the differences of the rumen and fecal consortia are interesting, but the authors should clarify how fecal microbiomes correspond to hindgut ones

Thanks for the suggestions. We have included a clarification at lines 432-433.

Line 425-427: I agree that the biotechnological use of the consortia is one of the interesting applications of this research. However, the authors should be more clear about the complexity of the hindgut microbiomes where subculturing and enrichments did not provide the best possible model system due to forced selection of certain taxa. This paper is rather about simplified (nonnatural) consortia and their (novel) members, but not so much about hindgut functioning. If the latter is the goal, metatranscriptomics/metabolomics of the real hindgut samples would perhaps be the way to go.

We agree with the reviewer that this study is not about the functioning of hindgut microbiome in its native host environment. We are interested in comparing the metabolic potentials of the prokaryotic and eukaryotic parts of the microbiome in herbivore's digestive tract, and the least invasive way to sample is to collect freshly produced fecal pellets from herbivores.

Line 514-514: References (2) and (4) are the same.

Thanks for pointing this out. This has been corrected.

Supplementary Figure 6-I: 18S sequencing here revealed many plant taxa (the orders Fagales, Myrtales, Poales, Rosales and Zingiberales are plants). It should be made clear what is the reason. Most probably, fecal pellets still contained DNA from plant material, but then the title "Microbial community composition" is misleading.

Thanks for pointing this out. Fecal pellets of ruminants include undigested plant material and microbial matter (Van Soest 1994). We have included this clarification in the caption of Supplementary Figure 6-I (Supplementary Figure 11-I in the revised manuscript) and removed "Microbial" from the caption title.

Supplementary Figure 6-J: The same is visible here, plant orders are present. But how is it possible that Fagales DNA of plant origin is also present in consortia on xylan (I suppose DNA-free chemical)?

Plant (e.g. *Fagales* and *Malpighiales*) DNA was present only in xylan samples at G0 (generation 0), in which there was a considerable amount of pellet material that contained residual plant. We have included this clarification in the caption of Figure 6-J and removed "Microbial" from the caption title.

Legends of some Supplementary Figures are too small to read.

Thanks for pointing this out. We have enlarged the legends of supplementary figures (i.e. Supplementary Figure 11) to improve legibility.

Supplementary Figure 9-K: Are the bacterial taxa Stap and Ery really unable to utilize glucose and other hexoses? This is difficult to believe.

Thanks for pointing this out. Both Stap and Ery1 can utilize hexoses and xylose (Supplementary Data 4), and the lines missing between them and hexoses and pentoses were a mistake. We have double checked all figures to make sure they are correct.

Supplementary Figure 13 mentione "Green dotted patterns" but there are none in the figure.

Thanks for pointing this out. The “green dotted patterns” referred to an earlier version of the figure. We have corrected it to “yellow bars” to be consistent with the current version of the figure.

Online methods:

The authors mention RNA extraction but not its subsequent analysis and its utilization. RNA can be highly valuable for analyzing the composition of the microbiome and tracking activity, so its use should be better explained and, if possible, additional data provided (see above).

Indeed, metatranscriptome sequencing was part of our experimental design, and we performed extensive testing on practice samples to optimize our protocols. Unfortunately, the quality of all RNA from the samples where the DNA were co-extracted with the QIAGEN AllPrep kit was very low and none of them passed the quality control at the Joint Genome Institute. Therefore, we do not have metatranscriptome data available and we have clarified this at lines 588-589. Nonetheless, the experimental data included in the manuscript are very extensive and support all major conclusions.

ITS2 sequencing: the authors mention that merged reads <360 bases were discarded, but there are many fungi with short ITS2 (as short as 40 bases). This for sure needs to be re-done. It is best to run the whole analysis and use ITS extraction afterwards, only excluding ITS2 sequences below 40 bases.

Thanks for the suggestion. We have rerun the ITS2 analysis and included all ITS2 sequences > 39 bases, and the corresponding section in the Extended Methods has been updated. However, we respectfully disagree with the reviewer on the details of the bioinformatics workflow. There are multiple sets of primer sets in the literature commonly used for sequencing the ITS2 region, and they all start somewhere in the 5.8S region and end in the 28S region. Additionally, the reference database (UNITE) include many full-length ITS sequences. To be consistent for comparison between different dataset, we used the tool ITSxpress (Rivers et al. 2018) to remove the ribosomal RNA regions flanking the ITS region, for both our experimental dataset and the UNITE reference database. This allowed for the most efficient denoising/clustering step using DADA2 and accurate taxonomic assignment to operational taxonomic units. If the ribosomal RNA regions flanking the ITS2 region were not removed before denoising/clustering, they could introduce additional variations independent of the ITS2 sequences.

Primer specificity, choice and biases need to be discussed here. It is difficult to believe that there are 143 species-level fungal taxa and only 24 species-level eukaryotes (that include fungi). The authors should mention that SSU does not have discrimination power in certain fungi where whole families may have exactly the same sequence, so any mentions of "species or genus" level divisions should be removed. These are simply similarity clusters.

Thanks for the suggestion. We agree that SSU does not have as much discrimination power in certain fungi compared to ITS, which explains why there are 143 amplicon sequence variant (ASV) clusters from ITS2 and fewer (24) ASV clusters from SSU. It should be noted that this was not explicitly compared in the main text. We also agree that the amplicon sequence variant (ASV) clusters we generated are simply similarity clusters. However, the ITS2 ASV clusters we generated can be unambiguously assigned to one of the three anaerobic fungal genera (*Neocallimastix*, *Piromyces*, and *Caecomycetes*), so we do not think it is necessary to remove mentions of genus level division. We did not attempt to assign species-level taxonomy to the ASV clusters, so there is no mention of species-level division to be removed.

The authors should provide the analysis of all MG reads and specifically mention what was the identity of those not mapping to MAGs and eukMAGs.

The analysis used to analyze the metagenomes have been described in detail in the updated Methods section. The reads that did not map to MAGs and eukMAGs belong to the part of the microbial community with low relative abundance for which no high-quality MAGs or eukMAGs could be reconstructed. The identity of these "rare" part of each sample could be inferred by the amplicon-based assays (16S and ITS2), which showed that they were largely from the same lineages (e.g. *Ruminococcaceae*, *Lachnospiraceae*, and *Bacteroidales*) as those with high-quality MAGs reconstructed (Supplementary Figure 11).

AA10 is missing among cellulolytic CAZymes; GH5 is mentioned for both cellulose and hemicellulose: this should be more clear. Groups within the GH5 family have different functions and the authors should list those cellulolytic ones and hemicellulolytic ones that they considered. For sure, whole GH5 can neither be cellulolytic or hemicellulolytic.

Thanks for pointing out that AA10 is a cellulolytic CAZyme. We did not include it in our list (Supplementary Table 3) because unlike the GH cellulases, it is a monooxygenase and requires oxygen to function. Therefore, AA10 are found primarily in aerobes and facultative anaerobes, and they are rarely reported from obligate anaerobes such as those from the class *Clostridia* and the phylum *Bacteroidetes*, which account for the vast majority of the microbiota in our study. Additionally, strictly anaerobic conditions are maintained in all enrichment cultures in our study, so AA10 is not relevant.

We agree that there remains much uncertainty in defining the functions of all the GH5 subfamilies. While some of the subfamilies are known to act on cellulose or hemicellulose (or both), the function of other subfamilies remains to be verified biochemically. Our study aims to highlight the different combinations of CAZymes used by consortia dominated by bacteria vs. fungi, and the related arguments and conclusions are supported by our analysis focusing at the family level of CAZymes. The main text has included clarification that GH families 5, 8, 44, and 51 are versatile in function (lines 146-148).

References

Godoy-Vitorino, Filipa, Katherine C. Goldfarb, Ulas Karaoz, Sara Leal, Maria A. Garcia-Amado, Philip Hugenholtz, Susannah G. Tringe, Eoin L. Brodie, and Maria Gloria Dominguez-Bello. 2012.

“Comparative Analyses of Foregut and Hindgut Bacterial Communities in Hoatzins and Cows.” *The ISME Journal* 6 (3): 531–41. <https://doi.org/10.1038/ismej.2011.131>.

Huang, Le, Han Zhang, Peizhi Wu, Sarah Entwistle, Xueqiong Li, Tanner Yohe, Haidong Yi, Zhenglu Yang, and Yanbin Yin. 2018. “DbCAN-Seq: A Database of Carbohydrate-Active Enzyme (CAZyme) Sequence and Annotation.” *Nucleic Acids Research* 46 (D1): D516–21.

<https://doi.org/10.1093/nar/gkx894>.

Kamke, Janine, Sandra Kittelmann, Priya Soni, Yang Li, Michael Tavendale, Siva Ganesh, Peter H. Janssen, et al. 2016. “Rumen Metagenome and Metatranscriptome Analyses of Low Methane Yield Sheep

Reveals a Sharpea-Enriched Microbiome Characterised by Lactic Acid Formation and Utilisation.” *Microbiome* 4 (1): 56. <https://doi.org/10.1186/s40168-016-0201-2>.

Kim, Minseok, and James. E. Wells. 2016. “A Meta-Analysis of Bacterial Diversity in the Feces of Cattle.” *Current Microbiology* 72 (2): 145–51. <https://doi.org/10.1007/s00284-015-0931-6>.

Kurtz, Stefan, Adam Phillippy, Arthur L. Delcher, Michael Smoot, Martin Shumway, Corina Antonescu, and Steven L. Salzberg. 2004. “Versatile and Open Software for Comparing Large Genomes.” *Genome Biology* 5 (2): R12. <https://doi.org/10.1186/gb-2004-5-2-r12>.

Letunic, Ivica, and Peer Bork. 2016. “Interactive Tree of Life (ITOL) v3: An Online Tool for the Display and Annotation of Phylogenetic and Other Trees.” *Nucleic Acids Research* 44 (W1): W242–45. <https://doi.org/10.1093/nar/gkw290>.

Olm, Matthew R., Christopher T. Brown, Brandon Brooks, and Jillian F. Banfield. 2017. “DRep: A Tool for Fast and Accurate Genomic Comparisons That Enables Improved Genome Recovery from Metagenomes through de-Replication.” *The ISME Journal* 11 (12): 2864–68. <https://doi.org/10.1038/ismej.2017.126>.

Ondov, Brian D., Todd J. Treangen, Páll Melsted, Adam B. Mallonee, Nicholas H. Bergman, Sergey Koren, and Adam M. Phillippy. 2016. “Mash: Fast Genome and Metagenome Distance Estimation Using MinHash.” *Genome Biology* 17 (1): 132. <https://doi.org/10.1186/s13059-016-0997-x>.

Price, Morgan N., Paramvir S. Dehal, and Adam P. Arkin. 2010. “FastTree 2 – Approximately Maximum-Likelihood Trees for Large Alignments.” *PLOS ONE* 5 (3): e9490. <https://doi.org/10.1371/journal.pone.0009490>.

Rivers, Adam R., Kyle C. Weber, Terrence G. Gardner, Shuang Liu, and Shalamar D. Armstrong. 2018. “ITSxpress: Software to Rapidly Trim Internally Transcribed Spacer Sequences with Quality Scores for Marker Gene Analysis.” *F1000Research* 7 (September): 1418. <https://doi.org/10.12688/f1000research.15704.1>.

Simão, Felipe A., Robert M. Waterhouse, Panagiotis Ioannidis, Evgenia V. Kriventseva, and Evgeny M. Zdobnov. 2015. “BUSCO: Assessing Genome Assembly and Annotation Completeness with Single-Copy Orthologs.” *Bioinformatics* 31 (19): 3210–12. <https://doi.org/10.1093/bioinformatics/btv351>.

Stewart, Rob D., Marc D. Auffret, Rainer Roehe, and Mick Watson. 2018. “Open Prediction of Polysaccharide Utilisation Loci (PUL) in 5414 Public Bacteroidetes Genomes Using PULpy.” *BioRxiv*, September, 421024. <https://doi.org/10.1101/421024>.

Stewart, Robert D., Marc D. Auffret, Amanda Warr, Alan W. Walker, Rainer Roehe, and Mick Watson. 2019. “Compendium of 4,941 Rumen Metagenome-Assembled Genomes for Rumen Microbiome Biology and Enzyme Discovery.” *Nature Biotechnology* 37 (8): 953–61. <https://doi.org/10.1038/s41587-019-0202-3>.

Stewart, Robert D., Marc D. Auffret, Amanda Warr, Andrew H. Wiser, Maximilian O. Press, Kyle W. Langford, Ivan Liachko, et al. 2018. “Assembly of 913 Microbial Genomes from Metagenomic Sequencing of the Cow Rumen.” *Nature Communications* 9 (1): 870. <https://doi.org/10.1038/s41467-018-03317-6>.

Terrapon, Nicolas, Vincent Lombard, Harry J. Gilbert, and Bernard Henrissat. 2015. “Automatic Prediction of Polysaccharide Utilization Loci in Bacteroidetes Species.” *Bioinformatics* 31 (5): 647–55. <https://doi.org/10.1093/bioinformatics/btu716>.

Van Soest, Peter J. 1994. *Nutritional Ecology of the Ruminant*. 2nd ed. Cornell University Press. <https://www.jstor.org/stable/10.7591/j.ctv5rf668>.

Varghese, Neha J., Supratim Mukherjee, Natalia Ivanova, Konstantinos T. Konstantinidis, Kostas Mavrommatis, Nikos C. Kyrpides, and Amrita Pati. 2015. “Microbial Species Delineation Using Whole Genome Sequences.” *Nucleic Acids Research* 43 (14): 6761–71. <https://doi.org/10.1093/nar/gkv657>.

Waterhouse, Robert M., Mathieu Seppey, Felipe A. Simão, Mosè Manni, Panagiotis Ioannidis, Guennadi Klioutchnikov, Evgenia V. Kriventseva, and Evgeny M. Zdobnov. 2018. “BUSCO Applications from Quality Assessments to Gene Prediction and Phylogenomics.” *Molecular Biology and Evolution* 35 (3): 543–48. <https://doi.org/10.1093/molbev/msx319>.

Wilken, St. Elmo, Susanna Seppälä, Thomas S. Lankiewicz, Mohan Saxena, John K. Henske, Asaf A. Salamov, Igor V. Grigoriev, and Michelle A. O’Malley. 2020. “Genomic and Proteomic Biases Inform Metabolic Engineering Strategies for Anaerobic Fungi.” *Metabolic Engineering Communications* 10 (June): e00107. <https://doi.org/10.1016/j.mec.2019.e00107>.

Zdobnov, Evgeny M., Fredrik Tegenfeldt, Dmitry Kuznetsov, Robert M. Waterhouse, Felipe A. Simão, Panagiotis Ioannidis, Mathieu Seppey, Alexis Loetscher, and Evgenia V. Kriventseva. 2017. “OrthoDB v9.1: Cataloging Evolutionary and Functional Annotations for Animal, Fungal, Plant, Archaeal, Bacterial and Viral Orthologs.” *Nucleic Acids Research* 45 (D1): D744–49. <https://doi.org/10.1093/nar/gkw1119>.

Decision Letter, first revision:

Dear Professor O’Malley,

Thank you for your patience while your manuscript "Targeted enrichment of gut microbial communities alters fermentation products and methane release" was under peer review at Nature Microbiology. It has now been seen by our referees, and in the light of their advice I am delighted to say that we can in principle offer to publish it. First, however, we would like you to revise your paper to address the points made by the reviewers, and to ensure that it is in Nature Microbiology format.

The referees’ remaining comments are clear, and should not be difficult to implement. Editorially, we will need you to make some changes so that the paper complies with our Guide to Authors at

<http://www.nature.com/nmicrobiol/info/gta>.

Please find attached a lightly-edited version of the manuscript that you should use to finalize the main text for resubmission. Also please consider submission of a cover image -- perhaps you have more photos of the goat from the zoo (we would need a high quality image if there is one available?)

Specific points:

In particular, while checking through the manuscript and associated files, we noticed the following specific points which we will need you to address:

1. Extended data. Per journal guidelines, we use "Extended Data". Please see below for additional information on how to format and refer to Extended Data. We can allow a maximum of 10 extended data figures and so we would suggest that you move the remaining 9 Extended Data Figures into supplementary figures. Extended data tables are not allowed, so you move them into Supplementary Information as Supplementary tables.
2. The main and extended data figures need to be uploaded individually and without captions; they should be provided in eps, tiff or jpeg format. The main text file needs to be submitted in Word or LaTeX format, and all (main and extended) figure legends need to be included at the end of this file. Do NOT include any figures in the main text file.
3. Figure legends need to be less than 375 words long. Figure legend titles should be on one line where possible, I've suggested some shortening.
4. Priority claims. Per journal guidelines, we recommend that you avoid the use of terms like 'new', 'novel' and other priority claims throughout the text in order to avoid any perception of grandstanding and so that the reader can focus on the significance, rather than the novelty, of the findings.
5. Title. We have suggested three alternate titles, each is designed to include as many keywords as possible that might be included in a pubmed/semantic scholar/google search. Please edit your favourite to suit, and note that words in the title should not be capitalized.
6. The competing interest statement needs to be included in the manuscript text (before or after the Acknowledgements).
7. Please provide a detailed and specific author contributions statement. A good example can be found at the end of the following article <http://www.nature.com/nature/journal/v532/n7599/full/nature17433.html> .
8. Molecular weight markers. Per style guidelines, MW markers must be displayed in all panels in which this information is relevant.
9. Full length blots. Per journal guidelines, we require that full length versions of all blots are included in SI. These should be raw, uncropped, unmanipulated versions of all gels, ideally showing the original molecular weight markers and using text boxes to indicate which sections of the full gels were cropped to generate the figures shown.

10. Source data. Please provide all numerical data behind the presented in vitro and in vivo experiments as source data. Please see below for additional information on how to format and refer to source data.

11. Main text at more than 5122 words is too long. Please trim to 4500 maximum.

12. Data availability. Please note that all accession codes provided must be live by the time of publication of the piece. Are the metagenome sequencing reads accessible to all at the Joint Genome Institute under JGI 828 Project ID's listed in Supplementary Data 13? I think we usually mandate an approved repository such as the SRA. Please let me know if you can also submit the sequences there.

General points:

We will also need you to check through all of the following general points when preparing the final version of your manuscript:

13. Choosing the right electronic format for your figures at this stage will speed up the processing of your paper. We would like the figures to be supplied as vector files - EPS, PDF, AI or postscript (PS) file formats (not raster or bitmap files), preferably generated with vector-graphics software (Adobe Illustrator for example). Please try to ensure that all figures are non-flattened and fully editable. All images should be at least 300 dpi resolution (when figures are scaled to approximately the size that they are to be printed at) and in RGB colour format. Please do not submit Jpeg or flattened TIFF files. Please see also 'Guidelines for Electronic Submission of Figures' at the end of this letter for further detail.

Please view http://www.nature.com/authors/editorial_policies/image.html for more detailed guidelines.

14. We will edit your figures/tables electronically so they conform to Nature Microbiology style. If necessary, we will re-size figures to fit single or double column width. If your figures contain several parts, the parts should be labelled lower case a, b, and so on, and form a neat rectangle when assembled.

15. Please check the PDF of the whole paper and figures (on our manuscript tracking system) VERY CAREFULLY when you submit the revised manuscript. This will be used as the 'reference copy' to make sure no details (such as Greek letters or symbols) have gone missing during file-transfer/conversion and re-drawing.

16. All Supplementary Information must be submitted in accordance with the instructions in the attached Inventory of Supporting Information, and should fit into one of three categories:

1. EXTENDED DATA: Extended Data are an integral part of the paper and only data that directly contribute to the main message should be presented. These figures will be integrated into the full-text HTML version of your paper and will be appended to the online PDF. There is a limit of 10 Extended Data figures, and each must be referred to in the main text. Each Extended Data figure should be of the same quality as the main figures, and should be supplied at a size that will allow both the figure and legend to be presented on a single legal-sized page. Each figure should be submitted as an

individual .jpg, .tif or .eps file with a maximum size of 10 MB each. All Extended Data figure legends must be provided in the attached Inventory of Accessory Information, not in the figure files themselves.

2. SUPPLEMENTARY INFORMATION: Supplementary Information is material that is essential background to the study but which is not practical to include in the printed version of the paper (for example, video files, large data sets and calculations). Each item must be referred to in the main manuscript and detailed in the attached Inventory of Accessory Information. Tables containing large data sets should be in Excel format, with the table number and title included within the body of the table. All textual information and any additional Supplementary Figures (which should be presented with the legends directly below each figure) should be provided as a single, combined PDF. Please note that we cannot accept resupplies of Supplementary Information after the paper has been formally accepted unless there has been a critical scientific error.

All Extended Data must be called out in your manuscript and cited as Extended Data 1, Extended Data 2, etc. Additional Supplementary Figures (if permitted) and other items are not required to be called out in your manuscript text, but should be numerically numbered, starting at one, as Supplementary Figure 1, not SI1, etc.

3. SOURCE DATA: We encourage you to provide source data for your figures whenever possible. Full-length, unprocessed gels and blots must be provided as source data for any relevant figures, and should be provided as individual PDF files for each figure containing all supporting blots and/or gels with the linked figure noted directly in the file. Statistics source data should be provided in Excel format, one file for each relevant figure, with the linked figure noted directly in the file. For imaging source data, we encourage deposition to a relevant repository, such as figshare (<https://figshare.com/>) or the Image Data Resource (<https://idr.openmicroscopy.org>).

17. Nature Research journals [encourage authors to share their step-by-step experimental protocols](https://www.nature.com/nature-research/editorial-policies/reporting-standards#protocols) on a protocol sharing platform of their choice. Nature Research's Protocol Exchange is a free-to-use and open resource for protocols; protocols deposited in Protocol Exchange are citable and can be linked from the published article. More details can found at www.nature.com/protocolexchange/about.

18. Please note that after the paper has been formally accepted you can only provide amended Supplementary Information files for critical changes to the scientific content, not for style. You should clearly explain what changes have been made if you do resupply any such files.

18. It is a condition of publication that you include a statement before the acknowledgements naming the author to whom correspondence and requests for materials should be addressed.

19. Finally, we require authors to include a statement of their individual contributions to the paper -- such as experimental work, project planning, data analysis, etc. -- immediately after the acknowledgements. The statement should be short, and refer to authors by their initials. For details please see the Authorship section of our joint Editorial policies at http://www.nature.com/authors/editorial_policies/authorship.html

20. Please use the checks provided by our partner SNTPS to amend any legends, figures or the reporting checklist. We require resubmitted and finalised checklist for publication.

We will not send your revised paper for further review if, in the editors' judgement, the referees' comments on the present version have been addressed. If the revised paper is in Nature Microbiology format, in accessible style and of appropriate length, we shall accept it for publication immediately.

Please resubmit electronically

- * the final version of the text (not including the figures) in either Word or Latex.
- * publication-quality figures. For more details, please refer to our Figure Guidelines, which is available here: https://mts-nmicrobiol.nature.com/letters/Figure_guidelines.pdf
- * Extended Data & Supplementary Information, as instructed
- * a point-by-point response to any issues raised by our referees and to any editorial suggestions.
- * any suggestions for cover illustrations, which should be provided at high resolution as electronic files. Please note that such pictures should be selected more for their aesthetic appeal than for their scientific content. I am sure you will understand that we cannot make any promise as to whether any of your suggestions might be selected for the cover of Nature Microbiology.

Please use the following link to access your home page:

{REDACTED}

* This url links to your confidential homepage and associated information about manuscripts you may have submitted or be reviewing for us. If you wish to forward this e-mail to co-authors, please delete this link to your homepage first.

Please also send the following forms as a PDF by email to microbiology@nature.com.

* Please sign and return the <http://www.nature.com/documents/snl-ltp.docx> target="_blank">Licence to Publish form .

* Or, if the corresponding author is either a Crown government employee (including Great Britain and Northern Ireland, Canada and Australia), or a US Government employee, please sign and return the <http://www.nature.com/documents/snl-ltp-crown.docx> target="_blank"> Licence to Publish form for Crown government employees, or a <http://www.nature.com/documents/snl-ltp-govus.docx> target="_blank"> Licence to Publish form for US government employees.

* Should your Article contain any items (figures, tables, images, videos or text boxes) that are the same as (or are adaptations of) items that have previously been published elsewhere and/or are owned by a third party, please note that it is your responsibility to obtain the right to use such items and to give proper attribution to the copyright holder. This includes pictures taken by professional photographers and images downloaded from the internet. If you do not hold the copyright for any such item (in whole or part) that is included in your paper, please complete and return this .

ORCID

Nature Microbiology is committed to improving transparency in authorship. As part of our efforts in this direction, we are now requesting that all authors identified as 'corresponding author' create and link their Open Researcher and Contributor Identifier (ORCID) with their account on the Manuscript Tracking System (MTS) prior to acceptance. ORCID helps the scientific community achieve unambiguous attribution of all scholarly contributions. For more information please visit <http://www.springernature.com/orcid>

For all corresponding authors listed on the manuscript, please follow the instructions in the link below to link your ORCID to your account on our MTS before submitting the final version of the manuscript. If you do not yet have an ORCID you will be able to create one in minutes.
<https://www.springernature.com/gp/researchers/orcid/orcid-for-nature-research>

IMPORTANT: All authors identified as 'corresponding author' on the manuscript must follow these instructions. Non-corresponding authors do not have to link their ORCIDs but are encouraged to do so. Please note that it will not be possible to add/modify ORCIDs at proof. Thus, if they wish to have their ORCID added to the paper they must also follow the above procedure prior to acceptance.

To support ORCID's aims, we only allow a single ORCID identifier to be attached to one account. If you have any issues attaching an ORCID identifier to your MTS account, please contact the [Platform Support Helpdesk](http://platformsupport.nature.com/).

Nature Research journals [encourage authors to share their step-by-step experimental protocols](https://www.nature.com/nature-research/editorial-policies/reporting-standards#protocols) on a protocol sharing platform of their choice. Nature Research's Protocol Exchange is a free-to-use and open resource for protocols; protocols deposited in Protocol Exchange are citable and can be linked from the published article. More details can found at www.nature.com/protocolexchange/about.

We hope that you will support this initiative and supply the required information. Should you have any query or comments, please do not hesitate to contact me.

We hope to hear from you within two weeks; please let us know if the revision process is likely to take longer.

Yours sincerely,

Reviewer Comments:

Reviewer #1 (Remarks to the Author):

Dear Authors,

You have addressed all my comments very well. Many researchers interested in mammalian gut microbiomes and the efficiency of bioreactors will most likely work more on Fungi based on your results presented in this paper.

Kind regards

Reviewer #3 (Remarks to the Author):

This is a much improved manuscript, there is now much greater context provided for the reader. The authors have addressed most of my initial comments and questions. Remaining comments:

1. Regarding the lack of statistics in the initial manuscript, stats are still required to support the results in Figures 4 and 5. I cannot see any stats presented in Supplementary Figures 6, 8 or 17. For supplementary figure 16 panel b please present the actual p-values. It is not clear what the letters a and b mean in this figure.

2. Regarding what is now line 29: "94% of which were previously uncultured novel microbes". This line implies that previously these microbes were uncultured but now they are not. As this describes MAGS and not pure isolates, then these microbes remain uncultured. Changing the word 'uncultured' to 'unidentified' would be more correct.

3. Line 86- please provide references

Reviewer #4 (Remarks to the Author):

The authors did a good job in addressing the (widely varying) concerns of all four reviewers. I am happy to see that in most cases they have the reviewers concerns seriously and changed the manuscript to the present shape. I have no additional comments on the paper.

Final Decision Letter:

Dear Professor O'Malley,

I am pleased to accept your Article "Genomic and functional analyses of fungal and bacterial consortia that enable lignocellulose breakdown in goat gut microbiomes" for publication in Nature Microbiology. Thank you for having chosen to submit your work to us it is my pleasure to be able to publish this manuscript.

Before your manuscript is typeset, we will edit the text and look particularly carefully at the titles of all papers to ensure that they are relatively brief and understandable.

The subeditor may send you the edited text for your approval. Once your manuscript is typeset you will receive a link to your electronic proof via email within 20 working days, with a request to make any corrections within 48 hours. If you have queries at any point during the production process then please contact the production team at rjsproduction@springernature.com. Once your paper has been scheduled for online publication, the Nature press office will be in touch to confirm the details.

Acceptance of your manuscript is conditional on all authors' agreement with our publication policies (see www.nature.com/nmicrobiolate/authors/gta/content-type/index.html). In particular your manuscript must not be published elsewhere and there must be no announcement of the work to any media outlet until the publication date (the day on which it is uploaded onto our website).

The Author's Accepted Manuscript (the accepted version of the manuscript as submitted by the author) may only be posted 6 months after the paper is published, consistent with our [self-archiving embargo](http://www.nature.com/authors/policies/license.html). Please note that the Author's Accepted Manuscript may not be released under a Creative Commons license. For Nature Research Terms of Reuse of archived manuscripts please see: <http://www.nature.com/authors/policies/license.html#terms>

We welcome the submission of potential cover material (including a short caption of around 40 words) related to your manuscript; suggestions should be sent to Nature Microbiology as electronic files (the image should be 300 dpi at 210 x 297 mm in either TIFF or JPEG format). Please note that such pictures should be selected more for their aesthetic appeal than for their scientific content, and that

colour images work better than black and white or grayscale images. Please do not try to design a cover with the Nature Microbiology logo etc., and please do not submit composites of images related to your work. I am sure you will understand that we cannot make any promise as to whether any of your suggestions might be selected for the cover of the journal.
